# Optimality and Adaptivity of Deep Neural Features for Instrumental Variable Regression

**Juno Kim**[1,2*] **Dimitri Meunier**[3] **Arthur Gretton**[3] **Taiji Suzuki**[1,2] **Zhu Li**[3]

[1]Department of Mathematical Informatics, University of Tokyo
[2]Center for Advanced Intelligence Project, RIKEN
[3]Gatsby Computational Neuroscience Unit, University College London

## ABSTRACT

We provide a convergence analysis of *deep feature instrumental variable* (DFIV) regression (Xu et al., 2021), a nonparametric approach to IV regression using data-adaptive features learned by deep neural networks in two stages. We prove that the DFIV algorithm achieves the minimax optimal learning rate when the target structural function lies in a Besov space. This is shown under standard nonparametric IV assumptions, and an additional smoothness assumption on the regularity of the conditional distribution of the covariate given the instrument, which controls the difficulty of Stage 1. We further demonstrate that DFIV, as a data-adaptive algorithm, is superior to fixed-feature (kernel or sieve) IV methods in two ways. First, when the target function possesses low spatial homogeneity (i.e., it has both smooth and spiky/discontinuous regions), DFIV still achieves the optimal rate, while fixed-feature methods are shown to be strictly suboptimal. Second, comparing with kernel-based two-stage regression estimators, DFIV is provably more data efficient in the Stage 1 samples.

## 1 INTRODUCTION

We study the nonparametric instrumental variable (NPIV) regression problem (Newey & Powell, 2003; Ai & Chen, 2003; Darolles et al., 2011). For random variables $X$, $Y$, and $\xi$, we have

$$Y = f_{\text{str}}(X) + \xi, \quad \mathbb{E}[\xi|X] \neq 0, \tag{1}$$

where $X \in \mathcal{X}$ is the endogenous variable, $Y$ is the outcome, and $\xi$ denotes unobserved confounding which affects both $X$ and $Y$. The central object of interest is the structural function $f_{\text{str}}$: this may be estimated by utilizing an exogenous *instrumental variable* $Z \in \mathcal{Z}$, which satisfies

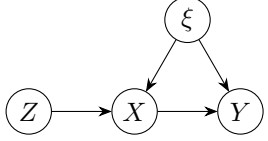

Figure 1: Causal graph of IV.

$$\mathbb{E}[\xi|Z] = 0, \quad Z \not\perp\!\!\!\perp X, \quad (Z \perp\!\!\!\perp Y)_{G_{\bar{X}}} \tag{2}$$

(the second requirement is the *relevance* assumption, and the final requirement denotes independence of $Z$ and $Y$ when incoming edges to $X$ are removed in the graphical model, thus excluding a hidden common cause of $Z$ and $Y$; see Figure 1). The IV setting can be understood in terms of an example (Angrist & Krueger, 2001): let $X$ denote the price of coffee, $Y$ denote coffee consumption, and $\xi$ denote the level of demand, unknown to the coffee vendor (e.g. increased demand due to conference deadlines), which affects both $X$ and $Y$. In this case, $Z$ might be the wholesale price of coffee beans, which influences the price of coffee and is assumed to be independent of $\xi$.

Given (1) and (2), taking the conditional expectation w.r.t. $Z$, $f_{\text{str}}$ satisfies the functional equation $Tf_{\text{str}} = \mathbb{E}[Y|Z]$, where $\mathcal{P}_{\mathcal{X}}, \mathcal{P}_{\mathcal{Z}}$ are the marginal laws of $X, Z$, and $T : L^2(\mathcal{P}_{\mathcal{X}}) \to L^2(\mathcal{P}_{\mathcal{Z}})$ is the bounded linear operator that maps each $h \in L^2(\mathcal{P}_{\mathcal{X}})$ to $\mathbb{E}[h(X)|Z] \in L^2(\mathcal{P}_{\mathcal{Z}})$ (Darolles et al., 2011; Bennett et al., 2023b). The aim of NPIV is to solve for this possibly under-determined system, where $f_{\text{str}}$ is uniquely identified if and only if $T$ is injective. This task is inherently challenging due to its ill-posed nature. For example, a small perturbation in the outcome can lead to estimators that are far

---

*Correspondence to: <junokim@berkeley.edu>

Table 1: Summary of main results for IV regression, when the structural function lies in a Besov space $B_{p,q}^s(\mathcal{X})$. $m, n$ denote the number of Stage 1, 2 samples, respectively. The constants $\gamma_1 \leq \gamma_0$ denote the decay rates of the link and reverse link conditions and the degree of separation $\Delta = d_x(1/p - 1/2)_+$. The upper bound of DFIV assumes $T$ has *maximal smoothness* (see Lemma 3.2), while the projected fixed-feature lower bound assumes $d_z \leq d_x$ and $d' := d_z \vee (d_x - 2(s - \Delta))$.

| Rates | DFIV | Optimal | Fixed-feature |
|---|---|---|---|
| Projected | $\widetilde{O}\left(m^{-\frac{2s+2\gamma_1}{2s+2\gamma_1+d_x}} + n^{-\frac{2s+2\gamma_0}{2s+2\gamma_0+d_x}}\right)$ | $\Omega\left(n^{-\frac{2s+2\gamma_0}{2s+2\gamma_1+d_x}}\right)$ | $\widetilde{\Omega}\left(n^{-\frac{2(s-\Delta)+2\gamma_0}{2(s-\Delta)+2\gamma_1+d'}}\right)$ |
| Full | $\widetilde{O}\left(m^{-\frac{2(s-\gamma_0+\gamma_1)}{2s+2\gamma_1+d_x}} + n^{-\frac{2(s-\gamma_0+\gamma_1)}{2s+2\gamma_0+d_x}\frac{s+\gamma_0}{s+\gamma_1}}\right)$ | $\Omega\left(n^{-\frac{2s}{2s+2\gamma_1+d_x}}\right)$ | $\Omega\left(n^{-\frac{2(s-\Delta)}{2(s-\Delta)+2\gamma_1+d_x}}\right)$ |

from the true solution as $T^{-1}$ is often unbounded (Carrasco et al., 2007). There has been a growing interest in addressing this challenge using modern statistical and machine learning techniques.

NPIV estimation plays a crucial role in various fields, including causal inference (Angrist & Imbens, 1995; Newey & Powell, 2003), addressing missing data challenges (Wang et al., 2014; Miao et al., 2015), and reinforcement learning (Liao et al., 2021; Uehara et al., 2021; Xu et al., 2021; Chen et al., 2022). Existing methods can be broadly categorized into two main approaches: conditional moments methods (Muandet et al., 2020; Liao et al., 2020; Dikkala et al., 2020; Bennett et al., 2019; 2023a;c) and two-stage estimation techniques (Newey & Powell, 2003; Carrasco et al., 2007; Chen, 2007; Horowitz, 2011; Darolles et al., 2011; Hartford et al., 2017; Chen & Christensen, 2018; Singh et al., 2019; Xu et al., 2021; Ren et al., 2024; Wang et al., 2022; Meunier et al., 2024). Conditional moment methods approach NPIV by constructing a min-max optimization problem of the form $\min_{f \in \mathcal{F}} \max_{g \in \mathcal{G}} \mathcal{L}(f, g)$ for certain function classes $\mathcal{F}, \mathcal{G}$. However, obtaining these estimators can be difficult in practice since the solutions are typically saddle points. Furthermore, information-theoretic lower bounds for these methods have not yet been established, so it is unknown whether the proposed estimators can achieve the optimal learning rate.

In the present work, we consider two-stage methods. These decompose NPIV into the following steps: Stage 1 learns either the conditional expectation operator $T$ or the conditional density $X \mid Z$ depending on the algorithm. In Stage 2, the outcome $Y$ is regressed using the estimator obtained from Stage 1. While this offers more stability compared to conditional moment methods by avoiding saddle-point optimization, a key challenge remains to represent the conditional distribution $X \mid Z$ from the first stage. One approach is to explicitly learn the conditional density $X \mid Z$ from data (Darolles et al., 2011; Hartford et al., 2017; Li et al., 2024b). Density estimation is challenging when $X$ and $Z$ are high dimensional, however, and convergence rates can be correspondingly slow (see e.g. Wasserman, 2006, for convergence properties of density estimates).

A second approach is to learn the conditional mean of features of $X$ given $Z$, where the features of $X$ are the input features of Stage 1 (see e.g., Chen & Reiss, 2011; Singh et al., 2019; Chen & Christensen, 2018; Meunier et al., 2024). This approach, known as two-stage least-squares (2SLS) regression, has the advantage that the Stage 1 problem is only as difficult as it needs to be in order to solve Stage 2, and does not require to address the harder problem of conditional density estimation. The question of feature dictionary choice remains a challenge, however, and the above methods employ dictionaries for classes of smooth functions (i.e., spline and RKHS classes), which are not data adaptive. More recently, two-stage IV approaches have been proposed with data-adaptive feature dictionaries represened by neural nets (Xu et al., 2021; Ren et al., 2024; Wang et al., 2022). While these have shown strong empirical performance, the corresponding theoretical guarantees remain incomplete, and minimax rates are yet to be established. Finally, while sample splitting is predominantly used in two-stage NPIV, the ratio between Stage 1 ($m$) and Stage 2 ($n$) samples is rarely studied in the literature. Singh et al. (2019) and Meunier et al. (2024) recently showed that $m > n$ is required in order to achieve the minimax optimal rate in the case of fixed-feature estimators.

**Our contributions.** We study the *deep feature instrumental variable* (DFIV) algorithm proposed by Xu et al. (2021), where learnable deep neural network (DNN) features are employed to jointly optimize both stages. As a data-adaptive estimator, DFIV exhibits superior empirical performance

compared with fixed-feature methods, most notably in cases where the structural function may have both smooth and non-smooth regions (Xu et al., 2021; Chen et al., 2022). In this work, we provide minimax convergence guarantees for DFIV, and formally characterize the conditions under which deep neural features yield a performance advantage over fixed-dictionary approaches. These build on existing statistical analyses of DNNs (e.g. Suzuki, 2019; Hayakawa & Suzuki, 2020), which study ordinary regression settings, by contrast with our two-stage setting where two dependent DNNs must be jointly learned. Our result are as follows:

- We prove an upper bound for the risk of the DFIV estimator $\hat{f}_{\text{str}}$ in terms of the number $m, n$ of Stage 1 and 2 samples, respectively, when $f_{\text{str}}$ lies in a Besov space $B^s_{p,q}(\mathcal{X})$. We obtain rates with respect to both the projected pseudometric $\|T\hat{f}_{\text{str}} - Tf_{\text{str}}\|^2_{L^2(\mathcal{P}_{\mathcal{Z}})}$ (Theorem 3.1) and the full non-projected metric $\|\hat{f}_{\text{str}} - f_{\text{str}}\|^2_{L^2(\mathcal{P}_{\mathcal{X}})}$ (Theorem 3.5).

- We obtain information-theoretic lower bounds for NPIV in Besov spaces and demonstrate that DFIV can achieve the minimax optimal rate (Proposition 3.3, 3.7). Moreover, we show that DFIV can attain the optimal rate with Stage 1 samples $m \approx n$, whereas kernel IV estimators require $m/n \to \infty$ (Singh et al., 2019; Meunier et al., 2024).

- Our rates recover known rates for smooth $f_{\text{str}}$, established previously (Chen & Reiss, 2011; Meunier et al., 2024) with fixed-feature estimators. More importantly, we demonstrate a strict separation between DFIV and any fixed-feature estimator (Chen & Reiss, 2011; Meunier et al., 2024) when the target function has both smooth and spiky/discontinuous regions (Corollary 4.2).

A detailed discussion of prior two-stage approaches, and of relevant works addressing convergence of regression with DNNs, is provided in Appendix A. A summary of our rates is given in Table 1. The rest of the paper is structured as follows: In Section 2, we outline the DFIV algorithm and define our target and hypothesis classes. The upper bounds for DFIV and comparison to NPIV minimax lower bounds are presented in Section 3. The separation between fixed-feature methods and DFIV is established in Section 4. All proofs are deferred to the appendix.

## 2 BACKGROUND AND DFIV ALGORITHM

### 2.1 INSTRUMENTAL VARIABLE REGRESSION

Nonparametric instrumental variable (NPIV) regression typically has a treatment $X \in \mathcal{X}$ and outcome $Y \in \mathbb{R}$, related via the structural function $f_{\text{str}} : \mathcal{X} \to \mathbb{R}$ and an unobserved $\xi$ that affects both $X, Y$:

$$Y = f_{\text{str}}(X) + \xi, \quad \mathbb{E}[\xi] = 0, \quad \mathbb{E}[\xi|X] \neq 0. \tag{3}$$

This implies that $f_{\text{str}}(x) \neq \mathbb{E}[Y|X = x]$ so that ordinary supervised regression methods cannot be used. Instead, we introduce an instrumental variable $Z \in \mathcal{Z}$ known to be uncorrelated with $\xi$, that is $\mathbb{E}[\xi|Z] = 0$, and satisfying the requirements (2). For simplicity we assume that $X, Z$ are bounded and set $\mathcal{X} = [0,1]^{d_x}$, $\mathcal{Z} = [0,1]^{d_z}$, equipped with the induced probability measures $\mathcal{P}_{\mathcal{X}}, \mathcal{P}_{\mathcal{Z}}$. By defining the projection operator

$$T : L^2(\mathcal{P}_{\mathcal{X}}) \to L^2(\mathcal{P}_{\mathcal{Z}}), \quad (Tf)(Z) = \mathbb{E}[f(X)|Z],$$

eq. (3) can be described as the nonparametric indirect regression (NPIR) model (Chen & Reiss, 2011)

$$Y = Tf_{\text{str}}(Z) + \eta, \quad \eta = f_{\text{str}}(X) - Tf_{\text{str}}(Z) + \xi, \tag{4}$$

where $\mathbb{E}[\eta|Z] = 0$, and hence we seek to solve the inverse problem $Tf_{\text{str}} = \mathbb{E}[Y|Z]$. The problem is generally ill-posed, however; the solution may be ill-behaved or not unique (Nashed & Wahba, 1974; Carrasco et al., 2007). To analyze this problem, we require a mild regularity of the noise.

**Assumption 1.** *(i) There exists $\sigma_1 > 0$ such that $\eta|(Z = z)$ is $\sigma_1$-subgaussian for all $z \in \mathcal{Z}$.*

*(ii) (For lower bound only) There exists $\sigma_0 > 0$ such that $\text{Var}(\eta|Z = z) \geq \sigma_0^2$ and the KL divergence between $\eta|(Z = z)$, $\mu + \eta|(Z = z)$ is bounded above by $\frac{\mu^2}{2\sigma_0^2}$ for all $z \in \mathcal{Z}, \mu \in \mathbb{R}$.*

This is satisfied for example if $\eta|Z = z$ is $N(0, \sigma^2(z))$-distributed with $\sigma_0 \leq \sigma(\cdot) \leq \sigma_1$, which is a standard assumption in the literature (Bissantz et al., 2007; Chen & Reiss, 2011).

**Besov spaces.** In order to facilitate a learning-theoretic analysis, we suppose that the structural function $f_{\text{str}}$ belongs to (the unit ball of) a Besov space $B_{p,q}^s(\mathcal{X})$ of smoothness $s$ on $\mathcal{X}$. Besov spaces are a well-studied class of functions of generalized smoothness, and include Hölder spaces, Sobolev spaces, and classes of bounded variation. See Definition A.1 for details. In particular, Besov spaces with smoothness below the threshold $d_x/p$ contain functions with discontinuities which are more difficult to learn; our theory accounts for this regime. We also assume $s > d_x(1/p - 1/2)_+$ so that $B_{p,q}^s(\mathcal{X})$ continuously embeds into $L^2(\mathcal{X})$.

**Assumption 2** (Structural function). $f_{\text{str}} \in \mathbb{U}(B_{p,q}^s(\mathcal{X}))$ and $|f_{\text{str}}| \le C$ for some $0 < p, q \le \infty$, $s > d_x(1/p - 1/2)_+$ and $C > 0$. If $s \le d_x/p$, $\mathcal{P}_\mathcal{X}$ also has Lebesgue density bounded above.

Our analysis utilizes the B-spline system, which forms a *hierarchical* basis for Besov spaces: the span of B-splines up to resolution $k$ is strictly contained in the span of B-splines at resolution $k+1$.

**Definition 2.1** (B-spline basis). The *cardinal B-spline* of order $r \in \mathbb{N}$ is iteratively defined as the repeated convolution $\iota_r = \iota_{r-1} * \iota_0 \in C(\mathbb{R})$ where $\iota_0(x) = 1_{[0,1]}(x)$. The *tensor product B-spline* of order $r$ with resolution $k \in \mathbb{Z}_{\ge 0}$ and location $\ell \in I_k = \prod_{i=1}^{d_x}\{-r, -r+1, \cdots, 2^k\}$ is

$$\omega_{k,\ell}(x) = \prod_{i=1}^{d_x} \iota_r(2^k x_i - \ell_i).$$

## 2.2 Two-Stage Least Squares Regression

A popular method for performing IV analysis is two-stage least squares (2SLS) regression. In 2SLS, the structural function $f_{\text{str}}(x)$ is modeled as a linear combination $u^\top \psi(x)$ of fixed basis functions $\psi(x)$, for instance Hermite polynomials (Newey & Powell, 2003) or reproducing kernel Hilbert spaces (RKHS) (Singh et al., 2019; Meunier et al., 2024). The weight vector $u$ is estimated by performing two successive regressions. In Stage 1, the conditional mean embedding (CME) $\mathbb{E}_{X|Z}[\psi(X)]$ (Song et al., 2009; Park & Muandet, 2020; Klebanov et al., 2020) is approximated by another set of basis functions $V\phi(Z)$, where the coefficient matrix $V$ is computed by solving the vector-valued regression (Grünewälder et al., 2012; Mollenhauer & Koltai, 2020; Li et al., 2022; 2024a):

$$\hat{V} = \arg\min_V \mathbb{E}_{X,Z}[\|\psi(X) - V\phi(Z)\|^2] + \lambda_1 \|V\|^2. \tag{5}$$

In Stage 2, we obtain $u$ by regressing $Y$ against $\mathbb{E}_{X|Z}[f_{\text{str}}(X)] \approx u^\top \hat{V}\phi(Z)$, where the Stage 1 CME estimate is utilized:

$$\hat{u} = \arg\min_u \mathbb{E}_{Y,Z}[\|Y - u^\top \hat{V}\phi(Z)\|^2] + \lambda_2 \|u\|^2. \tag{6}$$

Finally, the structural function is estimated as $\hat{f}_{\text{str}}(x) = \hat{u}^\top \psi(x)$. Both stages can be solved in closed form by performing ridge regression, including for infinite (RKHS) feature dictionaries.

## 2.3 Deep Feature Instrumental Variable Regression

We now recall the *deep feature instrumental variable* (DFIV) regression algorithm (Xu et al., 2021), which extends the 2SLS framework to incorporate learnable feature representations $\psi_{\theta_x}(x)$, $\phi_{\theta_z}(z)$ in both stages. The respective hypothesis classes, equipped with the $L^\infty$-norm, are denoted as

$$\mathcal{F}_x = \{\psi_{\theta_x} : \mathcal{X} \to \mathbb{R} \mid \theta_x \in \Theta_x\}, \quad \mathcal{F}_z = \{\phi_{\theta_z} : \mathcal{Z} \to \mathbb{R} \mid \theta_z \in \Theta_z\}.$$

As in Singh et al. (2019); Meunier et al. (2024); Xu et al. (2021), we do not presume access to samples from the joint law $(X, Y, Z)$, and instead are provided with $m$ i.i.d. samples $\mathcal{D}_1 = \{(x_i, z_i)\}_{i=1}^m$ from $(X, Z)$ for Stage 1, and $n$ i.i.d. samples $\mathcal{D}_2 = \{(\tilde{y}_i, \tilde{z}_i)\}_{i=1}^n$ from $(Y, Z)$ for Stage 2.

**Stage 1 regression:** Given data $\mathcal{D}_1$ and fixed Stage 2 parameter $\theta_x$, the conditional expectation $\mathbb{E}_{X|Z}[\psi_{\theta_x}(X)]$ is learned using the network $\phi_{\theta_z}(Z)$ by minimizing the empirical loss,

$$\hat{\theta}_z = \hat{\theta}_z(\theta_x) = \arg\min_{\theta_z \in \Theta_z} \frac{1}{m} \sum_{i=1}^m (\psi_{\theta_x}(x_i) - \phi_{\theta_z}(z_i))^2. \tag{7}$$

The resulting estimate is also denoted as $\hat{\mathbb{E}}_{X|Z}[\psi_{\theta_x}] := \phi_{\hat{\theta}_z(\theta_x)}(\cdot)$ to emphasize that $\hat{\theta}_z$ is a function of the current Stage 2 input feature parameters $\theta_x$.

**Stage 2 regression:** Given data $\mathcal{D}_2$ and the Stage 1 estimate $\phi_{\hat{\theta}_z}$, we solve the following regression where $R : \Theta_x \to \mathbb{R}_{\geq 0}$ is an optional regularizer:

$$\hat{\theta}_x = \arg\min_{\theta_x \in \Theta_x} \frac{1}{n} \sum_{i=1}^{n} \left( \tilde{y}_i - \hat{\mathbb{E}}_{X|Z}[\psi_{\theta_x}](\tilde{z}_i) \right)^2 + \lambda R(\theta_x). \tag{8}$$

The final DFIV for estimate $f_{\mathrm{str}}$ is returned as $\hat{f}_{\mathrm{str}} = \psi_{\hat{\theta}_x}$.[1]

**Remark 2.2.** We note that jointly optimizing (7) and (8) is challenging as the optimal $\hat{\theta}_z$ is itself a function of the current $\theta_x$. The DFIV algorithm as originally proposed in Xu et al. (2021) takes $\psi_{\theta_x}, \phi_{\theta_z}$ to be vector-valued mappings, and treats separately the linear maps $u, V$ (eqs. (5) & (6)) via ridge regression. This enables more effective optimization by partially backpropagating $\theta_x$ through the analytical solutions of $u, V$; see Section 3 of Xu et al. (2021) for details. From the sample complexity viewpoint, however, it suffices to consider the scalar output $u^\top \psi_{\theta_x}$ as a single neural network and learn its conditional expectation in Stage 1. A more sophisticated bilevel optimization approach is described by Petrulionyte et al. (2024).

**Smooth DNN class.** In our paper, $\psi_{\theta_x}, \phi_{\theta_z}$ are chosen from a class of smooth DNNs, defined with depth $L$, width $W$, sparsity $S$, norm bound $M$ and a $C^r$ activation $\sigma$ (applied elementwise) as

$$\mathcal{F}_{\mathrm{DNN}}(L, W, S, M) := \Big\{ \mathrm{clip}_{C,\bar{C}} \circ (\mathbf{W}^{(L)}\sigma + b^{(L)}) \circ \cdots \circ (\mathbf{W}^{(1)}\mathrm{id}_{\mathcal{X}} + b^{(1)}) : \mathcal{X} \to \mathbb{R} \ \Big|$$
$$\mathbf{W}^{(1)} \in \mathbb{R}^{W \times d}, \mathbf{W}^{(\ell)} \in \mathbb{R}^{W \times W}, \mathbf{W}^{(L)} \in \mathbb{R}^{W}, b^{(\ell)} \in \mathbb{R}^{W}, b^{(L)} \in \mathbb{R},$$
$$\textstyle\sum_{\ell=1}^{L} \|\mathbf{W}^{(\ell)}\|_0 + \|b^{(\ell)}\|_0 \leq S, \max_{\ell \leq L} \|\mathbf{W}^{(\ell)}\|_\infty \vee \|b^{(\ell)}\|_\infty \leq M \Big\}.$$

Here $\mathrm{clip}_{C,\bar{C}} : \mathbb{R} \to [-\bar{C}, \bar{C}]$ is any bounded, 1-Lipschitz $C^\infty$ function equal to the identity when $|x| \leq \underline{C}$; we fix any $\bar{C} > C > \underline{C}$ and omit them from the notation. The use of smooth activations and smooth clipping is in order to ensure $\mathcal{F}_{\mathrm{DNN}} \subset W_p^r(\mathcal{X}) \subset B_{p,q}^s(\mathcal{X})$ (Suzuki, 2019) and thus to obtain Besov norm guarantees for the DNN estimator. Moreover, we specifically consider sigmoid activations $\sigma(x) = (1 + e^{-x})^{-1}$ for ease of analysis, however the results can be extended to any sufficiently smooth activation e.g. SiLU, GELU, Softplus, as well as piecewise polynomial activations such as ReQU; see the discussion in Appendix C. Some of our results also hold when the ReLU DNN class, defined with activation $\sigma(x) = 0 \vee x$ and $\mathrm{clip}_{\bar{C}}(x) = (-\bar{C}) \vee (x \wedge \bar{C})$, is used instead.

## 3 THEORETICAL ANALYSIS OF DFIV

We now establish the sample complexity of DFIV when $f_{\mathrm{str}}$ lives in the Besov space $B_{p,q}^s(\mathcal{X})$ with smoothness $s$. We obtain upper and lower bounds for both the projected error (Section 3.2) and non-projected error with $p \geq 2$ (Section 3.3), and obtain precise conditions for when the minimax optimal rate is achieved. We begin by stating our assumptions on the projection operator $T$.

### 3.1 LINK AND SMOOTHNESS CONDITIONS

The operator $T$ plays a crucial role in determining the sample complexity due to the following factors.

- The NPIV model (4) shows that Stage 2 essentially learns $f_{\mathrm{str}}$ using data from the projected function $T f_{\mathrm{str}}$, leading to a loss of information. Hence the rate will worsen depending on how contractive $T$ is in an $L^2$-sense, which is usually quantified by a *link condition* (see e.g., Nair et al., 2005; Chen & Reiss, 2011; Meunier et al., 2024), here given as Assumption 3.

- The difficulty of Stage 1, where we estimate the conditional expectation $T\psi_{\theta_x}$ for the output $\psi_{\theta_x}$ of Stage 2, depends on the regularity properties of $Tf$ for $f$ in certain function classes. We will control this using a novel smoothness condition (Assumption 4 or 4*).

---

[1]A minimizer of Stage 1 always exists as we take $\Theta_z$ to be compact and (7) is continuous w.r.t. $\theta_z$. A minimizer of Stage 2 is not guaranteed to exist as the output $\hat{\theta}_z$ of Stage 1 may not depend continuously on $\theta_x$, but we may simply take an approximate minimizer $\epsilon$-close to the infimum and obtain the same results as $\epsilon \to 0$.

- To obtain the final non-projected rate $\|\hat{f}_{\text{str}} - f_{\text{str}}\|_{L^2(\mathcal{P}_\mathcal{X})}$, we must start with the projected rate $\|T\hat{f}_{\text{str}} - Tf_{\text{str}}\|_{L^2(\mathcal{P}_\mathcal{Z})}$ and then use a *reverse link condition* (Assumption 6) to derive the former.

We now state and discuss each assumption in full. Denote by $P_r^{/k} = \text{span}\{\omega_{k,\ell} \mid \ell \in I_k\} \subset B_{p,q}^s(\mathcal{X})$ the linear span of all B-splines of order $r$ up to resolution $k$ and $\Pi_r^{/k}$ the projection operator to $P_r^{/k}$.

**Assumption 3** (link condition). *There exists $\gamma_1 \geq 0$ such that for all $f \in B_{p,q}^s(\mathcal{X})$,*

$$\|T(f - \Pi_r^{/k} f)\|_{L^2(\mathcal{P}_\mathcal{Z})} \lesssim 2^{-\gamma_1 k} \|f - \Pi_r^{/k} f\|_{L^2(\mathcal{P}_\mathcal{X})}. \tag{9}$$

That is, $T$ is more contractive at higher resolutions. Assumption 3, and the reverse link condition Assumption 6, are equivalent to or follow from standard conditions imposed in the IV literature; for example, they follow from the link condition in Chen & Reiss (2011) and Meunier et al. (2024) stated in terms of the Hilbert scale generated by a certain operator.[2] Note that $\text{rank } \Pi_r^{/k} = \dim P_r^{/k} \asymp 2^{kd_x}$, hence (9) corresponds to polynomial (mildly ill-posed) rather than exponential (severely ill-posed) decay w.r.t. the number of basis elements (Blundell et al., 2007).

We now discuss the issue of smoothness. Stage 1 aims to learn the conditional expectation $T\psi_{\hat{\theta}_x}$ for the data-dependent predictor $\psi_{\hat{\theta}_x} \in \mathcal{F}_x$ with the DNN class $\mathcal{F}_z$. This is a separate regression problem on $\mathcal{Z}$, so its risk depends on the smoothness of $T\psi_{\hat{\theta}_x}$; a nonsmooth predictor $\psi_{\hat{\theta}_x}$ (possibly due to overfitting) may have an irregular projection and end up incurring a larger error in Stage 1. Thus we would like to understand the smoothness properties of functions residing in the projection of the Stage 2 DNN class $T[\mathcal{F}_x]$. In particular, an estimate such as the following is desirable.

**Assumption 4\*.** $T[\mathcal{F}_{\text{DNN}}] \subseteq C_T \cdot \mathbb{U}(B_{p',q'}^{s'}(\mathcal{Z}))$ *for each class of sigmoid or ReLU DNNs on $\mathcal{X}$, for some $0 < p', q' \leq \infty$, $s' > d_z(1/p' - 1/2)_+$ and $C_T > 0$ that do not depend on $L, W, S, M$.*

Although Assumption 4\* may hold if $T$ is sufficiently mollifying (see the example below), it is quite restrictive as the class $\mathcal{F}_{\text{DNN}}$ is much larger than the unit ball of any Besov space[3] and varies depending on design choice. However, motivated by the observation that the neural network $\psi_{\hat{\theta}_x}$ is approximating $f_{\text{str}} \in \mathbb{U}(B_{p,q}^s(\mathcal{X}))$, we can use a much less restrictive assumption:

**Assumption 4** (smoothness of $T$). $T[\mathbb{U}(B_{p,q}^s(\mathcal{X}))] \subseteq C_T \cdot \mathbb{U}(B_{p',q'}^{s'}(\mathcal{Z}))$ *for some $0 < p', q' \leq \infty$, $s' > d_z(1/p' - 1/2)_+$ and $C_T > 0$. If $s' \leq d_z/p'$, $\mathcal{P}_\mathcal{Z}$ also has Lebesgue density bounded above.*

The above formulation is a natural condition as it quantifies how much $T$ preserves smoothness of the target Besov space. In comparison, the link conditions essentially quantify how much $T$ contracts in an $L^2$-sense, giving us two distinct perspectives on the action of $T$. Assumption 4 also interacts with the link conditions to relate $s'$ back to $s$; see Lemma 3.2 for details.

We give some concrete examples in which Assumption 4 is satisfied. Suppose $T$ is smooth in the sense that the conditional density of $X$ given $Z$ exists and satisfies a Hölder condition w.r.t. $z$,

$$|f_{X|Z}(x|z_1) - f_{X|Z}(x|z_2)| \leq \eta(x)\|z_1 - z_2\|^\alpha \quad \text{for some} \quad \eta \in L^1(\mathcal{X}).$$

It follows that $|Tf(z_1) - Tf(z_2)| \leq \|\eta\|_1 \|f\|_\infty \|z_1 - z_2\|^\alpha$ for any bounded integrable function $f$. Since $C^\alpha(\mathcal{X})$ continuously embeds into the Zygmund space $B_{\infty,\infty}^\alpha(\mathcal{X})$ (Giné & Nickl, 2015, Proposition 4.3.23), Assumption 4 holds with $s' = \alpha$; in this scenario, Assumption 4\* also holds. At the opposite extreme, consider the case where the instrument is perfect, $d_x = d_z$ and $T$ restricts to a quasi-isometry between Besov spaces of the same order, the simplest example being $T = \text{id}_\mathcal{X}$. Then Assumption 4 holds with $s' = s$, however Assumption 4\* cannot be true. Thus Assumption 4 is a much more general condition encompassing both high and low degrees of smoothing.

**Controlling Stage 2 smoothness.** In order to replace Assumption 4\* with Assumption 4, we need to ensure that the Stage 2 predictor $\psi_{\hat{\theta}_x}$ lies in the $\mathbb{U}(B_{p,q}^s(\mathcal{X}))$, up to normalization. Using the

---

[2]This can be seen by taking the spectrum $\nu_k = k$ in Assumption 1 of Chen & Reiss (2011) and restricting to the span of the first $N$ basis elements or its complement in Assumptions 2, 5 with link function $\phi(t) = t^{\gamma_1/d_x}$.

[3]Even when using a smooth DNN class, the Besov norm of a generic DNN can scale polynomially in the specified weight norm bound, which we in turn take to scale polynomially in $m, n$ in our results.

smooth DNN class ensures $\psi_{\hat{\theta}_x} \in B_{p,q}^s(\mathcal{X})$ but does not by itself guarantee a norm bound (see footnote 3). A simple way to ensure this is to restrict the domain as follows:

$$\Theta_x' = \left\{ \theta_x \in \Theta_x \,\middle|\, |\psi_{\theta_x}|_{B_{p,q}^s(\mathcal{X})} \leq C_W \right\}. \tag{10}$$

In practice, since the restriction cannot be applied a priori, we can run the DFIV algorithm from random initialization and simply discard solutions which are ill-behaved: we show that the subset (10) still contains a rate-optimal solution. A more sophisticated method to guarantee smoothness which we also consider in our theoretical framework is to explicitly add the Besov (semi)norm as a penalty to be optimized for in Stage 2,

$$R(\theta_x) = |\psi_{\theta_x}|_{B_{p,q}^s(\mathcal{X})}^{\bar{q}} \quad \text{or} \quad R(\theta_x) = \|\psi_{\theta_x}\|_{B_{p,q}^s(\mathcal{X})}^{\bar{q}} \tag{11}$$

for any exponent $\bar{q} > 2$. The seminorm can be easily estimated from its equivalent discretization $|f|_{B_{p,q}^s(\mathcal{X})} \asymp (\sum_{k=0}^{\infty} [2^{ks} w_{r,p}(f, 2^{-k})]^q)^{1/q}$ (Giné & Nickl, 2015, p.330); the $L^p$ component can be ignored since $\psi_{\theta_x}$ is bounded. It is also bounded above by the Sobolev seminorm $|f|_{W_p^r(\mathcal{X})} = \sum_{|\alpha|=r} \|D^\alpha f\|_{L^p(\mathcal{X})}$ due to the string of continuous embeddings $W_p^r(\mathcal{X}) \hookrightarrow B_{p,\infty}^r(\mathcal{X}) \hookrightarrow B_{p,q}^s(\mathcal{X})$ (Giné & Nickl, 2015, Proposition 4.3.20). This can be numerically computed or backpropagated through $\theta_x$ to any desired accuracy independently of the data by differentiating the network output with respect to its input at mesh points; thus we assume $R$ can be computed exactly for simplicity.

Such smoothness-based or gradient-norm penalties have been considered before. For example, regularizers of the form $\mathbb{E}[\|D\psi_{\theta_x}\|_2^2], \mathbb{E}[\|D\psi_{\theta_x}\|_2^4]$ (equivalent to the Besov seminorm penalty when $p = q = 2, s = 1$) or similar have been successfully used to improve training or generalization of deep networks (Gulrajani et al., 2017; Sokolić et al., 2017; Arbel et al., 2018). Furthermore, Rosca et al. (2020) argue that smoothness regularization leads to benefits in inductive capabilities, robustness, and modeling performance.

## 3.2 PROJECTED UPPER AND LOWER BOUNDS

We now present the upper bound for the projected mean squared error (MSE) of DFIV. The proof is quite involved and is presented throughout Appendix D, with a brief sketch provided in Appendix B. We also require some new theory on rate-optimal estimation with sigmoid DNNs, which is developed in Appendix C.

**Theorem 3.1** (projected upper bound for DFIV). *Under Assumptions 1(i),2,3 and Assumption 4 with domain restriction (10) or regularization (11) with $\lambda$ asymptotic to the rate below, or Assumption 4\* without regularization, by choosing $\mathcal{F}_x = \mathcal{F}_{\text{DNN}}(\lceil \log_2 d_x \rceil + 1, \text{poly}(m), \text{poly}(m), \text{poly}(m))$ and similarly $\mathcal{F}_z$ to be smooth DNN classes, it holds that*

$$\mathbb{E}_{\mathcal{D}_1, \mathcal{D}_2} \left[ \|T\hat{f}_{\text{str}} - Tf_{\text{str}}\|_{L^2(\mathcal{P}_\mathcal{Z})}^2 \right] \lesssim m^{-\frac{2s+2\gamma_1}{2s+2\gamma_1+d_x}} \log m + (m \wedge n)^{-\frac{2s'}{2s'+d_z}} \log(m \wedge n). \tag{12}$$

See below for discussion. Rather surprisingly, the rate in Stage 2 samples $n$ depends only on the Stage 1 smoothness $s'$, while the rate in Stage 1 samples $m$ depends on the smoothness of both stages. We remark that $\mathcal{F}_z$ (and $\mathcal{F}_x$ if Assumption 4\* is used) can be replaced with ReLU DNNs since smoothness of the network output is not required. In this case, the network depth must scale logarithmically in $m, n$ and the log factors in (12) are replaced by $\log^3$.

**Projected minimax lower bound.** We now demonstrate the minimax optimality of the above learning rate. As usual, we state our assumptions first. The following is required to lower bound the error in $L^2(\mathcal{P}_\mathcal{X})$ by the ordinary $L^2$ error over the domain.

**Assumption 5.** *$\mathcal{P}_\mathcal{X}$ has Lebesgue density bounded below; more generally, the Radon-Nikodym derivative $\frac{\mathrm{d}m_\mathcal{X}}{\mathrm{d}\mathcal{P}_\mathcal{X}}$ w.r.t. Lebesgue measure $m_\mathcal{X}$ exists and is bounded above, so $\|\cdot\|_{L^2(\mathcal{P}_\mathcal{X})} \gtrsim \|\cdot\|_{L^2(\mathcal{X})}$.*

**Assumption 6** (reverse link condition). *There exists $\gamma_0 \geq \gamma_1$ such that for all $f \in P_r^{/k}$,*

$$2^{-\gamma_0 k} \|f\|_{L^2(\mathcal{P}_\mathcal{X})} \lesssim \|Tf\|_{L^2(\mathcal{P}_\mathcal{Z})}.$$

This is equivalent to assuming a polynomial rate for the *sieve measure of ill-posedness*, which depends on the operator $T$ and can be estimated from the data along with $\gamma_1$ with the method in (Blundell

et al., 2007). We are most interested in the case $\gamma_0 = \gamma_1$, which gives a precise characterization of the decay of $T$. This assumption also allows us to connect the exponent $s'$ appearing in (12) to the target smoothness $s$. Comparing the sizes of the standard and projected unit balls, we can show:

**Lemma 3.2.** *For Assumptions 4 and 6 to both be satisfied, it must hold that*

$$s'/d_z \leq (s + \gamma_0)/d_x. \tag{13}$$

The lemma is proved in Appendix F.2, where we also show that equality can hold if e.g. $d_x = d_z$ and $T$ maps B-splines to (scaled) B-splines. If equality is achieved in (13), we say that $T$ has *maximal smoothness*. In this case, the DFIV rate in Table 1 is retrieved.

Now the minimax lower bound for any estimator for the NPIV problem can be derived as follows. This is a nontrivial extension of the known lower bound in the Sobolev setting (Hall & Horowitz, 2005; Chen & Reiss, 2011), as B-splines are neither orthogonal nor independent.

**Proposition 3.3** (projected minimax lower bound). *Under Assumptions 1(ii),2,3,5,6, it holds that*

$$\inf_{\hat{f}_{\text{str}}:\text{NPIV}} \sup_{f_{\text{str}} \in \mathbb{U}(B^s_{p,q}(\mathcal{X}))} \mathbb{E}\left[ \|T\hat{f}_{\text{str}} - Tf_{\text{str}}\|^2_{L^2(\mathcal{P}_{\mathcal{Z}})} \right] \gtrsim n^{-\frac{2s+2\gamma_0}{2s+2\gamma_1+d_x}}, \tag{14}$$

*where $\hat{f}_{\text{str}}$ : NPIV is taken over all measurable functions of the data $(\mathcal{D}_1, \mathcal{D}_2)$.*

The proof is given in Appendix F.1 and relies on reduction to the NPIR model to which the Yang-Barron method can be applied; in fact, the lower bound holds for all valid estimators of $Tf_{\text{str}}$. The bound is independent of $m$, hence is most informative in the limit $m \to \infty$. The usual nonparametric rate is retrieved when $\gamma_0 = \gamma_1 = 0$. Comparing with the upper bound (12), we conclude:

**Corollary 3.4** (projected optimality of DFIV). *If $\gamma_0 = \gamma_1$ and $T$ has maximal smoothness, DFIV attains the nearly minimax optimal rate $n^{-\frac{2s+2\gamma_0}{2s+2\gamma_0+d_x}} \log n$ in the projected metric if $m = \Omega(n)$.*

We also observe two potential factors of suboptimality. If the inequality (13) is strict, DFIV is upper bounded by the rate $n^{-\frac{2s'}{2s'+d_z}} > n^{-\frac{2s+2\gamma_0}{2s+2\gamma_0+d_x}}$ which is suboptimal due to the increased difficulty (reduced regularity) of the Stage 1 regression. If $\gamma_0 > \gamma_1$, the link conditions are loose, so the lower bound also becomes comparatively weaker. In the latter case, it may also be beneficial to scale $m$ as $\Omega\left(n^{1 \vee \frac{2s+2\gamma_1+d_x}{2s+2\gamma_1} \frac{2s'}{2s'+d_z}}\right)$ rather than $\Omega(n)$, since the rate in $m$ can become slower than $n$ in (12).

## 3.3 Full Upper and Lower Bounds

Corollary 3.4 establishes minimax optimality of the projected MSE, however since $T^{-1}$ is often unbounded in NPIV, we cannot directly deduce the full MSE from the projected rate. Therefore, a more interesting result is whether DFIV can achieve minimax optimality in the non-projected rate. To this end, we prove the following non-projected upper bound in Appendix E.2.

**Theorem 3.5** (full upper bound for DFIV). *Let $p \geq 2$. Under Assumptions 1(i),2,3,6 and Assumption 4 with domain restriction (10) or regularization (11), by choosing $\lambda$ as in Theorem 3.1, $\mathcal{F}_x = \mathcal{F}_{\text{DNN}}(\lceil \log_2 d_x \rceil + 1, \text{poly}(m), \text{poly}(m), \text{poly}(m))$ and $\mathcal{F}_z$ to be smooth DNN classes, it holds that*

$$\mathbb{E}_{\mathcal{D}_1,\mathcal{D}_2}\left[ \|\hat{f}_{\text{str}} - f_{\text{str}}\|^2_{L^2(\mathcal{P}_{\mathcal{X}})} \right] \lesssim m^{-\frac{2(s-\gamma_0+\gamma_1)}{2s+2\gamma_1+d_x}} \log m + (m \wedge n)^{-\frac{2s'}{2s'+d_z} \frac{s-\gamma_0+\gamma_1}{s+\gamma_1}} \log(m \wedge n). \tag{15}$$

*The log factors can be improved to $\log^{\frac{s-\gamma_0+\gamma_1}{s+\gamma_1}}$.*

**Stage 2 smoothness revisited.** The control over the smoothness of $\hat{f}_{\text{str}}$ now plays a dual role: besides bounding the difficulty of Stage 1 through $T\hat{f}_{\text{str}}$, it also determines the strength of the reverse link condition that determines the final non-projected MSE, since Assumption 6 is stated in terms of the B-spline basis. To be precise, the $L^2$ risk must be bounded by taking $f_N, \hat{f}_N$ to be the width $N$ approximations of $f_{\text{str}}, \hat{f}_{\text{str}}$, respectively, and evaluating the truncation

$$\|\hat{f}_{\text{str}} - f_{\text{str}}\|_{L^2(\mathcal{P}_{\mathcal{X}})} \lesssim \|\hat{f}_{\text{str}} - \hat{f}_N\|_{L^2(\mathcal{P}_{\mathcal{X}})} + 2^{\gamma_0 k}\|T\hat{f}_N - Tf_N\|_{L^2(\mathcal{P}_{\mathcal{Z}})} + \|f_{\text{str}} - f_N\|_{L^2(\mathcal{P}_{\mathcal{X}})}. \tag{16}$$

Then the truncation error $\|\hat{f}_{\text{str}} - \hat{f}_N\|_{L^2(\mathcal{P}_{\mathcal{X}})}$ directly depends on the regularity of $\hat{f}_{\text{str}}$, and smoothness-based restriction or regularization becomes unavoidable to obtain theoretical guarantees.

**Remark 3.6.** We point out that Theorem 3.5 only studies the situation when $p \geq 2$. The regime $p < 2$ – where separation with fixed-feature estimators is achieved – requires special treatment, and is proved under an extended link condition in Section 4; see the discussion therein.

We now provide the uniform NPIV lower bound, proved in Appendix F.1.

**Proposition 3.7** (full minimax lower bound). *Under Assumptions 1(ii),2,3, it holds that*

$$\inf_{\hat{f}_{\mathrm{str}}:\mathrm{NPIV}} \sup_{f_{\mathrm{str}} \in \mathbb{U}(B_{p,q}^s(\mathcal{X}))} \mathbb{E}\left[\|\hat{f}_{\mathrm{str}} - f_{\mathrm{str}}\|_{L^2(\mathcal{X})}^2\right] \gtrsim n^{-\frac{2s}{2s+2\gamma_1+d_x}}.$$

From this, we conclude the following result, which to our knowledge is the first result along with Corollary 3.4 to establish minimax optimal rates for NPIV with deep neural networks.

**Corollary 3.8** (optimality of DFIV). *If $p \geq 2$, $\gamma_0 = \gamma_1$ and $T$ has maximal smoothness, DFIV attains the nearly minimax optimal rate $n^{-\frac{2s}{2s+2\gamma_1+d_x}}(\log n)^{\frac{s}{s+\gamma_1}}$ if $m = \Omega(n)$.*

More generally, the upper bound always scales as $n^{-\frac{2s'}{2s'+d_z}\frac{s-\gamma_0+\gamma_1}{s+\gamma_1}}$ for large enough $m$. If $T$ does not have maximal smoothness, the first exponent will deteriorate due to increased difficulty of Stage 1; if the link conditions are loose ($\gamma_0 > \gamma_1$), the second exponent will deteriorate.

**Optimal splitting.** We find that the sample requirement for Stage 1 of DFIV in Corollaries 3.4, 3.8 is only $m = \Omega(n)$. In contrast, for kernel IV, Singh et al. (2019) and Meunier et al. (2024) require $m > \Omega(n)$ to obtain minimax optimal rates, prescribing an asymmetric splitting if the Stage 1, 2 samples are split from a single dataset. However, this leads to a suboptimal scaling in the total number of samples $m + n$. Our results show that DFIV requires strictly less Stage 1 data, retrieving the same optimal rate in the total number of samples.

## 4 SEPARATION WITH FIXED-FEATURE IV

It has been empirically observed that DFIV outperforms existing methods especially when the structural function is non-smooth or discontinuous; see the experiments on the conditional average treatment effect and behavior reinforcement learning tasks in Xu et al. (2021). In this section, we study the efficiency of DFIV in the setting where $f_{\mathrm{str}}$ lives in a Besov space with $p < 2$. In this regime, we rigorously establish a separation between DFIV and *linear* estimators including sieve or kernel-based algorithms, proving the superiority of deep neural features over non-adaptive methods.

For ordinary regression in a $d$-dimensional Besov space, the minimax rate over all linear estimators is known to be lower bounded by $n^{-\frac{2(s-\Delta)}{2(s-\Delta)+d}}$ where $\Delta = d(1/p - 1/2)_+$ determines the degree of separation from the optimal rate (Donoho & Johnstone, 1998; Zhang et al., 2002). This occurs because functions in the regime $p < 2$ are highly spatially inhomogeneous, consisting of both jagged (possibly discontinuous when $s < d/p$) and smooth regions. Adaptive models such as DNNs have an inherent advantage over linear estimators by allocating more complexity (i.e. choosing higher-frequency basis elements as needed) in regions of low smoothness to capture spatial variability efficiently (Donoho & Johnstone, 1998; Suzuki, 2019).

A *linear IV estimator* is formally defined as any estimator of $f_{\mathrm{str}}$ of the following form,

$$\hat{f}_L(x) = \sum_{i=1}^n u_i(x, (\tilde{z}_i)_{i=1}^n, \mathcal{D}_1)\tilde{y}_i, \quad u_1, \cdots, u_n : \mathcal{X} \times \mathcal{Z}^n \times (\mathcal{X} \times \mathcal{Z})^m \to \mathbb{R}.$$

We only require linearity w.r.t. the Stage 2 responses, as the lower bound will again hold for any reduction to the NPIR model (4). fixed-feature methods such as 2SLS, nonparametric 2SLS (Newey & Powell, 2003), Nadaraya-Watson methods (Carrasco et al., 2007; Darolles et al., 2011), sieve IV (Newey & Powell, 2003; Chen & Christensen, 2018), kernel IV (Singh et al., 2019; Meunier et al., 2024), and moment-based methods (Zhang et al., 2023) are all examples of linear IV estimators.

We first show that all linear IV estimators incur a degree of suboptimality when the target function possesses low homogeneity. The following lower bound is proved in Appendix F.3 by adapting the approach in Zhang et al. (2002) for univariate regression to IV regression.

**Theorem 4.1** (lower bound for linear IV estimators). *Under Assumptions 1(ii),2,3 and assuming $\mathcal{P}_{\mathcal{X}}$ has Lebesgue density bounded above, for $\Delta = d_x(1/p - 1/2)_+$, it holds that*

$$\inf_{\hat{f}_L:\text{linear}} \sup_{f_{\text{str}} \in \mathbb{U}(B_{p,q}^s(\mathcal{X}))} \mathbb{E}\left[\|\hat{f}_L - f_{\text{str}}\|_{L^2(\mathcal{X})}^2\right] \gtrsim n^{-\frac{2(s-\Delta)}{2(s-\Delta)+2\gamma_1+d_x}}.$$

This is not directly comparable with DFIV, however, as we assumed $p \geq 2$ in Theorem 3.5 so as to not incur any looseness when evaluating (16) with the reverse link condition. More precisely, $f_{\text{str}}, \hat{f}_{\text{str}}$ had to be approximated using B-splines up to a fixed resolution cutoff, which lower bounds the truncation error by the best $N$-term linear approximation error or *Kolmogorov width*, which is suboptimal when $p < 2$ (Vybíral, 2008). Nonetheless, DNNs are capable of choosing from a much wider pool of basis elements to achieve the optimal *nonlinear* approximation rate $N^{-s/d_x}$ for functions of low spatial homogeneity (Suzuki, 2019). To account for such adaptive selection, we now extend Assumption 6 to varying linear subsets of a higher resolution horizon.

**Assumption 7** (extended reverse link condition). *Fix $C^* > \frac{\Delta d_x}{s-\Delta} + 1$ and let $S$ be any subset of size $O(2^{kd_x})$ of the set of B-splines up to resolution $\lceil C^* k \rceil$. Then $2^{-\gamma_0 k}\|f\|_{L^2(\mathcal{P}_{\mathcal{X}})} \lesssim \|Tf\|_{L^2(\mathcal{P}_{\mathcal{Z}})}$ holds for all $f \in \text{span } S$.*

The intuition is that $T$ is more contractive for functions with higher complexity (i.e. requiring many basis elements to construct), and hence the tighter link constant $2^{-\gamma_0 k}$ holds for any subset with size similar to $P_r^{/k}$, instead of $2^{-\gamma_0 C^* k}$ guaranteed for the whole space $P_r^{/C^* k}$. Such a dimensional separation is in fact easily satisfied: for example, suppose $T$ is the projection in $\mathbb{R}^D$ to the orthogonal complement of $\text{span}\{v\}$ for a unit vector $v$ such that $|v_1|, \cdots, |v_D| \geq \frac{c}{\sqrt{D}}$ for some $c \in (0, 1]$. Then for any size $D'$ subset $A \subset \{1, \cdots, D\}$ and unit vector $w \in \text{span}\{e_j \mid j \in A\}$,

$$\|Tw\|_2 = 1 - (\textstyle\sum_{j \in A} w_j v_j)^2 \geq 1 - \textstyle\sum_{j \in A} v_j^2 = \textstyle\sum_{j \notin A} v_j^2 \geq c(1 - D'/D).$$

Hence as long as $D' = o(D)$ (indeed $D' = D^{1/C^*}$ in Assumption 7), it is possible that $T$ is 'minimally contractive' on all subspaces $P$ spanned by $D'$ basis elements ($\inf_{w \in P, w \neq 0}\|Tw\|_2/\|w\|_2 \approx 1$) even as $D, D' \to \infty$, but 'maximally contractive' on the entire space ($\inf_{w \neq 0}\|Tw\|_2/\|w\|_2 = 0$).

With this modification, we conclude the following strict separation in Appendix E.2. A notable consequence is that the assumption of maximal smoothness of $T$ can be weakened to a range of $s'$ depending on the gap $\Delta$.

**Corollary 4.2.** *Under the setting of Theorem 3.5 and also Assumption 7, the upper bound (15) and Corollary 3.8 hold for $0 < p < 2$.*

Hence comparing with Theorem 4.1, DFIV is faster than any linear IV estimator if $\gamma_0 = \gamma_1$ and

$$\frac{(s-\Delta)d_x}{(s-\Delta)d_x + \Delta(2\gamma_0 + d_x)} \frac{s+\gamma_0}{d_x} < \frac{s'}{d_z} \leq \frac{s+\gamma_0}{d_x} \quad \text{cf. (13)}.$$

**Remark 4.3.** We also show a separation result for the projected rates in Appendix F.4. In this case, Assumption 7 is not needed since Theorem 3.1 is valid for all $p > 0$, being independent of the reverse link condition. On the other hand, the lower bound requires an additional assumption that $\mathcal{Z}$ is sufficiently 'spatially covered' by the projection operator so as to be difficult for spatially non-adaptive estimators. We then prove the projected minimax error for any linear IV estimator is at least $\widetilde{\Omega}\left(n^{-\frac{2(s-\Delta)+2\gamma_0}{(2(s-\Delta)+2\gamma_1+d_z)\vee(2\gamma_1+d_x)}}\right)$ (Theorem F.2), which is worse than the DFIV upper bound (12) if e.g. $\gamma_0 = \gamma_1$, $T$ has maximal smoothness and $d_z > (d_x - 2(s-\Delta)) \vee \frac{s-\Delta+\gamma_0}{s+\gamma_0} d_x$.

## 5 CONCLUSION

In this paper, we developed a novel estimation error analysis for two-stage IV regression with neural features. We proved that the DFIV algorithm achieves the minimax optimal rate when the structural function lies in a Besov space and further demonstrated that DFIV can outperform fixed-feature IV estimators by adaptively learning functions with low spatial homogeneity. Furthermore, we showed that a balanced number of Stage 1 and 2 samples suffices to attain optimal performance. These results provide a rigorous foundation for the advantages of DFIV in terms of both adaptivity and sample efficiency, paving the way for further exploration of neural features for causal inference.

## ACKNOWLEDGMENTS

Juno Kim was partially supported by JST CREST (JPMJCR2015). Taiji Suzuki was partially supported by JSPS KAKENHI (24K02905) and JST CREST (JPMJCR2115). Dimitri Meunier, Arthur Gretton and Zhu Li were supported by the Gatsby Charitable Foundation.

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

TABLE OF CONTENTS

## TABLE OF SYMBOLS

Table 2: List of symbols, given roughly in order of appearance in the main text. Some important symbols from the appendix are also included, while auxiliary symbols are omitted.

| Symbol | Description |
|--------|-------------|
| $X, Y, Z$ | endogeneous variable, outcome, instrumental variable |
| $\xi, \eta$ | confounder, NPIR model error |
| $f_{\text{str}}$ | structure function |
| $\hat{f}_{\text{str}}$ | DFIV estimator |
| $C$ | upper bound for $f_{\text{str}}$ |
| $d_x, d_z$ | dimension of $X \in \mathcal{X}, Z \in \mathcal{Z}$ |
| $\mathcal{P}_{\mathcal{X}}, \mathcal{P}_{\mathcal{Z}}$ | marginal distribution of $X, Z$ |
| $T$ | NPIV projection operator |
| $\sigma_0, \sigma_1$ | regularity constants of $\eta$ |
| $B_{p,q}^s(\mathcal{X})$ | Besov space on $\mathcal{X}$ with smoothness parameters $s, p, q$ |
| $C^s(\mathcal{X})$ | Hölder space of smoothness $s$ |
| $W_p^r(\mathcal{X})$ | Sobolev space of order $r$ in $L^p$-norm |
| $\mathbb{U}(\cdot)$ | unit ball of (quasi-)Banach space |
| $\iota_r$ | cardinal B-spline of order $r$ |
| $\omega_{k,\ell}$ | B-spline basis element with resolution $k$, location $\ell$ |
| $u, V$ | parameters of 2SLS algorithm |
| $\phi_{\theta_z}, \psi_{\theta_x}$ | Stage 1,2 DNNs with parameters in $\Theta_z, \Theta_x$ |
| $\mathcal{F}_z, \mathcal{F}_x$ | Stage 1,2 DNN function classes |
| $\mathcal{D}_1, \mathcal{D}_2$ | Stage 1,2 datasets |
| $m, n$ | number of Stage 1, 2 samples $(x_i, z_i), (\tilde{y}_i, \tilde{z}_i)$ |
| $\hat{\mathbb{E}}_{X|Z}[\cdot]$ | Stage 1 estimator |
| $R$ | Stage 2 regularizer with regularization strength $\lambda$ |
| $\sigma$ | smooth activation function |
| $L, W, S, M$ | parameters of DNN class |
| $\underline{C}, \bar{C}$ | thresholds of clipping operation |
| $\gamma_0, \gamma_1$ | exponents in link and reverse link condition |
| $P_r^{/k}$ | linear span of resolution $k$ B-splines |
| $\Pi_r^{/k}$ | projection to $P_r^{/k}$ |
| $\Delta$ | degree of separation |
| $\hat{f}_L$ | linear IV estimator |
| | |
| $\mathcal{N}_z, \mathcal{N}_x$ | $\delta_z, \delta_x$-covering numbers of $\mathcal{F}_z, \mathcal{F}_x$ in $L^\infty$-norm |
| $\theta_{z,k}, \theta_{x,j}$ | covers corresponding to $\mathcal{N}_z, \mathcal{N}_x$ |
| $\hat{\mathcal{C}}$ | dynamic extended cover |
| $\mathcal{C}^*$ | population extended cover |
| $\check{\omega}$ | smooth DNN approximation of $\omega_{0,0}$ |
| $\beta_{k,\ell}$ | coefficient sequence of B-spline expansion |
| $\|\cdot\|_{b_{p,q}^s}$ | sequence quasi-norm with smoothness parameters $s, p, q$ |
| $f_N$ | $N$-basis truncation of $f_{\text{str}}$ |
| $\check{f}_N$ | smooth DNN approximation of $f_N$ |
| $\Pi$ | Stage 1 population error function |
| $\hat{\Pi}$ | Stage 1 empirical error function |
| $\mathcal{R}_L$ | linear minimax optimal rate |

# APPENDIX

## A   COMPARISON WITH EXISTING WORKS

**Error analysis of NPIV estimators.**   Many existing analyses of NPIV study linear (kernel or sieve) methods, where the estimator is constructed by regressing against a known basis expansion in an RKHS or Sobolev ball; some are further shown to attain the minimax optimal rate (Ai & Chen, 2003; Newey & Powell, 2003; Blundell et al., 2007; Chen & Reiss, 2011; Horowitz, 2011; Chen & Christensen, 2018; Singh et al., 2019; Meunier et al., 2024). It is also possible to relax the restrictions on smoothness via e.g. spectral or Tikhonov regularization (Hall & Horowitz, 2005; Carrasco et al., 2007; Darolles et al., 2011). However, these methods do not readily extend to DNN classes, which are highly nonlinear in their parameters and misspecified for both stages.

Theoretical guarantees for adaptive neural network-based IV estimators are at present very limited. Xu et al. (2021) give an initial analysis of DFIV with no specific learning rate, under strong assumptions on $T$ and identifiability. Hartford et al. (2017); Li et al. (2024b) study a two-stage NPIV estimator with neural networks, however there are two key differences. First, from the algorithmic side, the two works both employ a DNN to learn the conditional density $f_{X|Z}(\cdot)$ in Stage 1. However, our Stage 1 only needs to estimate the conditional mean of the relevant features for Stage 2, which can be much easier depending on the Stage 2 network. Second, Li et al. (2024b) only derive an upper bound in terms of abstract function class complexity measures. It is not clear whether the derived rate is minimax optimal since neither a concrete evaluation for DNNs nor a matching lower bound is provided.

**Estimation ability of DNNs.**   For ordinary least-squares regression, it is known that DNNs achieve the minimax optimal learning rate for various target classes and furthermore outperform fixed-feature estimators when the target function possesses low homogeneity (Suzuki, 2019) or directional smoothness (Suzuki & Nitanda, 2021). These works build on classical separation results between adaptive and fixed-feature estimators (Donoho & Johnstone, 1998; Zhang et al., 2002; Dũng, 2011). Our results in Section 4 extend this line of work to two-stage regression. A significant challenge of 2SLS with DNN is how can we control the smoothness of stage 2 DNN, as it is the target of stage 1 regression. This is achieved by using DNNs with smooth (e.g. sigmoid) activations, for which we extend existing approximation results (Bauer & Kohler, 2019; Langer, 2021; De Ryck et al., 2021) to obtain a new approximation guarantee for Besov functions which is both rate optimal in $L^2$ norm (see Suzuki, 2019) and also converge in Besov norm.

### A.1   DEFINITION OF BESOV SPACE

**Definition A.1.** Let $s > 0$ and $0 < p, q \leq \infty$. For $f \in L^p(\mathcal{X})$ and $r \in \mathbb{N}$, $r > s \vee s + 1 - 1/p$, the $r$th modulus of smoothness of $f$ is given as

$$w_{r,p}(f,t) := \sup_{\|h\|_2 \leq t} \|\Delta_h^r(f)\|_p, \quad \Delta_h^r(f)(x) = 1_{\{x, x+rh \in \mathcal{X}\}} \sum_{j=0}^r \binom{r}{j}(-1)^{r-j} f(x + jh).$$

Then the Besov space with parameters $s, p, q$ is defined as the following subspace of $L^p(\mathcal{X})$,

$$B_{p,q}^s(\mathcal{X}) = \{f \in L^p(\mathcal{X}) \mid \|f\|_{B_{p,q}^s(\mathcal{X})} := \|f\|_{L^p(\mathcal{X})} + |f|_{B_{p,q}^s(\mathcal{X})} < \infty\},$$

where the Besov seminorm is defined as $|f|_{B_{p,q}^s(\mathcal{X})} := (\int_0^\infty t^{-qs-1} w_{r,p}(f,t)^q \, \mathrm{d}t)^{1/q}$ if $q < \infty$ and $\sup_{t>0} t^{-s} w_{r,p}(f,t)$ if $q = \infty$.

The unit ball of $(B_{p,q}^s(\mathcal{X}), \|\cdot\|_{B_{p,q}^s})$ is denoted as $\mathbb{U}(B_{p,q}^s(\mathcal{X}))$. Moreover, Besov spaces on $\mathcal{Z}$ are defined in the same manner. We have the following classical results (Triebel, 1983):

- For $s > 0$, $s \notin \mathbb{N}$, the Hölder space $C^s(\mathcal{X}) = B_{\infty,\infty}^s(\mathcal{X})$. For $m \in \mathbb{N}$, $C^m(\mathcal{X}) \hookrightarrow B_{\infty,\infty}^m(\mathcal{X})$.
- For $m \in \mathbb{N}$, the Sobolev space $W_2^m(\mathcal{X}) = B_{2,2}^m(\mathcal{X})$ and $B_{p,1}^m(\mathcal{X}) \hookrightarrow W_p^m(\mathcal{X}) \hookrightarrow B_{p,\infty}^m(\mathcal{X})$.
- If $s > d_x/p$ then $B_{p,q}^s(\mathcal{X}) \hookrightarrow C^0(\mathcal{X})$.
- If $s > d_x(1/p - 1/r)_+$ then $B_{p,q}^s(\mathcal{X}) \hookrightarrow L^r(\mathcal{X})$.
- $B_{p,q}^s(\mathcal{X})$ is a Banach space unless $p$ or $q < 1$, when it becomes a quasi-Banach space.

## B  PROOF SKETCH OF DFIV UPPER BOUND

In this section, we give a brief outline of our proof of the upper bound for DFIV in Theorem 3.1 which is one of our key contributions. The full proof is presented throughout Appendices C, D and the non-projected MSE upper bounds (Theorem 3.5 and Corollary 4.2) are derived using this result in Appendix E.

First note that in the population limit of (7), Stage 1 is essentially minimizing the loss

$$\mathbb{E}_{X,Z}\left[\|\psi_{\theta_x}(X) - \phi_{\theta_z}(Z)\|^2\right] = \mathbb{E}_{X,Z}\left[\|\psi_{\theta_x}(X) - T\psi_{\theta_x}(Z)\|^2\right] + \|T\psi_{\theta_x} - \phi_{\theta_z}\|^2_{L^2(\mathcal{P}_{\mathcal{Z}})},$$

and so factoring out the projection error, we may view $\|T\psi_{\theta_x} - \hat{\mathbb{E}}_{X|Z}[\psi_{\theta_x}]\|_{L^2(\mathcal{P}_{\mathcal{Z}})}$ as the 'effective' Stage 1 estimation error.

Denote the $\delta_x, \delta_z$-covering numbers of the hypothesis classes $\mathcal{F}_x, \mathcal{F}_z$ as $\mathcal{N}_x = \mathcal{N}(\mathcal{F}_x, \|\cdot\|_{L^\infty(\mathcal{X})}, \delta_x)$ and $\mathcal{N}_z = \mathcal{N}(\mathcal{F}_z, \|\cdot\|_{L^\infty(\mathcal{Z})}, \delta_z)$, respectively. We begin by showing the following oracle inequality from the definition of $\hat{\theta}_x$ as the empirical risk minimizer of (8) conditioned on $\mathcal{D}_1$,

$$\mathbb{E}_{\mathcal{D}_2}\left[\|Tf_{\mathrm{str}} - \hat{\mathbb{E}}_{X|Z}[\psi_{\hat{\theta}_x}]\|^2_{L^2(\mathcal{P}_{\mathcal{Z}})}\right] \lesssim \inf_{\theta_x \in \Theta_x} \|Tf_{\mathrm{str}} - \hat{\mathbb{E}}_{X|Z}[\psi_{\theta_x}]\|^2_{L^2(\mathcal{P}_{\mathcal{Z}})} + \frac{\log \mathcal{N}_z}{n} + \delta_z,$$

where the second term can be further bounded by

$$2 \inf_{\theta_x \in \Theta_x} \|Tf_{\mathrm{str}} - T\psi_{\theta_x}\|^2_{L^2(\mathcal{P}_{\mathcal{Z}})} + 2 \sup_{\theta_x \in \Theta_x} \|T\psi_{\theta_x} - \hat{\mathbb{E}}_{X|Z}[\psi_{\theta_x}]\|^2_{L^2(\mathcal{P}_{\mathcal{Z}})}.$$

With some manipulation, the projected error can be bounded as

$$\|Tf_{\mathrm{str}} - T\hat{f}_{\mathrm{str}}\|^2_{L^2(\mathcal{P}_{\mathcal{Z}})} \leq \|Tf_{\mathrm{str}} - \hat{\mathbb{E}}_{X|Z}[\psi_{\hat{\theta}_x}]\|^2_{L^2(\mathcal{P}_{\mathcal{Z}})} + 2\|\hat{\mathbb{E}}_{X|Z}[\psi_{\hat{\theta}_x}] - T\hat{f}_{\mathrm{str}}\|^2_{L^2(\mathcal{P}_{\mathcal{Z}})}$$

$$\lesssim \inf_{\theta_x \in \Theta_x} \|Tf_{\mathrm{str}} - T\psi_{\theta_x}\|^2_{L^2(\mathcal{P}_{\mathcal{Z}})} + \sup_{\theta_x \in \Theta_x} \|T\psi_{\theta_x} - \hat{\mathbb{E}}_{X|Z}[\psi_{\theta_x}]\|^2_{L^2(\mathcal{P}_{\mathcal{Z}})} + \frac{\log \mathcal{N}_z}{n} + \delta_z.$$

Thus it becomes necessary to bound the expected *supremum* of the effective Stage 1 estimation error, $\sup\|T\psi_{\theta_x} - \hat{\mathbb{E}}_{X|Z}[\psi_{\theta_x}]\|_{L^2(\mathcal{P}_{\mathcal{Z}})}$ over the hypothesis space, which is highly nontrivial. We aim to achieve this by controlling both the supremum of the corresponding empirical process

$$\sup_{\theta_x \in \Theta_x} \frac{1}{m} \sum_{i=1}^{m} \left(T\psi_{\theta_x}(z_i) - \hat{\mathbb{E}}_{X|Z}[\psi_{\theta_x}](z_i)\right)^2,$$

and the supremum of their difference. Naively these quantities are evaluated by reducing to a finite cover $(\psi_{x,j})_{j \leq \mathcal{N}_x}$ of $\mathcal{F}_x$. This does not work for DFIV, however, since the Stage 1 estimator $\hat{\mathbb{E}}_{X|Z}[\cdot]$ can be ill-behaved in general; approximating $\psi_{\theta_x}$ by $\psi_{\theta_{x,j}}$ does not guarantee that $\hat{\mathbb{E}}_{X|Z}[\psi_{\theta_{x,j}}]$ is a good approximation of $\hat{\mathbb{E}}_{X|Z}[\psi_{\theta_x}]$. Instead, we construct an extended *dynamic* or data-dependent cover $\hat{\mathcal{C}}$, which maps to a subset of the product cover for $\mathcal{F}_x \times \mathcal{F}_z$, that approximates all possible combinations of $\psi_{\theta_x}$ and $\hat{\mathbb{E}}_{X|Z}[\psi_{\theta_x}]$. A similar construction $\mathcal{C}^*$ is given for the population version of Stage 1 risk minimization. By converting to the supremum over $\hat{\mathcal{C}}, \mathcal{C}^*$ and applying classical concentration bounds, we obtain the following general bound:

$$\mathbb{E}_{\mathcal{D}_1,\mathcal{D}_2}\left[\|Tf_{\mathrm{str}} - T\hat{f}_{\mathrm{str}}\|^2_{L^2(\mathcal{P}_{\mathcal{Z}})}\right] \lesssim \inf_{\theta_x \in \Theta_x} \|Tf_{\mathrm{str}} - T\psi_{\theta_x}\|^2_{L^2(\mathcal{P}_{\mathcal{Z}})} + \frac{\log \mathcal{N}_x}{m} + \delta_x$$

$$+ \sup_{\theta_x \in \Theta_x} \left[\inf_{\theta_z \in \Theta_z} \|T\psi_{\theta_x} - \phi_{\theta_z}\|^2_{L^2(\mathcal{P}_{\mathcal{Z}})}\right] + \frac{\log \mathcal{N}_z}{m \wedge n} + \delta_z,$$

in terms of the Stage 2 and Stage 1 (supremal) approximation errors and covering entropies. In particular, the Stage 1 approximation error for each $\theta_x \in \Theta_x$ is determined by the smoothness of $T\psi_{\theta_x}$, which can be bounded by Assumption 4* or Assumption 4 combined with domain restriction (10). If instead regularization is used, we can repeat the argument including the regularization term to also obtain a smoothness bound.

It remains to choose the appropriate DNN classes and explicitly evaluate the terms above. The approximation rates for DNNs equal the optimal nonlinear rate; this is known for ReLU DNNs (Suzuki, 2019), but we prove this result for sigmoid DNNs together with a stronger Besov norm convergence guarantee in Theorem C.8, which allows us to make the above smoothness argument rigorous. Taking the DNN class variables to balance the approximation and entropy terms, we conclude the projected upper bound (12). □

## C  DEEP SIGMOID NEURAL NETWORKS

In this section we prove key results on the approximation of Besov functions by deep sigmoid neural networks. The results are obtained by carefully applying the Sobolev norm bounds developed in De Ryck et al. (2021) to the B-spline system. The main difference is that instead of relying on local polynomial approximation, we exploit the fact that B-splines are piecewise polynomials to give a sparse construction similar to Suzuki (2019). This allows us to achieve rate optimality in the sense of best approximation width (Theorem C.8) as well as small covering number (Lemma C.9). Moreover, the proof will serve to demonstrate how to extend the construction to any $C^r$ activation with a decaying part (e.g. either of $\lim_{x \to \pm\infty} \sigma(x)$ exists) by manipulating Taylor expansions to approximate polynomials as in Mhaskar & Micchelli (1992); De Ryck et al. (2021); Langer (2021), or by constructing piecewise polynomials exactly in the case of ReQU.

### C.1  PRELIMINARIES

We first present some preliminary results. Denote the dimension of the domain as $d = d_x$ for brevity. It is easy to see that the cardinal B-spline $\iota_r$ is a piecewise polynomial of degree $r$ supported on $[0, r+1]$ and $\iota_r \in W^r_\infty(\mathbb{R})$, $\mathrm{im}\,\iota_r \subseteq [0, 1]$. We make use of the following expansion (Mhaskar & Micchelli, 1992):

$$\iota_r(x) = \sum_{j=0}^{r+1} a_j (x-j)^r_+ = \sum_{k=0}^{r+1} 1_{[k,k+1)}(x) \sum_{j=0}^{k} a_j (x-j)^r, \quad \text{where} \quad a_j = \frac{(-1)^j}{r!} \binom{r+1}{j}. \tag{17}$$

Note that the indicator $1_{[0,1)}(x)$ can be well approximated by the shallow sigmoid network

$$\delta_B(x) := \sigma(Bx) - \sigma(B(x-1))$$

for some $B > 1$, whose translates form a smooth partition of unity over $\mathbb{R}$.

**Lemma C.1** (smooth decay of $\delta_B$). *It holds that:*

(1) $\|\delta_B(x+1) + \delta_B(x) + \delta_B(x-1) - 1\|_{W^r_\infty([0,1])} \lesssim e^{-B}$.

(2) *For all $0 \le k, \ell \le r+1$ with $|k-\ell| \ge 2$ we have $\|\delta_B(x-k)\|_{W^r_\infty([\ell,\ell+1))} \lesssim B^r e^{-B}$.*

*Here, constants depending only on $r$ are hidden.*

*Proof.* (1) The $r$th order derivative of $\sigma$ satisfies $|\sigma^{(r)}(x)| \le r^{r+1}(e^x \wedge e^{-x})$ (De Ryck et al., 2021, Lemma A.4), so that

$$\|\delta_B(x+1) + \delta_B(x) + \delta_B(x-1) - 1\|_{W^r_\infty([0,1])}$$
$$= \|\sigma(B(x+1)) - \sigma(B(x-2)) - 1\|_{W^r_\infty([0,1])}$$
$$\le \|\sigma(-B(x+1))\|_{W^r_\infty([0,1])} + \|\sigma(B(x-2))\|_{W^r_\infty([0,1])}$$
$$\le \frac{2}{1+e^B} + 2r^{r+1}e^{-B} \lesssim e^{-B}.$$

(2) For any pair $k, \ell$ with $k - \ell \ge 2$ it holds that

$$\|\delta_B(x-k)\|_{W^r_\infty([\ell,\ell+1))} \le \|\sigma(Bx) - \sigma(B(x-1))\|_{W^r_\infty([\ell-k,\ell-k+1))}$$
$$\le \sigma(B(\ell-k+1)) + B^r\|\sigma^{(r)}(Bx) - \sigma^{(r)}(B(x-1))\|_{L^\infty([\ell-k,\ell-k+1))}$$
$$\le \frac{1}{1+e^B} + 2r^{r+1}B^r e^{-B} \lesssim B^r e^{-B}.$$

The same bound holds when $k - \ell \le -2$ by symmetry. $\qquad\square$

We also require a finer control over the boundary decay of $\delta_B$:

**Lemma C.2.** *It holds for sufficiently large $B$ that $\|\delta_B(x+1)x^r\|_{W^r_p([0,1])} \lesssim B^{-1/2}$.*

*Proof.* We first note for all $j = 1, \cdots, r$ that

$$\|x^j e^{-Bx}\|_{L^\infty([0,1])} \le \|x^j\|_{L^\infty([0,1/\sqrt{B}])} \vee \|e^{-Bx}\|_{L^\infty([1/\sqrt{B},1])} \le B^{-j/2} \vee e^{-\sqrt{B}} \le B^{-j/2}$$

for large $B$. We have that

$$
\begin{aligned}
&\|\delta_B(x+1)x^r\|_{W_p^r([0,1])} \\
&\leq \|(1-\sigma(B(x+1)))x^r\|_{W_\infty^r([0,1])} + \|(1-\sigma(Bx))x^r\|_{W_p^r([0,1])} \\
&= \|\sigma(-B(x+1))x^r\|_{W_\infty^r([0,1])} + \|\sigma(-Bx)x^r\|_{W_p^r([0,1])}.
\end{aligned}
$$

Then by the general Leibniz rule and the bound for $|\sigma^{(j)}|$ above, the first term is bounded as

$$
\begin{aligned}
&\|\sigma(-B(x+1))x^r\|_{W_\infty^r([0,1])} \\
&\leq \left\|x^r e^{-B(x+1)}\right\|_{L^\infty([0,1])} + \sum_{j=0}^r \binom{r}{j}\frac{r!B^j}{j!}\left\|\sigma^{(j)}(-B(x+1))x^j\right\|_{L^\infty([0,1])} \\
&\leq e^{-B} + r^{r+1}\sum_{j=0}^r \binom{r}{j}\frac{r!B^j}{j!}\left\|x^j e^{-B(x+1)}\right\|_{L^\infty([0,1])} \lesssim e^{-B},
\end{aligned}
$$

and the second term is bounded as

$$
\begin{aligned}
&\|\sigma(-Bx)x^r\|_{W_p^r([0,1])} \\
&\leq \left\|x^r e^{-Bx}\right\|_{L^\infty([0,1])} + r!\|\sigma(-Bx)\|_{L^p([0,1])} + \sum_{j=1}^r \binom{r}{j}\frac{r!B^j}{j!}\left\|x^j e^{-Bx}\right\|_{L^\infty([0,1])} \\
&\lesssim B^{-r/2} + \|\sigma(-Bx)\|_{L^p([0,1])} + (B^{-1/2}+\cdots+B^{-r/2}).
\end{aligned}
$$

For the remaining term above we further have

$$
\|\sigma(-Bx)\|_{L^p([0,1])} = \int_0^{1/\sqrt{B}}\frac{\mathrm{d}x}{(1+e^{Bx})^p} + \int_{1/\sqrt{B}}^1\frac{\mathrm{d}x}{(1+e^{Bx})^p} \leq \frac{1}{2^p\sqrt{B}} + \frac{1}{(1+e^{\sqrt{B}})^p},
$$

and hence $\|\sigma(-Bx)x^r\|_{W_p^r([0,1])} \lesssim B^{-1/2}$. $\qquad\square$

The following lemma will also be used to control Sobolev norms.

**Lemma C.3** (De Ryck et al. (2021), Lemma A.6 and A.7). *Let $d \in \mathbb{N}$, $r \in \mathbb{Z}_{\geq 0}$ and $\Omega \subset \mathbb{R}^d$.*

(1) *For $f, g \in W_\infty^r(\Omega)$ it holds that $\|fg\|_{W_\infty^r(\Omega)} \leq 2^r\|f\|_{W_\infty^r(\Omega)}\|g\|_{W_\infty^r(\Omega)}$.*

(2) *Let $d' \in \mathbb{N}$ and $\Omega' \subset \mathbb{R}^{d'}$. For $f \in C^r(\Omega', \mathbb{R})$, $g \in C^r(\Omega, \Omega')$ it holds that*

$$
\|f \circ g\|_{W_\infty^r(\Omega)} \leq 16(e^2 r^4 d'd^2)^r\|f\|_{W_\infty^r(\Omega')}\max_{i=1,\cdots,d'}\|g_i\|_{W_\infty^r(\Omega)}^r.
$$

The next two results provide the basic building blocks of our construction.

**Lemma C.4** (De Ryck et al. (2021), Lemma 3.2). *Let $r \in \mathbb{Z}_{\geq 0}$ and $M > 0$. For every $\epsilon > 0$, there exists a shallow sigmoid network $\zeta : [-M, M] \to \mathbb{R}$ with width at most $\lfloor\frac{3r}{2}\rfloor + 2$ and weights at most $\epsilon^{-r/2}\mathrm{poly}(M)$ such that*

$$
\|\zeta(x) - x^r\|_{W_\infty^r([-M,M])} \leq \epsilon.
$$

**Lemma C.5** (De Ryck et al. (2021), Corollary 3.7). *Let $d \in \mathbb{N}$, $r \in \mathbb{Z}_{\geq 0}$ and $M > 0$. For every $\epsilon > 0$, there exists a sigmoid network $\pi_d : [-M, M]^d \to \mathbb{R}$ with depth $\lceil\log_2 d\rceil$, width $3d$ and weights at most $\epsilon^{-1/2}\mathrm{poly}(M)$ such that*

$$
\left\|\pi_d(x_1,\cdots,x_d) - \prod_{i=1}^d x_i\right\|_{W_\infty^r([-M,M]^d)} \leq \epsilon.
$$

Here, $\mathrm{poly}(\cdot)$ notation ignores multiplicative constants depending only on $r, d$.

## C.2 Smooth Approximation of B-Splines

Recall that the base tensor product B-spline is defined as $\omega_{0,0}(x) = \prod_{i=1}^{d} \iota_r(x_i)$. This subsection is devoted to proving the following smooth approximation result:

**Proposition C.6.** *For all $\epsilon > 0$ sufficiently small, there exists a sigmoid neural network $\breve{\omega}$ with depth $\lceil \log_2 d \rceil + 1$, width $\lfloor \frac{3r}{2} \rfloor d + 4d$ and weights at most $\mathrm{poly}(\epsilon^{-1})$ such that*

$$\|\breve{\omega} - \omega_{0,0}\|_{W_p^r(\mathbb{R})} + \|\breve{\omega} - \omega_{0,0}\|_{L^\infty(\mathbb{R})} \lesssim \epsilon.$$

*Proof.* Fix $\epsilon, \epsilon' > 0$ and let $\zeta$ be the network given by Lemma C.4 with error $\epsilon$ and $M$ replaced by $2r + 3$. From the construction in De Ryck et al. (2021) we see that $\|\zeta\|_{L^\infty(\mathbb{R})} \lesssim \epsilon^{-r/2}$. Also, let $\pi_{2d}$ be the network given by Lemma C.4 with error $\epsilon'$ and $M = \|\zeta\|_{L^\infty(\mathbb{R})}$, so that the weights of $\pi_{2d}$ are bounded as $(\epsilon')^{-1/2} \mathrm{poly}(\epsilon^{-1})$.

From the decomposition (17), we aim to approximate the multivariate polynomial $\prod_{i=1}^{d}(x_i - j_i)^r$ on the interval $\prod_{i=1}^{d}[k_i, k_i + 1)$ by the networks

$$\breve{\pi}_{j,k}(x) := \pi_{2d}(\delta_B(x_1 - k_1), \cdots, \delta_B(x_d - k_d), \zeta(x_1 - j_1), \cdots, \zeta(x_d - j_d))$$

and the tensor product B-spline $\omega_{0,0}(x) = \prod_{i=1}^{d} \iota_r(x_i)$ by the network

$$\breve{\omega}(x) := \sum_{k_\bullet=0}^{r+1} \sum_{j_\bullet=0}^{k_\bullet} a_{j_1} \cdots a_{j_d} \breve{\pi}_{j,k}(x) = \sum_{k_1,\cdots,k_d=0}^{r+1} \sum_{j_1,\cdots,j_d=0}^{k_1,\cdots,k_d} a_{j_1} \cdots a_{j_d} \breve{\pi}_{j,k}(x). \tag{18}$$

Note that we have used the symbol $\bullet$ above to indicate a subscript to be iterated over dimensions $1, \cdots, d$ in a consistent manner. By taking $C > 1$, the clip operation will not affect the output due to the $L^\infty$ bound, so it is ignored.

We proceed to evaluate the error $\|\breve{\omega} - \omega_{0,0}\|_{W_p^r(\mathbb{R}^d)}$ by breaking down into several steps.

**Multiplicative approximation error.** It is easily seen that $\|\delta_B\|_{W_\infty^r(\mathbb{R})} \leq 2B^r \|\sigma\|_{W_\infty^r(\mathbb{R})}$ and

$$\|\zeta\|_{W_\infty^r([-2r-3,r+2])} \leq \|\zeta(x) - x^r + x^r\|_{W_\infty^r([-2r-3,r+2])} \leq \epsilon + (2r + 3)^r + r!.$$

Moreover, $|a_j| \leq 2$. Thus we can bound the error arising from the multiplication network $\pi_{2d}$ (here using the stronger $W_\infty^r$ norm) as

$$\left\| \breve{\omega}(x) - \sum_{k_\bullet=0}^{r+1} \sum_{j_\bullet=0}^{k_\bullet} \prod_{i=1}^{d} a_{j_i} \delta_B(x_i - k_i)\zeta(x_i - j_i) \right\|_{W_\infty^r([-r-2,r+2]^d)}$$

$$\leq \sum_{k_\bullet=0}^{r+1} \sum_{j_\bullet=0}^{k_\bullet} 2^d \left\| \breve{\pi}_{j,k}(x) - \prod_{i=1}^{d} \delta_B(x_i - k_i)\zeta(x_i - j_i) \right\|_{W_\infty^r([-r-2,r+2]^d)}$$

$$\lesssim \sum_{k_\bullet=0}^{r+1} \sum_{j_\bullet=0}^{k_\bullet} 2^d \left\| \pi_{2d}(x_1, \cdots, x_{2d}) - \prod_{i=1}^{2d} x_i \right\|_{W_\infty^r([-M,M]^d)}$$

$$\times \max_{i=1,\cdots,d} \left\{ \|\delta_B(x_i - k_i)\|_{W_\infty^r(\mathbb{R})}, \|\zeta(x_i - j_i)\|_{W_\infty^r([-r-2,r+2])} \right\}^r$$

$$\lesssim 2^d(r+2)^{2d} \max\{2B^r\|\sigma\|_{W_\infty^r(\mathbb{R})}, \epsilon + (2r+3)^r + r!\}^r \epsilon'$$

$$\lesssim B^{r^2} \epsilon' \tag{19}$$

by applying Lemma C.3.

**Monomial approximation error.** Next, it holds for all $0 \leq j_1, \cdots, j_d \leq r + 1$ that

$$\left\| \prod_{i=1}^{d} \zeta(x_i - j_i) - \prod_{i=1}^{d}(x_i - j_i)^r \right\|_{W_\infty^r([-r-2,r+2]^d)}$$

$$\leq \sum_{\ell=1}^{d} \left\| \prod_{i=1}^{\ell}(x_i - j_i)^r \prod_{i=\ell+1}^{d} \zeta(x_i - j_i) - \prod_{i=1}^{\ell-1}(x_i - j_i)^r \prod_{i=\ell}^{d} \zeta(x_i - j_i) \right\|_{W_\infty^r([-r-2,r+2]^d)}$$

$$\leq \sum_{\ell=1}^{d} \left\| ((x_i - j_i)^r - \zeta(x_i - j_i)) \prod_{i=1}^{\ell-1}(x_i - j_i)^r \prod_{i=\ell+1}^{d} \zeta(x_i - j_i) \right\|_{W_\infty^r([-r-2,r+2]^d)}$$

$$\lesssim \sum_{\ell=1}^{d} \|x^r - \zeta(x)\|_{W_\infty^r([-2r-3,r+2])} \|x^r\|_{W_\infty^r([-2r-3,r+2])}^{\ell-1} \|\zeta\|_{W_\infty^r([-2r-3,r+2])}^{d-\ell} \lesssim \epsilon$$

and hence the error due to approximating monomials by $\zeta$ is bounded as

$$\left\| \sum_{k_\bullet=0}^{r+1} \sum_{j_\bullet=0}^{k_\bullet} \prod_{i=1}^{d} a_{j_i} \delta_B(x_i - k_i) \left( \prod_{i=1}^{d} \zeta(x_i - j_i) - \prod_{i=1}^{d}(x_i - j_i)^r \right) \right\|_{W_\infty^r([-r-2,r+2]^d)}$$

$$\lesssim \sum_{k_\bullet=0}^{r+1} \sum_{j_\bullet=0}^{k_\bullet} \|\delta_B\|_{W_\infty^r(\mathbb{R})}^d \left\| \prod_{i=1}^{d} \zeta(x_i - j_i) - \prod_{i=1}^{d}(x_i - j_i)^r \right\|_{W_\infty^r([-r-2,r+2]^d)}$$

$$\lesssim B^{rd}\epsilon. \tag{20}$$

**Indicator approximation error.** For the approximation error of the indicators, one needs to be more careful since $1_{[0,1)} - \delta_B$ is nonsmooth near boundary points. Nonetheless, the difference of the piecewise polynomials (17) at each knot $k = 0, \cdots, r+1$ is always of the form $(x - k)^r$ which will smooth out the error terms. Restricting to each unit interval $J_\ell = \prod_{i=1}^{d}[\ell_i, \ell_i + 1)$ inside the box $[-r-2, r+2]^d$, that is for $-1 \leq \ell_1, \cdots, \ell_d \leq r+1$, gives that

$$\left\| \sum_{k_\bullet=0}^{r+1} \sum_{j_\bullet=0}^{k_\bullet} \prod_{i=1}^{d} a_{j_i} \delta_B(x_i - k_i)(x_i - j_i)^r - \sum_{j_\bullet=0}^{\ell_\bullet} \prod_{i=1}^{d} a_{j_i}(x_i - j_i)^r \right\|_{W_p^r(J_\ell)}$$

$$\leq \left\| \sum_{k_\bullet : |k_\bullet - \ell_\bullet| \leq 1} \sum_{j_\bullet=0}^{k_\bullet} \prod_{i=1}^{d} a_{j_i} \delta_B(x_i - k_i)(x_i - j_i)^r - \sum_{j_\bullet=0}^{\ell_\bullet} \prod_{i=1}^{d} a_{j_i}(x_i - j_i)^r \right\|_{W_p^r(J_\ell)} \tag{21}$$

$$+ \left\| \sum_{k_\bullet : \exists i', |k_{i'} - \ell_{i'}| \geq 2} \sum_{j_\bullet=0}^{k_\bullet} \prod_{i=1}^{d} a_{j_i} \delta_B(x_i - k_i)(x_i - j_i)^r \right\|_{W_p^r(J_\ell)}. \tag{22}$$

Here, we have split the sum over $k_1, \cdots, k_d$ into terms such that $|k_i - \ell_i| \leq 1$ for all $i$, and the remaining terms which contain at least one 'out-of-bounds' index $k_{i'}$ with $|k_{i'} - \ell_{i'}| \geq 2$. Also note that we are now using the $W_p^r$ norm on $J_\ell$ in order to control the boundary remainder terms, although we will again upper bound by the $W_\infty^r$ norm when necessary.

To bound (21), we rearrange the existing terms so that the sum of three neighboring translates of $\delta_B$ suffices to smoothly approximate the true indicator up to boundary terms. Indeed,

$$\sum_{k_\bullet : |k_\bullet - \ell_\bullet| \leq 1} \sum_{j_\bullet=0}^{k_\bullet} \prod_{i=1}^{d} a_{j_i} \delta_B(x_i - k_i)(x_i - j_i)^r$$

$$= \prod_{i=1}^{d} \sum_{k_i : |k_i - \ell_i| \leq 1} \sum_{j_i=0}^{k_i} a_{j_i} \delta_B(x_i - k_i)(x_i - j_i)^r$$

$$= \prod_{i=1}^{d} \sum_{j_i=0}^{\ell_i} \left( a_{j_i}(\delta_B(x_i - \ell_i + 1) + \delta_B(x_i - \ell_i) + \delta_B(x_i - \ell_i - 1))(x_i - j_i)^r \right.$$

$$\left. - a_{\ell_i} \delta_B(x_i - \ell_i + 1)(x_i - \ell_i)^r + a_{\ell_i+1} \delta_B(x_i - \ell_i - 1)(x_i - \ell_i - 1)^r \right).$$

Expanding the above product and separating all boundary terms gives

$$\left\| \sum_{k_\bullet : |k_\bullet - \ell_\bullet| \leq 1} \sum_{j_\bullet = 0}^{k_\bullet} \prod_{i=1}^{d} a_{j_i} \delta_B(x_i - k_i)(x_i - j_i)^r - \sum_{j_\bullet = 0}^{\ell_\bullet} \prod_{i=1}^{d} a_{j_i}(x_i - j_i)^r \right\|_{W_p^r(J_\ell)}$$

$$\leq \left\| \sum_{j_\bullet = 0}^{\ell_\bullet} \prod_{i=1}^{d} a_{j_i}(x_i - j_i)^r \left( \prod_{i=1}^{d} (\delta_B(x_i - \ell_i + 1) + \delta_B(x_i - \ell_i) + \delta_B(x_i - \ell_i - 1)) - 1 \right) \right\|_{W_\infty^r(J_\ell)}$$

$$+ \sum_{\substack{S \subseteq \{1,\cdots,d\} \\ S \neq \varnothing}} \left\| \prod_{i \in S} (a_{\ell_i + 1} \delta_B(x_i - \ell_i - 1)(x_i - \ell_i - 1)^r - a_{\ell_i} \delta_B(x_i - \ell_i + 1)(x_i - \ell_i)^r) \right\|_{W_p^r(J_\ell)}$$

$$\times \left\| \prod_{i \notin S} a_{j_i} (\delta_B(x_i - \ell_i + 1) + \delta_B(x_i - \ell_i) + \delta_B(x_i - \ell_i - 1))(x_i - j_i)^r \right\|_{W_\infty^r(J_\ell)}$$

$$\lesssim \left\| \prod_{i=1}^{d} (\delta_B(x_i - \ell_i + 1) + \delta_B(x_i - \ell_i) + \delta_B(x_i - \ell_i - 1)) - 1 \right\|_{W_\infty^r(J_\ell)}$$

$$+ \sum_{i=1}^{d} \|\delta_B(x_i - \ell_i + 1)(x_i - \ell_i)^r\|_{W_p^r([\ell_i, \ell_i+1))} + \|\delta_B(x_i - \ell_i - 1)(x_i - \ell_i - 1)^r\|_{W_p^r([\ell_i, \ell_i+1))}$$

$$\lesssim \sum_{j=1}^{d} \prod_{i=1}^{j-1} \|\delta_B(x_i - \ell_i + 1) + \delta_B(x_i - \ell_i) + \delta_B(x_i - \ell_i - 1)\|_{W_\infty^r(J_\ell)}$$

$$\times \|\delta_B(x_j - \ell_j + 1) + \delta_B(x_j - \ell_j) + \delta_B(x_j - \ell_j - 1) - 1\|_{W_\infty^r([\ell_j, \ell_j+1))}$$

$$+ \sum_{i=1}^{d} \|\delta_B(x_i - \ell_i + 1)(x_i - \ell_i)^r\|_{W_p^r([\ell_i, \ell_i+1))} + \|\delta_B(x_i - \ell_i - 1)(x_i - \ell_i - 1)^r\|_{W_p^r([\ell_i, \ell_i+1))}$$

$$\lesssim e^{-B} + B^{-1/2}. \tag{23}$$

To isolate the boundary terms, we have used that $\|fg\|_{W_p^r(\Omega)} \lesssim \|f\|_{W_p^r(\Omega)} \|g\|_{W_\infty^r(\Omega)}$ which is easily checked from the general Leibniz rule. The remaining terms can be bounded from above in a straightforward manner. Applying Lemma C.1(1) and Lemma C.2 gives the bound (23).

Finally, we use Lemma C.1(2) to bound (22) as

$$\left\| \sum_{k_\bullet : \exists i', |k_{i'} - \ell_{i'}| \geq 2} \sum_{j_\bullet = 0}^{k_\bullet} \prod_{i=1}^{d} a_{j_i} \delta_B(x_i - k_i)(x_i - j_i)^r \right\|_{W_\infty^r(J_\ell)}$$

$$\lesssim \sum_{k_\bullet = 0}^{r+1} \sum_{j_\bullet = 0}^{k_\bullet} 2^d \|\delta_B\|_{W_\infty^r(\mathbb{R})}^{d-1} \|x^r\|_{W_\infty^r([-2r-3, r+2])}^{d} \sup_{k, \ell : |k-\ell| \geq 2} \|\delta_B(x-k)\|_{W_\infty^r([\ell, \ell+1))}$$

$$\lesssim B^{rd} e^{-B}. \tag{24}$$

**Putting things together.** Combining the bounds (19), (20), (23) and (24), we have shown that the construction (18) approximates $\omega_{0,0}$ on the box $[-r-2, r+2]^d$ as

$$\|\breve{\omega} - \omega_{0,0}\|_{W_p^r([-r-2, r+2]^d)}$$

$$= \left\| \breve{\omega} - \sum_{\ell_\bullet = 0}^{r+1} 1_{J_\ell}(x) \sum_{j_\bullet = 0}^{\ell_\bullet} \prod_{i=1}^{d} a_{j_i}(x_i - j_i)^r \right\|_{W_\infty^r([-r-2, r+2]^d)}$$

$$\leq \left\| \breve{\omega}(x) - \sum_{k_\bullet = 0}^{r+1} \sum_{j_\bullet = 0}^{k_\bullet} \prod_{i=1}^{d} a_{j_i} \delta_B(x_i - k_i) \zeta(x_i - j_i) \right\|_{W_\infty^r([-r-2, r+2]^d)}$$

$$
+ \left\| \sum_{k_\bullet=0}^{r+1} \sum_{j_\bullet=0}^{k_\bullet} \prod_{i=1}^d a_{j_i} \delta_B(x_i - k_i) \left( \prod_{i=1}^d \zeta(x_i - j_i) - \prod_{i=1}^d (x_i - j_i)^r \right) \right\|_{W_\infty^r([-r-2,r+2]^d)}
$$

$$
+ \sum_{\ell_\bullet=-1}^{r+1} \left\| \sum_{k_\bullet=0}^{r+1} \sum_{j_\bullet=0}^{k_\bullet} \prod_{i=1}^d a_{j_i} \delta_B(x_i - k_i)(x_i - j_i)^r - \sum_{j_\bullet=0}^{\ell_\bullet} \prod_{i=1}^d a_{j_i}(x_i - j_i)^r \right\|_{W_p^r(J_\ell)}
$$

$$
\lesssim B^{r^2} \epsilon' + B^{rd} \epsilon + B^{-1/2} + B^{rd} e^{-B}.
$$

It remains to bound the approximation error outside $[-r-2, r+2]^d$. We may decompose the domain into a union of sets of the form $(\mathbb{R} \setminus [-r-2, r+2]) \times \mathbb{R}^{d-1}$ and use Lemma C.1(2) to bound the decay of the corresponding $\delta_B$ component, and an argument similar to (19) yields

$$
\|\breve{\omega} - \omega_{0,0}\|_{W_\infty^r(\mathbb{R}^d \setminus [-r-2,r+2]^d)}
$$

$$
= \left\| \sum_{k_\bullet=0}^{r+1} \sum_{j_\bullet=0}^{k_\bullet} a_{j_1} \cdots a_{j_d} \breve{\pi}_{j,k}(x) \right\|_{W_\infty^r(\mathbb{R}^d \setminus [-r-2,r+2]^d)}
$$

$$
\lesssim \sum_{k_\bullet=0}^{r+1} \sum_{j_\bullet=0}^{k_\bullet} \left\| \breve{\pi}_{j,k}(x) - \prod_{i=1}^d \delta_B(x_i - k_i)\zeta(x_i - j_i) \right\|_{W_\infty^r(\mathbb{R}^d)}
$$

$$
+ \sum_{k_\bullet=0}^{r+1} \sum_{j_\bullet=0}^{k_\bullet} \left\| \prod_{i=1}^d \delta_B(x_i - k_i)\zeta(x_i - j_i) \right\|_{W_\infty^r(\mathbb{R}^d \setminus [-r-2,r+2]^d)}
$$

$$
\lesssim \sum_{k_\bullet=0}^{r+1} \sum_{j_\bullet=0}^{k_\bullet} \left\| \pi_{2d}(x_1, \cdots, x_{2d}) - \prod_{i=1}^{2d} x_i \right\|_{W_\infty^r([-M,M]^d)} \max_{i=1,\cdots,d} \left\{ \|\delta_B\|_{W_\infty^r(\mathbb{R})}, \|\zeta\|_{W_\infty^r(\mathbb{R})} \right\}^r
$$

$$
+ \sum_{k_\bullet=0}^{r+1} \sum_{j_\bullet=0}^{k_\bullet} \|\delta_B\|_{W_\infty^r(\mathbb{R} \setminus [-1,2])} \|\delta_B\|_{W_\infty^r(\mathbb{R})}^{d-1} \|\zeta\|_{W_\infty^r(\mathbb{R})}^d
$$

$$
\lesssim (B^r \vee \|\zeta\|_{W_\infty^r(\mathbb{R})})^r \epsilon' + B^r e^{-B}(B^r)^{d-1} \|\zeta\|_{W_\infty^r(\mathbb{R})}^d.
$$

Moreover from the weight bound in Lemma C.4 we see that $\|\zeta\|_{W_\infty^r(\mathbb{R})} \lesssim (\epsilon^{-r/2})^{2r} = \epsilon^{-r^2}$ since $\zeta$ has depth 2. Adding the resulting bounds finally yields

$$
\|\breve{\omega} - \omega_{0,0}\|_{W_p^r(\mathbb{R}^d)} \lesssim (B^{r^2} \vee \epsilon^{-r^3})\epsilon' + B^{rd}\epsilon + B^{-1/2} + B^{rd} e^{-B} \epsilon^{-r^2 d}.
$$

Hence for any $\epsilon'' > 0$ small enough, we can ensure the above error is bounded as $O(\epsilon'')$ by taking $B \asymp (\epsilon'')^{-2}$, $\epsilon \asymp (\epsilon'')^{2rd+1}$ and $\epsilon' \asymp (\epsilon'')^{2r^4 d + r^3 + 1}$. In addition, we can check that the resulting network $\breve{\omega}$ has depth $\lceil \log_2 d \rceil + 1$, width $\lfloor \frac{3r}{2} \rfloor d + 4d$ and weights at most polynomial in $B, \epsilon, \epsilon'$, thus polynomial in $\epsilon''$.

For the $L^\infty$ approximation error, one can go over the proof and check that the same bounds apply, discarding the bounds for all higher-order derivatives. Finally, the results hold even if $0 < p < 1$ by replacing applications of the triangle inequality by the quasi-norm inequality. $\square$

## C.3 ERROR RATES IN BESOV SPACE

For a sequence of coefficients $\beta = (\beta_{k,\ell})_{k \geq 0, \ell \in I_k}$ define the quasi-norm

$$
\|\beta\|_{b_{p,q}^s} := \left( \sum_{k=0}^\infty \left[ 2^{k(s-d/p)} \left( \sum_{\ell \in I_k} |\beta_{k,\ell}|^p \right)^{1/p} \right]^q \right)^{1/q}
$$

with the appropriate modifications when $p$ or $q = \infty$. To approximate an arbitrary function in a Besov space $B_{p,q}^s(\mathcal{X})$, we make use of the following adaptive recovery result based on B-spline decomposition.

**Lemma C.7** (Dũng (2011; 2013))**.** *Suppose $0 < p, q, u \leq \infty$ and $\Delta < s < r \wedge (r - 1 + 1/p)$ where $\Delta = d(1/p - 1/u)_+$. For any $f \in B_{p,q}^s(\mathcal{X})$ and sufficiently large $N$ there exists $f_N$ which satisfies*

$$\|f - f_N\|_{L^u(\mathcal{X})} \lesssim N^{-s/d} \|f\|_{B_{p,q}^s(\mathcal{X})}$$

*and is of the form*

$$f_N = \sum_{k=0}^K \sum_{\ell \in I_k} \beta_{k,\ell} \omega_{k,\ell} + \sum_{k=K+1}^{K^*} \sum_{i=1}^{n_k} \beta_{k,\ell_i} \omega_{k,\ell_i},$$

*where $(\ell_i)_{i=1}^{n_k} \subset I_k$, $K = \lceil \lambda_1 \log N \rceil$, $K^* = \lceil \nu^{-1} \log \lambda_2 N \rceil + K + 1$, $n_k = \lceil 2^{-\nu(k-K)} \lambda_2 N \rceil$, with $\nu = \frac{s - \Delta}{2\Delta}$ and $\lambda_1, \lambda_2$ chosen independently of $N$ so that $\sum_{k=1}^K |I_k| + \sum_{k=K+1}^{K^*} n_k \leq N$. If $\Delta = 0$ then $K^* = K$. Moreover, the sequence of coefficients can be chosen to satisfy $\|\beta\|_{b_{p,q}^s} \lesssim \|f\|_{B_{p,q}^s(\mathcal{X})}$.*

Together with Proposition C.6, this implies the following approximation result.

**Theorem C.8.** *Suppose $0 < p, q \leq \infty$, $d/p < s < r \wedge (r - 1 + 1/p)$. For all $f \in \mathbb{U}(B_{p,q}^s(\mathcal{X}))$ with $|f| \leq C$ and sufficiently large $N$, there exists a sigmoid neural network $\check{f}_N \in \mathcal{F}_{\mathrm{DNN}}(L, W, S, M)$ where*

$$L = \lceil \log_2 d \rceil + 1, \quad W_0 = \lfloor \tfrac{3r}{2} d \rfloor + d, \quad W = NW_0, \quad S = LW_0^2 + NW_0, \quad M = \mathrm{poly}(N)$$

*such that $\|\check{f}_N - f\|_{L^\infty(\mathcal{X})} \lesssim N^{-s/d}$ and $\|\check{f}_N\|_{B_{p,q}^s(\mathcal{X})}$ is bounded. If $d(1/p - 1/2)_+ < s \leq d/p$, the same result holds with $L^\infty(\mathcal{X})$ replaced by $L^2(\mathcal{X})$.*

*Proof.* Consider the $N$-term approximation $f_N$ given by Lemma C.7. For each B-spline $\omega_{k,\ell}$ in its sum, we construct the network $\check{\omega}_{k,\ell}$ by taking the network $\check{\omega}$ of Proposition C.6 and scaling its input as $x_i \mapsto 2^k x_i - \ell_i$; note that the scaling only changes the input weights by a factor of at most $2^k \leq 2^{K^*} \lesssim \mathrm{poly}(N)$. It follows that

$$\|\check{\omega}_{k,\ell} - \omega_{k,\ell}\|_{W_p^r(\mathbb{R})} \lesssim 2^{kr} \|\check{\omega} - \omega_{0,0}\|_{W_p^r(\mathbb{R})} \lesssim 2^{kr} \epsilon \quad \text{and} \quad \|\check{\omega}_{k,\ell} - \omega_{k,\ell}\|_{L^\infty(\mathbb{R})} \lesssim \epsilon.$$

Now consider the network $\check{f}_N$ obtained by laying each $\check{\omega}_{k,\ell}$ in parallel and scaling the output weights by $\beta_{k,\ell}$, so that (ignoring the clip)

$$\check{f}_N = \sum_{k=0}^K \sum_{\ell \in I_k} \beta_{k,\ell} \check{\omega}_{k,\ell} + \sum_{k=K+1}^{K^*} \sum_{i=1}^{n_k} \beta_{k,\ell_i} \check{\omega}_{k,\ell_i}.$$

Then we have

$$\|\check{f}_N - f_N\|_{W_p^r(\mathcal{X})} + \|\check{f}_N - f_N\|_{L^\infty(\mathcal{X})}$$

$$\leq \sum_{k=0}^K \sum_{\ell \in I_k} \beta_{k,\ell} \left( \|\check{\omega}_{k,\ell} - \omega_{k,\ell}\|_{W_p^r(\mathbb{R})} + \|\check{\omega}_{k,\ell} - \omega_{k,\ell}\|_{L^\infty(\mathbb{R})} \right)$$

$$+ \sum_{k=K+1}^{K^*} \sum_{i=1}^{n_k} \beta_{k,\ell_i} \left( \|\check{\omega}_{k,\ell_i} - \omega_{k,\ell_i}\|_{W_p^r(\mathbb{R})} + \|\check{\omega}_{k,\ell_i} - \omega_{k,\ell_i}\|_{L^\infty(\mathbb{R})} \right)$$

$$\leq \sum_{k=0}^K \sum_{\ell \in I_k} |\beta_{k,\ell}| 2^{kr} \epsilon + \sum_{k=K+1}^{K^*} \sum_{i=1}^{n_k} |\beta_{k,\ell}| 2^{kr} \epsilon$$

$$\leq \sum_{k=0}^{K^*} \left( \sum_{\ell \in I_k} |\beta_{k,\ell}|^p \right)^{1/p} |I_k|^{1-1/p} 2^{kr} \epsilon$$

$$\leq \|\beta\|_{b_{p,q}^s} \sum_{k=0}^{K^*} 2^{k(d/p-s)} 2^{kd(1-1/p)} 2^{kr} \epsilon$$

$$\lesssim 2^{K^*(d+r-s)} \epsilon = \mathrm{poly}(N) \epsilon.$$

Therefore by taking $\epsilon$ (and hence all weights) polynomial in $N$, we can ensure
$$\|\check{f}_N - f_N\|_{W_p^r(\mathcal{X})} + \|\check{f}_N - f_N\|_{L^\infty(\mathcal{X})} \lesssim N^{-s/d},$$
and thus $\|\check{f}_N - f\|_{L^\infty(\mathcal{X})} \lesssim N^{-s/d}$ as desired. In particular, for sufficiently large $N$ it follows that $\|\check{f}_N\|_{L^\infty(\mathcal{X})} \lesssim C + N^{-s/d} < \bar{C}$ so that including the subsequent clip operation does not affect $\check{f}_N$. Moreover, since the B-spline expansion of $f - f_N$ has coefficient zero for all $\omega_{k,\ell}$ with resolution $k \leq K$, it follows that

$$\|f - f_N\|_{B_{p,q}^s(\mathcal{X})} \lesssim \left( \sum_{k=K+1}^\infty \left[ 2^{k(s-d/p)} \left( \sum_{\ell \in I_k} |\beta_{k,\ell}|^p \right)^{1/p} \right]^q \right)^{1/q} \to 0$$

as $N \to \infty$, $K = \lceil \lambda_1 \log N \rceil \to \infty$. Hence

$$\left| \|\check{f}_N\|_{B_{p,q}^s(\mathcal{X})} - \|f\|_{B_{p,q}^s(\mathcal{X})} \right| \leq \|\check{f}_N - f_N\|_{W_p^r(\mathcal{X})} + \|f_N - f\|_{B_{p,q}^s(\mathcal{X})} \to 0$$

and $\|\check{f}_N\|_{B_{p,q}^s(\mathcal{X})}$ is uniformly bounded. $\qquad \square$

The $\delta$-covering number $\mathcal{N}(\mathcal{S}, \rho, \delta)$ of a metric space $(\mathcal{S}, \rho)$ is defined as the minimal number of balls with radius $\delta$ needed to cover $\mathcal{S}$. The covering number of $\mathcal{F}_{\text{DNN}}(L, W, S, M)$ in $L^\infty$-norm can be bounded similarly to the ReLU DNN class (Suzuki, 2019, Lemma 3) with a slightly better bound.

**Lemma C.9** (covering number of sigmoid DNN class). *If the norm bound $M \geq 4$, it holds that*
$$\log \mathcal{N}(\mathcal{F}_{\text{DNN}}(L, W, S, M), \|\cdot\|_{L^\infty(\mathcal{X})}, \delta) \leq (L+3)S \log MW + S \log \delta^{-1}.$$

*Proof.* Consider $f, \tilde{f} \in \mathcal{F}_{\text{DNN}}(L, W, S, M)$ given as
$$f = \text{clip}_{\underline{C}, \bar{C}} \circ (\mathbf{W}^{(L)}\sigma + b^{(L)}) \circ \cdots \circ (\mathbf{W}^{(1)}\text{id} + b^{(1)}),$$
$$\tilde{f} = \text{clip}_{\underline{C}, \bar{C}} \circ (\tilde{\mathbf{W}}^{(L)}\sigma + \tilde{b}^{(L)}) \circ \cdots \circ (\tilde{\mathbf{W}}^{(1)}\text{id} + \tilde{b}^{(1)}),$$
such that $\|\mathbf{W}^{(\ell)} - \tilde{\mathbf{W}}^{(\ell)}\|_\infty, \|b^{(\ell)} - \tilde{b}^{(\ell)}\|_\infty \leq \delta$. Also denote $A_{L+1}(f) = B_1(f) = \text{id}$ and
$$A_\ell(f) = \text{clip}_{\underline{C}, \bar{C}} \circ (\mathbf{W}^{(L)}\sigma + b^{(L)}) \circ \cdots \circ (\mathbf{W}^{(\ell)}\text{id} + b^{(\ell)}),$$
$$B_\ell(f) = \sigma \circ (\mathbf{W}^{(\ell-1)}\sigma + b^{(\ell-1)}) \circ \cdots \circ (\mathbf{W}^{(1)}\text{id} + b^{(1)}),$$
so that $f = A_{\ell+1}(f) \circ (\mathbf{W}^{(\ell)}\text{id} + b^{(\ell)}) \circ B_\ell(f)$. Then $A_\ell(f)$ is $(\frac{1}{4})^{L-\ell}(MW)^{L-\ell+1}$-Lipschitz with respect to the $L^\infty$-norm and $\|B_\ell(f)\|_{L^\infty} \leq 1$. It follows that

$$\|f - \tilde{f}\|_{L^\infty(\mathcal{X})}$$
$$\leq \left\| \sum_{\ell=1}^L A_{\ell+1}(f) \circ (\mathbf{W}^{(\ell)}\text{id} + b^{(\ell)}) \circ B_\ell(\tilde{f}) - A_{\ell+1}(f) \circ (\tilde{\mathbf{W}}^{(\ell)}\text{id} + \tilde{b}^{(\ell)}) \circ B_\ell(\tilde{f}) \right\|_{L^\infty(\mathcal{X})}$$
$$\leq \sum_{\ell=1}^L \frac{(MW)^{L-\ell+1}}{4^{L-\ell}}(W+1)\delta \leq 8(W+1)\left(\frac{MW}{4}\right)^L \delta =: \delta',$$

assuming $M \geq 4$. Thus for a fixed sparsity pattern the $\delta'$-covering number is bounded by dividing the range $[-M, M]$ of all $S$ nonzero parameters into intervals of length $\delta$ and counting all possible combinations,

$$\left(\frac{2M}{\delta}\right)^S = \left(16M(W+1)\left(\frac{MW}{4}\right)^L \frac{1}{\delta'}\right)^S.$$

Moreover the number of possible sparsity patterns is bounded as $\binom{L(W^2+W)}{S} \leq (L(W^2+W))^S$. Noting that $4(W+1)^2 \leq 16W^2 \leq M^2W^2$ and $4L \leq 4^L$, we conclude:

$$\mathcal{N}(\mathcal{F}_{\text{DNN}}(L, W, S, M), \|\cdot\|_{L^\infty(\mathcal{X})}, \delta) \leq \left(16LMW(W+1)^2\left(\frac{MW}{4}\right)^L \frac{1}{\delta}\right)^S$$

$$\leq \left(4LM^3W^3\left(\frac{MW}{4}\right)^L \frac{1}{\delta}\right)^S \leq (MW)^{(L+3)S}\delta^{-S},$$

as desired. $\qquad \square$

# D    PROOF OF THEOREM 3.1

We first present the proof of the general upper bound for estimation risk, which contains our main techniques, over Sections D.1-D.6. We then specialize to the Besov setting in Section D.7 by applying the smooth DNN analysis from Section C.3, which proves the rate under Assumption 4* or Assumption 4 with domain restriction. Finally, we show how the proof can be modified to incorporate regularization under Assumption 4 in Section E.1 as part of the derivation of the non-projected rates.

## D.1    REDUCTION OF STAGE 2 ERROR

We begin with the following oracle inequality, which is essentially a consequence of e.g. Lemma 4 of Schmidt-Hieber (2020). This starting point is necessary to utilize the definition of $\hat{\theta}_x$ as the empirical risk minimizer of Stage 2.

**Lemma D.1** (oracle inequality for NPIR). *Denote the $\delta_z$-covering number of the Stage 2 DNN class $\mathcal{F}_z$ as $\mathcal{N}_z := \mathcal{N}(\mathcal{F}_z, \|\cdot\|_{L^\infty(\mathcal{Z})}, \delta_z)$. Then there exists a constant $C_1$ depending only on $\bar{C}$ such that for all $\delta_z > 0$ with $\log \mathcal{N}_z > 1$ it holds conditional on $\mathcal{D}_1$,*

$$
\mathbb{E}_{\mathcal{D}_2} \left[ \|T f_{\mathrm{str}} - \hat{\mathbb{E}}_{X|Z}[\psi_{\hat{\theta}_x}]\|_{L^2(\mathcal{P}_{\mathcal{Z}})}^2 \right]
$$
$$
\leq 4 \inf_{\theta_x \in \Theta_x} \|T f_{\mathrm{str}} - \hat{\mathbb{E}}_{X|Z}[\psi_{\theta_x}]\|_{L^2(\mathcal{P}_{\mathcal{Z}})}^2 + C_1 \left( \frac{\log \mathcal{N}_z}{n} + \delta_z \right).
$$

Note that even though the risk is being minimized with respect to the Stage 2 parameter $\theta_x$, the generalization gap depends on the covering number of the Stage 1 DNN class which contains the actual regression model $\hat{\mathbb{E}}_{X|Z}[\psi_{\hat{\theta}_x}]$ for $T f_{\mathrm{str}}$. Also, the multiplicative factor 4 can be replaced by any constant larger than 1.

*Proof.* We may repeat the proof of Theorem 2.6 of Hayakawa & Suzuki (2020) while replacing the regression model $Y = f^\circ(X) + \xi$, where $\xi$ was assumed to be i.i.d. Gaussian noise, with the NPIR model $Y = T f_{\mathrm{str}}(Z) + \eta$ (4) and the class of estimators by $\mathcal{F}_z$. Here, we only provide the necessary modifications. For the loss

$$
\Pi(\theta_x) = \|T f_{\mathrm{str}} - \hat{\mathbb{E}}_{X|Z}[\psi_{\theta_x}]\|_{L^2(\mathcal{P}_{\mathcal{Z}})}^2
$$

and the corresponding empirical quantity[4]

$$
\hat{\Pi}(\theta_x) = \mathbb{E}_{\mathcal{D}_2} \left[ \frac{1}{n} \sum_{i=1}^{n} \left( T f_{\mathrm{str}}(\tilde{z}_i) - \hat{\mathbb{E}}_{X|Z}[\psi_{\theta_x}](\tilde{z}_i) \right)^2 \right],
$$

it can be shown in the same way that

$$
\left| \hat{\Pi}(\hat{\theta}_x) - \Pi(\hat{\theta}_x) \right| \leq \frac{1}{2} \Pi(\hat{\theta}_x) + C_1' \left( \frac{\log \mathcal{N}_z}{n} + \delta_z \right). \tag{25}
$$

Moreover, using that $\mathbb{E}[\eta] = 0$ and $\mathbb{E}[\eta \phi_{\theta_z}(Z)] = \mathbb{E}[\phi_{\theta_z}(Z) \mathbb{E}[\eta|Z]] = 0$ for all $\phi_{\theta_z} \in \mathcal{F}_z$, it can be shown that

$$
\hat{\Pi}(\hat{\theta}_x) \leq 2 \|T f_{\mathrm{str}} - \hat{\mathbb{E}}_{X|Z}[\psi_{\theta_x}]\|_{L^2(\mathcal{P}_{\mathcal{Z}})}^2 + \frac{4\delta_z}{n} \mathbb{E} \left[ \sum_{i=1}^{n} |\eta_i| \right] + \frac{4}{n} \mathbb{E} \left[ \max_{j \leq \mathcal{N}_z} \varepsilon_j^2 \right] + 4\bar{C}\delta_z
$$

for all $\theta_x \in \Theta_x$, where $\phi_{\theta_{z,j}}$ for $j \leq \mathcal{N}_z$ is a $\delta_z$-covering of $\mathcal{F}_z$ and

$$
\varepsilon_j := \frac{\sum_{i=1}^{n} \eta_i (\phi_{\theta_{z,j}}(\tilde{z}_i) - T f_{\mathrm{str}}(\tilde{z}_i))}{\sqrt{\sum_{i=1}^{n} (\phi_{\theta_{z,j}}(\tilde{z}_i) - T f_{\mathrm{str}}(\tilde{z}_i))^2}}.
$$

---

[4]Here we are abusing notation to allow for both a fixed $\theta_x \in \Theta_x$ and also an estimator $\hat{\theta}_x$ which is a map from the data to $\Theta_x$. For the former, it is clear that $\Pi(\theta_x) = \hat{\Pi}(\theta_x)$ always. We are interested in bounding the gap (25) for the latter.

Here, since $\eta_i | Z = \tilde{z}_i$ is $\sigma_1$-subgaussian, we have $\mathbb{E}[|\eta_i| \,|\, Z = \tilde{z}_i] \leq \sqrt{2\pi}\sigma_1$ and $\mathbb{E}[|\eta_i|] \leq \sqrt{2\pi}\sigma_1$. Furthermore, each $\varepsilon_j$ is also $\sigma_1$-subgaussian conditioned on the data $\tilde{z}_1, \cdots, \tilde{z}_n$ as an $L^2$-projection of $(\eta_1, \cdots, \eta_n)$, and hence

$$
\begin{aligned}
\exp\left( \frac{1}{4\sigma_1^2} \mathbb{E}\left[ \max_{j \leq \mathcal{N}_z} \varepsilon_j^2 \right] \right) &\leq \mathbb{E}\left[ \max_{j \leq \mathcal{N}_z} \exp\left( \frac{\varepsilon_j^2}{4\sigma_1^2} \right) \right] \\
&\leq \sum_{j=1}^{\mathcal{N}_z} \mathbb{E}\left[ \exp\left( \frac{\varepsilon_j^2}{4\sigma_1^2} \right) \right] \\
&= \sum_{j=1}^{\mathcal{N}_z} \int_0^\infty P\left( |\varepsilon_j| \geq 2\sigma_1 \sqrt{\log u} \right) \mathrm{d}u \\
&\leq \mathcal{N}_z \int_1^\infty \frac{2}{u^2} \wedge 1 \, \mathrm{d}u = 2\sqrt{2}\mathcal{N}_z.
\end{aligned}
$$

This shows that $\mathbb{E}[\max_j \varepsilon_j^2] \leq 4\sigma_1^2 \log 2\sqrt{2}\mathcal{N}_z$, which combined with (25) concludes the desired statement. □

Now let $\theta_x = \theta_x^*$ denote a minimizer of $\|Tf_{\mathrm{str}} - T\psi_{\theta_x}\|_{L^2(\mathcal{P}_{\mathcal{Z}})}$. It holds that

$$
\begin{aligned}
\inf_{\theta_x \in \Theta_x} \|Tf_{\mathrm{str}} - \hat{\mathbb{E}}_{X|Z}[\psi_{\theta_x}]\|_{L^2(\mathcal{P}_{\mathcal{Z}})}^2 &\leq \|Tf_{\mathrm{str}} - \hat{\mathbb{E}}_{X|Z}[\psi_{\theta_x^*}]\|_{L^2(\mathcal{P}_{\mathcal{Z}})}^2 \\
&\leq 2\|Tf_{\mathrm{str}} - T\psi_{\theta_x^*}\|_{L^2(\mathcal{P}_{\mathcal{Z}})}^2 + 2\|T\psi_{\theta_x^*} - \hat{\mathbb{E}}_{X|Z}[\psi_{\theta_x^*}]\|_{L^2(\mathcal{P}_{\mathcal{Z}})}^2.
\end{aligned}
$$

Then the projected error can be bounded using Lemma D.1 as

$$
\begin{aligned}
&\mathbb{E}_{\mathcal{D}_2}\left[ \|Tf_{\mathrm{str}} - T\hat{f}_{\mathrm{str}}\|_{L^2(\mathcal{P}_{\mathcal{Z}})}^2 \right] \\
&\leq 2\mathbb{E}_{\mathcal{D}_2}\left[ \|Tf_{\mathrm{str}} - \hat{\mathbb{E}}_{X|Z}[\psi_{\hat{\theta}_x}]\|_{L^2(\mathcal{P}_{\mathcal{Z}})}^2 \right] + 2\mathbb{E}_{\mathcal{D}_2}\left[ \|\hat{\mathbb{E}}_{X|Z}[\psi_{\hat{\theta}_x}] - T\hat{f}_{\mathrm{str}}\|_{L^2(\mathcal{P}_{\mathcal{Z}})}^2 \right] \\
&\leq 16\|Tf_{\mathrm{str}} - T\psi_{\theta_x^*}\|_{L^2(\mathcal{P}_{\mathcal{Z}})}^2 + 16\|T\psi_{\theta_x^*} - \hat{\mathbb{E}}_{X|Z}[\psi_{\theta_x^*}]\|_{L^2(\mathcal{P}_{\mathcal{Z}})}^2 \\
&\quad + 2C_1 \left( \frac{\log \mathcal{N}_z}{n} + \delta_z \right) + 2\mathbb{E}_{\mathcal{D}_2}\left[ \|\hat{\mathbb{E}}_{X|Z}[\psi_{\hat{\theta}_x}] - T\psi_{\hat{\theta}_x}\|_{L^2(\mathcal{P}_{\mathcal{Z}})}^2 \right] \\
&\leq 16 \inf_{\theta_x \in \Theta_x} \|Tf_{\mathrm{str}} - T\psi_{\theta_x}\|_{L^2(\mathcal{P}_{\mathcal{Z}})}^2 + 2C_1 \left( \frac{\log \mathcal{N}_z}{n} + \delta_z \right) \tag{26} \\
&\quad + 18 \sup_{\theta_x \in \Theta_x} \|T\psi_{\theta_x} - \hat{\mathbb{E}}_{X|Z}[\psi_{\theta_x}]\|_{L^2(\mathcal{P}_{\mathcal{Z}})}^2, \tag{27}
\end{aligned}
$$

where the projected Stage 2 approximation error and covering number (26) will be explicitly evaluated later.

It thus becomes necessary to uniformly control the expected supremum of the Stage 1 estimation error over the hypothesis space $\mathcal{F}_x$ which is highly nontrivial. This is achieved by controlling both the supremum of the corresponding empirical process:

$$
\sup_{\theta_x \in \Theta_x} \frac{1}{m} \sum_{i=1}^m \left( T\psi_{\theta_x}(z_i) - \hat{\mathbb{E}}_{X|Z}[\psi_{\theta_x}](z_i) \right)^2, \tag{28}
$$

and the supremum of their difference:

$$
\sup_{\theta_x \in \Theta_x} \left[ \frac{1}{m} \sum_{i=1}^m \left( T\psi_{\theta_x}(z_i) - \hat{\mathbb{E}}_{X|Z}[\psi_{\theta_x}](z_i) \right)^2 - \|T\psi_{\theta_x} - \hat{\mathbb{E}}_{X|Z}[\psi_{\theta_x}]\|_{L^2(\mathcal{P}_{\mathcal{Z}})}^2 \right], \tag{29}
$$

by carefully reducing to a well-chosen *dynamic* cover of the hypothesis spaces $\mathcal{F}_x, \mathcal{F}_z$. We detail this approach over the following subsections.

### D.2 Constructing the Dynamic Cover

We first introduce the essential tools for our proof technique. Fix $\delta_x, \delta_z > 0$. Let the functions $\psi_{\theta_{x,j}}$ for $j = 1, \cdots, \mathcal{N}_x := \mathcal{N}(\mathcal{F}_x, \|\cdot\|_{L^\infty(\mathcal{X})}, \delta_x)$ form a $\delta_x$-cover of $\mathcal{F}_x$ and let $\phi_{\theta_{z,k}}$ for $k = 1, \cdots, \mathcal{N}_z := \mathcal{N}(\mathcal{F}_z, \|\cdot\|_{L^\infty(\mathcal{Z})}, \delta_z)$ be a $\delta_z$-cover of $\mathcal{F}_z$. Naively one would attempt to bound the desired supremum over $\mathcal{F}_x$, say

$$\sup_{\theta_x \in \Theta_x} \|T\psi_{\theta_x} - \hat{\mathbb{E}}_{X|Z}[\psi_{\theta_x}]\|^2_{L^2(\mathcal{P}_{\mathcal{Z}})},$$

with the supremum over the cover

$$\sup_{j \le \mathcal{N}_x} \|T\psi_{\theta_{x,j}} - \hat{\mathbb{E}}_{X|Z}[\psi_{\theta_{x,j}}]\|^2_{L^2(\mathcal{P}_{\mathcal{Z}})}.$$

However this is not directly feasible since the Stage 1 estimation operator $\hat{\mathbb{E}}_{X|Z}$ can be ill-behaved; approximating $\psi_{\theta_x}$ by the element $\psi_{\theta_{x,j}}$ satisfying $\|\psi_{\theta_x} - \psi_{\theta_{x,j}}\|_{L^\infty(\mathcal{X})} \le \delta_x$ does not guarantee that $\hat{\mathbb{E}}_{X|Z}[\psi_{\theta_{x,j}}]$ is a good approximation of $\hat{\mathbb{E}}_{X|Z}[\psi_{\theta_x}]$, even in $L^2(\mathcal{P}_{\mathcal{Z}})$-norm. Instead, we must construct an extended cover $\hat{\mathcal{C}}$ that approximates all possible combinations of $\psi_{\theta_x}$ and $\hat{\mathbb{E}}_{X|Z}[\psi_{\theta_x}]$. We also give a similar construction for the population conditional mean approximation.

**Definition D.2** (dynamic extended cover). The pair of elements $(\psi_{\theta_{x,j}}, \phi_{\theta_{z,k}})$ is said to be a *joint empirical approximator* of $\psi_{\theta_x} \in \mathcal{F}_x$ if

$$\|\psi_{\theta_x} - \psi_{\theta_{x,j}}\|_{L^\infty(\mathcal{X})} \le \delta_x \quad \text{and} \quad \|\hat{\mathbb{E}}_{X|Z}[\psi_{\theta_x}] - \phi_{\theta_{z,k}}\|_{L^\infty(\mathcal{Z})} \le \delta_z \tag{30}$$

are both satisfied, which always exists for each $\theta_x$. Similarly, $(\psi_{\theta_{x,j}}, \phi_{\theta_{z,k}})$ is said to be a *joint population approximator* of $\psi_{\theta_x}$ if for the $L^2$-minimizer $\theta_z^* = \arg\min_{\theta_z \in \Theta_z} \|T\psi_{\theta_x} - \phi_{\theta_z}\|_{L^2(\mathcal{P}_{\mathcal{Z}})}$ corresponding to $\theta_x$,

$$\|\psi_{\theta_x} - \psi_{\theta_{x,j}}\|_{L^\infty(\mathcal{X})} \le \delta_x \quad \text{and} \quad \|\phi_{\theta_z^*} - \phi_{\theta_{z,k}}\|_{L^\infty(\mathcal{Z})} \le \delta_z \tag{31}$$

are both satisfied. Moreover, the subsets $\hat{\mathcal{C}}, \mathcal{C}^* \subset \{1, \cdots, \mathcal{N}_x\} \times \{1, \cdots, \mathcal{N}_z\}$ are defined as

$$\hat{\mathcal{C}} := \left\{ (j,k) \mid (\psi_{\theta_{x,j}}, \phi_{\theta_{z,k}}) \text{ is a joint empirical approximator of } \psi_{\theta_x} \text{ for some } \theta_x \in \Theta_x \right\},$$

$$\mathcal{C}^* := \left\{ (j,k) \mid (\psi_{\theta_{x,j}}, \phi_{\theta_{z,k}}) \text{ is a joint population approximator of } \psi_{\theta_x} \text{ for some } \theta_x \in \Theta_x \right\}.$$

Note that $\hat{\mathcal{C}}$ is a random subset dependent on the data $\mathcal{D}_1$ and hence one must be careful when proving uniform bounds using $\hat{\mathcal{C}}$. In contrast, $\mathcal{C}^*$ depends only on the operator $T$ and predetermined covers $\theta_{x,j}, \theta_{z,k}$. Also note that $|\hat{\mathcal{C}}|, |\mathcal{C}^*| \le \mathcal{N}_x \times \mathcal{N}_z$ by definition.

### D.3 Reduction of Supremal Stage 1 Error

We now demonstrate how to appropriately reduce the supremal empirical error (28) over the extended covers $\hat{\mathcal{C}}, \mathcal{C}^*$ defined above. For each $\theta_x \in \Theta_x$, by the definition of $\hat{\mathbb{E}}_{X|Z}[\psi_{\theta_x}]$ as the empirical risk minimizer for the Stage 1 loss, it holds for $\theta_z^* = \arg\min_{\theta_z \in \Theta_z} \|T\psi_{\theta_x} - \phi_{\theta_z}\|_{L^2(\mathcal{P}_{\mathcal{Z}})}$ that

$$\frac{1}{m} \sum_{i=1}^m \left( \psi_{\theta_x}(x_i) - \hat{\mathbb{E}}_{X|Z}[\psi_{\theta_x}](z_i) \right)^2 \le \frac{1}{m} \sum_{i=1}^m \left( \psi_{\theta_x}(x_i) - \phi_{\theta_z^*}(z_i) \right)^2.$$

By adding and subtracting the true conditional means $T\psi_{\theta_x}(z_i)$ from both sides and rearranging, we obtain

$$\frac{1}{m} \sum_{i=1}^m \left( T\psi_{\theta_x}(z_i) - \hat{\mathbb{E}}_{X|Z}[\psi_{\theta_x}](z_i) \right)^2$$

$$\le \frac{1}{m} \sum_{i=1}^m \left( T\psi_{\theta_x}(z_i) - \phi_{\theta_z^*}(z_i) \right)^2 + \frac{2}{m} \sum_{i=1}^m \left( \hat{\mathbb{E}}_{X|Z}[\psi_{\theta_x}](z_i) - \phi_{\theta_z^*}(z_i) \right) \left( \psi_{\theta_x}(x_i) - T\psi_{\theta_x}(z_i) \right)$$

$$\le \inf_{\theta_z \in \Theta_z} \|T\psi_{\theta_x} - \phi_{\theta_z}\|^2_{L^2(\mathcal{P}_{\mathcal{Z}})} \tag{32}$$

$$+ \frac{1}{m} \sum_{i=1}^{m} \left( T\psi_{\theta_x}(z_i) - \phi_{\theta_z^*}(z_i) \right)^2 - \| T\psi_{\theta_x} - \phi_{\theta_z^*} \|_{L^2(\mathcal{P}_{\mathcal{Z}})}^2 \tag{33}$$

$$+ \frac{2}{m} \sum_{i=1}^{m} \left( \hat{\mathbb{E}}_{X|Z}[\psi_{\theta_x}](z_i) - T\psi_{\theta_x}(z_i) \right) \left( \psi_{\theta_x}(x_i) - T\psi_{\theta_x}(z_i) \right) \tag{34}$$

$$+ \frac{2}{m} \sum_{i=1}^{m} \left( T\psi_{\theta_x}(z_i) - \phi_{\theta_z^*}(z_i) \right) \left( \psi_{\theta_x}(x_i) - T\psi_{\theta_x}(z_i) \right). \tag{35}$$

Now that the Stage 1 approximation error (32) has been isolated, we may reduce each of (33)-(35) to the supremum over the respective extended covers. For (34), let $(\psi_{\theta_{x,j}}, \phi_{\theta_{z,k}})$ be a joint empirical approximator of $\psi_{\theta_x}$ and define the residuals $\xi_{j,i} := \psi_{\theta_{x,j}}(x_i) - T\psi_{\theta_{x,j}}(z_i)$. By the condition (30) and due to the $L^\infty$-contractivity

$$\| T\psi_{\theta_x} - T\psi_{\theta_{x,j}} \|_{L^\infty(\mathcal{Z})} \leq \| \psi_{\theta_x} - \psi_{\theta_{x,j}} \|_{L^\infty(\mathcal{X})} \leq \delta_x$$

of $T$, it follows that

$$\frac{2}{m} \sum_{i=1}^{m} \left( \hat{\mathbb{E}}_{X|Z}[\psi_{\theta_x}](z_i) - T\psi_{\theta_x}(z_i) \right) \left( \psi_{\theta_x}(x_i) - T\psi_{\theta_x}(z_i) \right)$$

$$\leq \frac{2}{m} \sum_{i=1}^{m} \left( \phi_{\theta_{z,k}}(z_i) - T\psi_{\theta_{x,j}}(z_i) \right) \left( \psi_{\theta_x}(x_i) - T\psi_{\theta_x}(z_i) \right) + 4\bar{C}(\delta_x + \delta_z)$$

$$\leq \frac{2}{m} \sum_{i=1}^{m} \left( \phi_{\theta_{z,k}}(z_i) - T\psi_{\theta_{x,j}}(z_i) \right) \xi_{j,i} + 4\bar{C}(3\delta_x + \delta_z).$$

Similarly for (33) and (35), letting $(\psi_{\theta_{x,j'}}, \phi_{\theta_{z,k'}})$ be a joint population approximator of $\psi_{\theta_x}$, it is easily checked that

$$\frac{1}{m} \sum_{i=1}^{m} \left( T\psi_{\theta_x}(z_i) - \phi_{\theta_z^*}(z_i) \right)^2 - \| T\psi_{\theta_x} - \phi_{\theta_z^*} \|_{L^2(\mathcal{P}_{\mathcal{Z}})}^2$$

$$\leq \frac{1}{m} \sum_{i=1}^{m} \left( T\psi_{\theta_{x,j'}}(z_i) - \phi_{\theta_{z,k'}}(z_i) \right)^2 - \| T\psi_{\theta_{x,j'}} - \phi_{\theta_{z,k'}} \|_{L^2(\mathcal{P}_{\mathcal{Z}})}^2 + 4(2\bar{C}+1)(\delta_x + \delta_z)$$

and

$$\frac{2}{m} \sum_{i=1}^{m} \left( T\psi_{\theta_x}(z_i) - \phi_{\theta_z^*}(z_i) \right) \left( \psi_{\theta_x}(x_i) - T\psi_{\theta_x}(z_i) \right)$$

$$\leq \frac{2}{m} \sum_{i=1}^{m} \left( T\psi_{\theta_{x,j'}}(z_i) - \phi_{\theta_{z,k'}}(z_i) \right) \xi_{j',i} + 4\bar{C}(3\delta_x + \delta_z)$$

by taking $\delta_x, \delta_z < 1$. Hence we have shown that

$$\sup_{\theta_x \in \Theta_x} \frac{1}{m} \sum_{i=1}^{m} \left( T\psi_{\theta_x}(z_i) - \hat{\mathbb{E}}_{X|Z}[\psi_{\theta_x}](z_i) \right)^2$$

$$\leq \sup_{\theta_x \in \Theta_x} \left[ \inf_{\theta_z \in \Theta_z} \| T\psi_{\theta_x} - \phi_{\theta_z} \|_{L^2(\mathcal{P}_{\mathcal{Z}})}^2 \right] + (32\bar{C}+4)(\delta_x + \delta_z) \tag{36}$$

$$+ \sup_{(j,k) \in \mathcal{C}^*} \left| \frac{1}{m} \sum_{i=1}^{m} \left( T\psi_{\theta_{x,j}}(z_i) - \phi_{\theta_{z,k}}(z_i) \right)^2 - \| T\psi_{\theta_{x,j}} - \phi_{\theta_{z,k}} \|_{L^2(\mathcal{P}_{\mathcal{Z}})}^2 \right| \tag{37}$$

$$+ \sup_{(j,k) \in \hat{\mathcal{C}}} \frac{2}{m} \sum_{i=1}^{m} \left( \phi_{\theta_{z,k}}(z_i) - T\psi_{\theta_{x,j}}(z_i) \right) \xi_{j,i} \tag{38}$$

$$+ \sup_{(j,k) \in \mathcal{C}^*} \frac{2}{m} \sum_{i=1}^{m} \left( T\psi_{\theta_{x,j}}(z_i) - \phi_{\theta_{z,k}}(z_i) \right) \xi_{j,i}. \tag{39}$$

In addition, repeating the above argument for the supremal difference term (29) yields the following reduction to the dynamic cover $\hat{\mathcal{C}}$,

$$\sup_{\theta_x \in \Theta_x} \left[ \frac{1}{m} \sum_{i=1}^{m} \left( T\psi_{\theta_x}(z_i) - \hat{\mathbb{E}}_{X|Z}[\psi_{\theta_x}](z_i) \right)^2 - \|T\psi_{\theta_x} - \hat{\mathbb{E}}_{X|Z}[\psi_{\theta_x}]\|_{L^2(\mathcal{P}_Z)}^2 \right]$$

$$\leq \sup_{(j,k) \in \hat{\mathcal{C}}} \left| \frac{1}{m} \sum_{i=1}^{m} \left( T\psi_{\theta_{x,j}}(z_i) - \phi_{\theta_{z,k}}(z_i) \right)^2 - \|T\psi_{\theta_{x,j}} - \phi_{\theta_{z,k}}\|_{L^2(\mathcal{P}_Z)}^2 \right| \tag{40}$$

$$+ (8\bar{C} + 4)(\delta_x + \delta_z).$$

We now evaluate the expected value of each of the suprema (37)-(40) by adapting standard complexity-based arguments to exploit the definitions of $\hat{\mathcal{C}}, \mathcal{C}^*$. Again, the approximation error (36) will be analyzed later.

## D.4 BOUNDING SUBGAUSSIAN COMPLEXITIES (38), (39)

Define the auxiliary random variables

$$\varepsilon_{j,k} = \frac{\sum_{i=1}^{m}(\phi_{\theta_{z,k}}(z_i) - T\psi_{\theta_{x,j}}(z_i))\xi_{j,i}}{\sqrt{\sum_{i=1}^{m}(\phi_{\theta_{z,k}}(z_i) - T\psi_{\theta_{x,j}}(z_i))^2}}, \quad 1 \leq j \leq \mathcal{N}_x, \quad 1 \leq k \leq \mathcal{N}_z,$$

where $\varepsilon_{j,k} = 0$ if the denominator is zero. Note that (38), (39) are similar to classical complexity measures of function classes, but with the noise terms replaced by the function-dependent residuals $\xi_{j,i} = \psi_{\theta_{x,j}}(x_i) - T\psi_{\theta_{x,j}}(z_i)$. Nonetheless, conditioned on $z_i$ we have $\mathbb{E}[\xi_{j,i}|z_i] = 0$ and since $|\psi_{\theta_{x,j}}(x_i)| \leq \bar{C}$, each $(\xi_{j,i})_{i=1}^{m}$ is independently $\bar{C}$-subgaussian conditioned on $(z_i)_{i=1}^{m}$. It follows that $\varepsilon_{j,k}$ is also $\bar{C}$-subgaussian, and repeating the tail bound argument in Lemma D.1 over all pairs $(j,k)$ we obtain

$$\exp\left( \frac{1}{4\bar{C}^2} \mathbb{E}\left[ \sup_{j \leq \mathcal{N}_x, k \leq \mathcal{N}_z} \varepsilon_{j,k}^2 \right] \right) \leq \sum_{j=1}^{\mathcal{N}_x} \sum_{k=1}^{\mathcal{N}_z} \mathbb{E}\left[ \exp\left( \frac{\varepsilon_{j,k}^2}{4\bar{C}^2} \right) \right] \leq 2\sqrt{2}\mathcal{N}_x \mathcal{N}_z,$$

so that

$$\mathbb{E}_{\mathcal{D}_1}\left[ \sup_{(j,k) \in \hat{\mathcal{C}}} \varepsilon_{j,k}^2 \right] \leq \mathbb{E}_{\mathcal{D}_1}\left[ \sup_{j \leq \mathcal{N}_x, k \leq \mathcal{N}_z} \varepsilon_{j,k}^2 \right] \leq 4\bar{C}^2 \log 2\sqrt{2}\mathcal{N}_x \mathcal{N}_z.$$

Moreover for each $(j,k) \in \hat{\mathcal{C}}$, the pair $(\psi_{\theta_{x,j}}, \phi_{\theta_{z,k}})$ constitutes a joint empirical approximator of some $\psi_{\theta_x^{\circ}} \in \mathcal{F}_x$ so that

$$\sum_{i=1}^{m} \left( \phi_{\theta_{z,k}}(z_i) - T\psi_{\theta_{x,j}}(z_i) \right)^2 \leq \sum_{i=1}^{m} \left( \hat{\mathbb{E}}_{X|Z}[\psi_{\theta_x^{\circ}}](z_i) - T\psi_{\theta_x^{\circ}}(z_i) \right)^2 + (4\bar{C} + 2)m(\delta_x + \delta_z).$$

Therefore, (38) is bounded in expectation as

$$\mathbb{E}_{\mathcal{D}_1}\left[ \sup_{(j,k) \in \hat{\mathcal{C}}} \frac{2}{m} \sum_{i=1}^{m} \left( \phi_{\theta_{z,k}}(z_i) - T\psi_{\theta_{x,j}}(z_i) \right) \xi_{j,i} \right]$$

$$= \mathbb{E}_{\mathcal{D}_1}\left[ \frac{2}{m} \sup_{(j,k) \in \hat{\mathcal{C}}} \sqrt{\sum_{i=1}^{m}(\phi_{\theta_{z,k}}(z_i) - T\psi_{\theta_{x,j}}(z_i))^2} \, \varepsilon_{j,k} \right]$$

$$\leq \frac{2}{m} \mathbb{E}_{\mathcal{D}_1}\left[ \sup_{(j,k) \in \hat{\mathcal{C}}} \varepsilon_{j,k}^2 \right] + \mathbb{E}_{\mathcal{D}_1}\left[ \sup_{(j,k) \in \hat{\mathcal{C}}} \frac{1}{2m} \sum_{i=1}^{m} \left( \phi_{\theta_{z,k}}(z_i) - T\psi_{\theta_{x,j}}(z_i) \right)^2 \right]$$

$$\leq \frac{8\bar{C}^2}{m} \log 2\sqrt{2}\mathcal{N}_x \mathcal{N}_z + \mathbb{E}_{\mathcal{D}_1}\left[ \sup_{\theta_x \in \Theta_x} \frac{1}{2m} \sum_{i=1}^{m} \left( T\psi_{\theta_x}(z_i) - \hat{\mathbb{E}}_{X|Z}[\psi_{\theta_x}](z_i) \right)^2 \right]$$

$$+ (2\bar{C} + 1)(\delta_x + \delta_z).$$

In particular, the second term is simply half of the supremal Stage 1 error to be bounded.

Furthermore, almost the same argument for (39) with the cover $\mathcal{C}^*$ yields the bound

$$
\mathbb{E}_{\mathcal{D}_1}\left[\sup_{(j,k)\in\mathcal{C}^*} \frac{2}{m}\sum_{i=1}^m \left(T\psi_{\theta_{x,j}}(z_i) - \phi_{\theta_{z,k}}(z_i)\right)\xi_{j,i}\right]
$$
$$
\leq \frac{8\bar{C}^2}{m}\log 2\sqrt{2}\mathcal{N}_x\mathcal{N}_z + \mathbb{E}_{\mathcal{D}_1}\left[\sup_{(j,k)\in\mathcal{C}^*}\frac{1}{2m}\sum_{i=1}^m\left(T\psi_{\theta_{x,j}}(z_i) - \phi_{\theta_{z,k}}(z_i)\right)^2\right],
$$

where the second term can be bounded by half the sum of (37) and

$$
\sup_{(j,k)\in\mathcal{C}^*}\|T\psi_{\theta_{x,j}} - \phi_{\theta_{z,k}}\|^2_{L^2(\mathcal{P}_{\mathcal{Z}})} \leq \sup_{\theta_x\in\Theta_x}\left[\inf_{\theta_z\in\Theta_z}\|T\psi_{\theta_x} - \phi_{\theta_z}\|^2_{L^2(\mathcal{P}_{\mathcal{Z}})}\right] + (4\bar{C}+2)(\delta_x + \delta_z),
$$

similarly as before (a tighter bound can be obtained with more analysis since $\mathcal{C}^*$ is not data-dependent, but this does not affect the result).

## D.5 Bounding Supremal Deviations (37), (40)

We derive the bound for (40) first. Define the auxiliary functions

$$
g_{j,k}(z) := \left(T\psi_{\theta_{x,j}}(z) - \phi_{\theta_{z,k}}(z)\right)^2 - \|T\psi_{\theta_{x,j}} - \phi_{\theta_{z,k}}\|^2_{L^2(\mathcal{P}_{\mathcal{Z}})}
$$

and set $\kappa_{j,k} := \|T\psi_{\theta_{x,j}} - \phi_{\theta_{z,k}}\|_{L^2(\mathcal{P}_{\mathcal{Z}})} \vee \kappa_0$ for some $\kappa_0 > 0$. For each $(j,k)\in\hat{\mathcal{C}}$ comprising a joint empirical approximator for some $\psi_{\theta_x^\circ}\in\mathcal{F}_x$, it holds that

$$
\kappa^2_{j,k} \leq \|T\psi_{\theta_{x,j}} - \phi_{\theta_{z,k}}\|^2_{L^2(\mathcal{P}_{\mathcal{Z}})} + \kappa_0^2
$$
$$
\leq \|T\psi_{\theta_x^\circ} - \hat{\mathbb{E}}_{X|Z}[\psi_{\theta_x^\circ}]\|^2_{L^2(\mathcal{P}_{\mathcal{Z}})} + (4\bar{C}+2)(\delta_x + \delta_z) + \kappa_0^2
$$

so that

$$
\sup_{(j,k)\in\hat{\mathcal{C}}}\kappa^2_{j,k} \leq \sup_{\theta_x\in\Theta_x}\|T\psi_{\theta_x} - \hat{\mathbb{E}}_{X|Z}[\psi_{\theta_x}]\|^2_{L^2(\mathcal{P}_{\mathcal{Z}})} + (4\bar{C}+2)(\delta_x + \delta_z) + \kappa_0^2, \tag{41}
$$

retrieving the supremal Stage 1 error to be bounded. In addition, it holds for all $z$ that

$$
\left|\frac{g_{j,k}(z)}{\kappa_{j,k}}\right| \leq \frac{4\bar{C}^2}{\kappa_0}, \quad \sum_{i=1}^m\mathbb{E}\left[\left(\frac{g_{j,k}(z)}{\kappa_{j,k}}\right)^2\right] \leq 4\bar{C}^2 m.
$$

Then for the sum over $(z_i)_{i=1}^m$, by Bernstein's inequality we have the tail bound

$$
P\left(\left|\sum_{i=1}^m\frac{g_{j,k}(z_i)}{\kappa_{j,k}}\right| \geq u\right) \leq 2\exp\left(-\frac{u^2}{8\bar{C}^2(m + u/3\kappa_0)}\right)
$$

for all $u > 0$. Hence the random variable

$$
G := \sup_{(j,k)\in\hat{\mathcal{C}}}\left|\sum_{i=1}^m\frac{g_{j,k}(z_i)}{\kappa_{j,k}}\right|
$$

satisfies by a union bound

$$
P(G^2 \geq u) \leq 2|\hat{\mathcal{C}}|\exp\left(-\frac{u}{8\bar{C}^2(m + \sqrt{u}/3\kappa_0)}\right)
$$
$$
\leq 2|\hat{\mathcal{C}}|\exp\left(-\frac{u}{16\bar{C}^2 m}\right) + 2|\hat{\mathcal{C}}|\exp\left(-\frac{3\kappa_0\sqrt{u}}{16\bar{C}^2}\right).
$$

Therefore for a cutoff $u_0 > 0$, we evaluate

$$
\mathbb{E}[G^2] = \int_0^\infty P(G^2 \geq u)\,\mathrm{d}u
$$

$$\leq u_0 + \int_{u_0}^{\infty} 2|\hat{\mathcal{C}}| \exp\left(-\frac{u}{16\bar{C}^2 m}\right) \mathrm{d}u + \int_{u_0}^{\infty} 2|\hat{\mathcal{C}}| \exp\left(-\frac{3\kappa_0 \sqrt{u}}{16\bar{C}^2}\right) \mathrm{d}u$$

$$= u_0 + 32|\hat{\mathcal{C}}|\bar{C}^2 m \exp\left(-\frac{u_0}{16\bar{C}^2 m}\right)$$

$$+ 4|\hat{\mathcal{C}}|\left(\frac{16\bar{C}^2\sqrt{u_0}}{3\kappa_0} + \frac{256\bar{C}^4}{9\kappa_0^2}\right)\exp\left(-\frac{3\kappa_0\sqrt{u_0}}{16\bar{C}^2}\right),$$

where we have used the fact that the antiderivative of $\exp(-\sqrt{u})$ is $-2(\sqrt{u}+1)\exp(-\sqrt{u})$. We now choose $\kappa_0, u_0$ such that

$$\frac{u_0}{16\bar{C}^2 m} = \frac{3\kappa_0\sqrt{u_0}}{16\bar{C}^2} = \log|\hat{\mathcal{C}}| \quad \Leftrightarrow \quad \kappa_0 = \frac{4\bar{C}}{3}\sqrt{\frac{\log|\hat{\mathcal{C}}|}{m}}, \quad u_0 = 16\bar{C}^2 m \log|\hat{\mathcal{C}}|,$$

which yields

$$\mathbb{E}[G^2] \leq 16\bar{C}^2 m \log|\hat{\mathcal{C}}| + 32\bar{C}^2 m + 4\left(16\bar{C}^2 m + \frac{16\bar{C}^2 m}{\log|\hat{\mathcal{C}}|}\right) \leq 16\bar{C}^2 m(\log|\hat{\mathcal{C}}| + 10). \quad (42)$$

Combining (41) and (42) and substituting in the value for $\kappa_0$, it follows that

$$\mathbb{E}_{\mathcal{D}_1}\left[\sup_{(j,k)\in\hat{\mathcal{C}}}\left|\frac{1}{m}\sum_{i=1}^{m} g_{j,k}(z_i)\right|\right]$$

$$\leq \frac{1}{m}\mathbb{E}_{\mathcal{D}_1}\left[G\sup_{(j,k)\in\hat{\mathcal{C}}}\kappa_{j,k}\right] \leq \frac{\mathbb{E}[G^2]}{2m^2} + \frac{1}{2}\mathbb{E}_{\mathcal{D}_1}\left[\sup_{(j,k)\in\hat{\mathcal{C}}}\kappa_{j,k}^2\right]$$

$$\leq \frac{80\bar{C}^2(\log|\hat{\mathcal{C}}|+9)}{9m} + \frac{1}{2}\mathbb{E}_{\mathcal{D}_1}\left[\sup_{\theta_x\in\Theta_x}\|T\psi_{\theta_x} - \hat{\mathbb{E}}_{X|Z}[\psi_{\theta_x}]\|_{L^2(\mathcal{P}_{\mathcal{Z}})}^2\right] + (2\bar{C}+1)(\delta_x + \delta_z).$$

Furthermore, a similar argument for (37) with the cover $\mathcal{C}^*$ gives the bound

$$\mathbb{E}_{\mathcal{D}_1}\left[\sup_{(j,k)\in\mathcal{C}^*}\left|\frac{1}{m}\sum_{i=1}^{m} g_{j,k}(z_i)\right|\right]$$

$$\leq \frac{80\bar{C}^2(\log|\mathcal{C}^*|+9)}{9m} + \frac{1}{2}\sup_{\theta_x\in\Theta_x}\left[\inf_{\theta_z\in\Theta_z}\|T\psi_{\theta_x} - \phi_{\theta_z}\|_{L^2(\mathcal{P}_{\mathcal{Z}})}^2\right] + (2\bar{C}+1)(\delta_x + \delta_z).$$

**Remark D.3.** The above uniform bounds can also be obtained up to constants via localization and chaining techniques, see e.g. Theorem 14.1 and Corollary 14.3 of Wainwright (2019). For this approach, we control deviations of empirical and population $L^2$-norms uniformly over the class

$$\mathcal{F}_{\hat{\mathcal{C}}} := T[\mathcal{F}_x] - \mathcal{F}_z = \left\{T\psi_{\theta_x} - \phi_{\theta_z} \mid (j,k) \in \hat{\mathcal{C}}\right\}.$$

One caveat is that the function class is usually required to be star-shaped, that is $f \in \mathcal{F}_{\hat{\mathcal{C}}}$ should imply $\alpha f \in \mathcal{F}_{\hat{\mathcal{C}}}$ for all $\alpha \in [0,1]$. However $\mathcal{F}_{\hat{\mathcal{C}}}$ is in fact only *nearly* star-shaped; while the output of a DNN in $\mathcal{F}_x, \mathcal{F}_z$ can generally be scaled by scaling the parameters of the final layer, the clip operation (when activated) prevents this for certain functions. This can be overcome by slightly extending $\mathcal{F}_x, \mathcal{F}_z$ to incorporate a scalable clipping $\mathrm{clip}_{\alpha\mathcal{C},\alpha\bar{C}}$.

### D.6  PUTTING THINGS TOGETHER

Plugging in the obtained bounds for (38) and (39), we have that

$$\mathbb{E}_{\mathcal{D}_1}\left[\sup_{\theta_x\in\Theta_x}\frac{1}{m}\sum_{i=1}^{m}\left(T\psi_{\theta_x}(z_i) - \hat{\mathbb{E}}_{X|Z}[\psi_{\theta_x}](z_i)\right)^2\right]$$

$$\leq \frac{3}{2}\sup_{\theta_x\in\Theta_x}\left[\inf_{\theta_z\in\Theta_z}\|T\psi_{\theta_x} - \phi_{\theta_z}\|_{L^2(\mathcal{P}_{\mathcal{Z}})}^2\right] + (36\bar{C}+6)(\delta_x + \delta_z)$$

$$+ \frac{3}{2}\mathbb{E}_{\mathcal{D}_1}\left[\sup_{(j,k)\in\mathcal{C}^*}\left|\frac{1}{m}\sum_{i=1}^m\left(T\psi_{\theta_{x,j}}(z_i)-\phi_{\theta_{z,k}}(z_i)\right)^2-\|T\psi_{\theta_{x,j}}-\phi_{\theta_{z,k}}\|_{L^2(\mathcal{P}_{\mathcal{Z}})}^2\right|\right]$$

$$+ \frac{16\bar{C}^2}{m}\log 2\sqrt{2}\mathcal{N}_x\mathcal{N}_z + \mathbb{E}_{\mathcal{D}_1}\left[\sup_{\theta_x\in\Theta_x}\frac{1}{2m}\sum_{i=1}^m\left(T\psi_{\theta_x}(z_i)-\hat{\mathbb{E}}_{X|Z}[\psi_{\theta_x}](z_i)\right)^2\right]$$

and so, substituting in the bound for (37) as well,

$$\mathbb{E}_{\mathcal{D}_1}\left[\sup_{\theta_x\in\Theta_x}\frac{1}{m}\sum_{i=1}^m\left(T\psi_{\theta_x}(z_i)-\hat{\mathbb{E}}_{X|Z}[\psi_{\theta_x}](z_i)\right)^2\right]$$

$$\leq 3\sup_{\theta_x\in\Theta_x}\left[\inf_{\theta_z\in\Theta_z}\|T\psi_{\theta_x}-\phi_{\theta_z}\|_{L^2(\mathcal{P}_{\mathcal{Z}})}^2\right] + \frac{32\bar{C}^2}{m}\log 2\sqrt{2}\mathcal{N}_x\mathcal{N}_z + (72\bar{C}+12)(\delta_x+\delta_z)$$

$$+ 3\mathbb{E}_{\mathcal{D}_1}\left[\sup_{(j,k)\in\mathcal{C}^*}\left|\frac{1}{m}\sum_{i=1}^m\left(T\psi_{\theta_{x,j}}(z_i)-\phi_{\theta_{z,k}}(z_i)\right)^2-\|T\psi_{\theta_{x,j}}-\phi_{\theta_{z,k}}\|_{L^2(\mathcal{P}_{\mathcal{Z}})}^2\right|\right]$$

$$\leq \frac{9}{2}\sup_{\theta_x\in\Theta_x}\left[\inf_{\theta_z\in\Theta_z}\|T\psi_{\theta_x}-\phi_{\theta_z}\|_{L^2(\mathcal{P}_{\mathcal{Z}})}^2\right]$$

$$+ \frac{176\bar{C}^2}{3m}(\log\mathcal{N}_x+\log\mathcal{N}_z+9)+(78\bar{C}+15)(\delta_x+\delta_z)$$

since $|\mathcal{C}^*|\leq\mathcal{N}_x\mathcal{N}_z$. Combining with the bound for (40) yields

$$\mathbb{E}_{\mathcal{D}_1}\left[\sup_{\theta_x\in\Theta_x}\|T\psi_{\theta_x}-\hat{\mathbb{E}}_{X|Z}[\psi_{\theta_x}]\|_{L^2(\mathcal{P}_{\mathcal{Z}})}^2\right]$$

$$\leq \mathbb{E}_{\mathcal{D}_1}\left[\sup_{\theta_x\in\Theta_x}\left|\frac{1}{m}\sum_{i=1}^m\left(T\psi_{\theta_x}(z_i)-\hat{\mathbb{E}}_{X|Z}[\psi_{\theta_x}](z_i)\right)^2-\|T\psi_{\theta_x}-\hat{\mathbb{E}}_{X|Z}[\psi_{\theta_x}]\|_{L^2(\mathcal{P}_{\mathcal{Z}})}^2\right|\right]$$

$$+ \mathbb{E}_{\mathcal{D}_1}\left[\sup_{\theta_x\in\Theta_x}\frac{1}{m}\sum_{i=1}^m\left(T\psi_{\theta_x}(z_i)-\hat{\mathbb{E}}_{X|Z}[\psi_{\theta_x}](z_i)\right)^2\right]$$

$$\leq \frac{80\bar{C}^2(\log|\hat{\mathcal{C}}|+9)}{9m} + \frac{1}{2}\mathbb{E}_{\mathcal{D}_1}\left[\sup_{\theta_x\in\Theta_x}\|T\psi_{\theta_x}-\hat{\mathbb{E}}_{X|Z}[\psi_{\theta_x}]\|_{L^2(\mathcal{P}_{\mathcal{Z}})}^2\right] + (10\bar{C}+4)(\delta_x+\delta_z)$$

$$+ \mathbb{E}_{\mathcal{D}_1}\left[\sup_{\theta_x\in\Theta_x}\frac{1}{m}\sum_{i=1}^m\left(T\psi_{\theta_x}(z_i)-\hat{\mathbb{E}}_{X|Z}[\psi_{\theta_x}](z_i)\right)^2\right]$$

$$\leq \frac{9}{2}\sup_{\theta_x\in\Theta_x}\left[\inf_{\theta_z\in\Theta_z}\|T\psi_{\theta_x}-\phi_{\theta_z}\|_{L^2(\mathcal{P}_{\mathcal{Z}})}^2\right] + \frac{1}{2}\mathbb{E}_{\mathcal{D}_1}\left[\sup_{\theta_x\in\Theta_x}\|T\psi_{\theta_x}-\hat{\mathbb{E}}_{X|Z}[\psi_{\theta_x}]\|_{L^2(\mathcal{P}_{\mathcal{Z}})}^2\right]$$

$$+ \frac{68\bar{C}^2}{m}(\log\mathcal{N}_x+\log\mathcal{N}_z+9)+(88\bar{C}+19)(\delta_x+\delta_z),$$

and thus we obtain the upper bound for the Stage 1 supremal error as

$$\mathbb{E}_{\mathcal{D}_1}\left[\sup_{\theta_x\in\Theta_x}\|T\psi_{\theta_x}-\hat{\mathbb{E}}_{X|Z}[\psi_{\theta_x}]\|_{L^2(\mathcal{P}_{\mathcal{Z}})}^2\right]$$

$$\leq 9\sup_{\theta_x\in\Theta_x}\left[\inf_{\theta_z\in\Theta_z}\|T\psi_{\theta_x}-\phi_{\theta_z}\|_{L^2(\mathcal{P}_{\mathcal{Z}})}^2\right] + C_2\left(\frac{\log\mathcal{N}_x+\log\mathcal{N}_z}{m}+\delta_x+\delta_z\right). \qquad (43)$$

for some constant $C_2$ depending only on $\bar{C}$.

Finally combining this with (26) and (27), we conclude that the projected $L^2$ error is bounded as

$$\mathbb{E}_{\mathcal{D}_1,\mathcal{D}_2}\left[\|Tf_{\text{str}}-T\hat{f}_{\text{str}}\|_{L^2(\mathcal{P}_{\mathcal{Z}})}^2\right]$$

$$\lesssim \inf_{\theta_x\in\Theta_x}\|Tf_{\text{str}}-T\psi_{\theta_x}\|_{L^2(\mathcal{P}_{\mathcal{Z}})}^2 + \sup_{\theta_x\in\Theta_x}\left[\inf_{\theta_z\in\Theta_z}\|T\psi_{\theta_x}-\phi_{\theta_z}\|_{L^2(\mathcal{P}_{\mathcal{Z}})}^2\right]$$

$$+ \frac{\log\mathcal{N}_x}{m}+\frac{\log\mathcal{N}_z}{m\wedge n}+\delta_x+\delta_z. \qquad (44)$$

### D.7 ERROR RATES FOR DNN CLASSES

Thus far, we have shown that the error can be bounded in terms of the (projected) Stage 2 approximation error, supremal Stage 1 approximation error, and covering numbers for the Stage 1 and 2 estimator function classes. We now evaluate each of these quantities for the introduced DNN classes to prove the final result. Concretely, for sufficiently large integers $N_x, N_z$ we set

$$\mathcal{F}_x = \mathcal{F}_{\mathrm{DNN}}(\lceil \log_2 d_x \rceil + 1, O(N_x), O(N_x), \mathrm{poly}(N_x)),$$
$$\mathcal{F}_z = \mathcal{F}_{\mathrm{DNN}}(\lceil \log_2 d_z \rceil + 1, O(N_z), O(N_z), \mathrm{poly}(N_z)),$$

as specified in Theorem C.8. As a technical note, we must also set the lower clip cutoff $\underline{C}_z$ of $\mathcal{F}_z$ to be greater than the higher cutoff $\bar{C}$ of $\mathcal{F}_x$. This is to ensure that the target of Stage 1, the conditional mean $T\psi_{\theta_x}$ which has sup norm bounded by $\bar{C}$, is contained in the identity region of the clip for $\mathcal{F}_z$ and thus can be properly learned.

First, the projected Stage 2 approximation error (26) can be evaluated as follows. Let the $N_x$-term B-spline decomposition of $f_{\mathrm{str}}$ according to Lemma C.7 be

$$f_{N_x} = \sum_{k=0}^{K} \sum_{\ell \in I_k} \beta_{k,\ell} \omega_{k,\ell} + \sum_{k=K+1}^{K^*} \sum_{i=1}^{n_k} \beta_{k,\ell_i} \omega_{k,\ell_i}.$$

By the adaptive recovery method given in Dũng (2011), it holds that $2^{K d_x} \asymp N_x$. Moreover, if the residual component $g = \Pi_r^{/K}(f_{\mathrm{str}} - f_{N_x})$ along $P_r^{/K}$ satisfies $g \neq 0$, by redefining $f_{N_x}$ as $g + f_{N_x}$ (modifying the coefficients of the B-splines up to resolution $K$ if necessary), we can ensure that $\Pi_r^{/K}(f_{\mathrm{str}} - f_{N_x}) = 0$, and the approximation error of $f_{\mathrm{str}}$ does not increase due to the Pythagorean theorem. It follows from Assumption 3 that

$$\|T(f_{\mathrm{str}} - f_{N_x})\|_{L^2(\mathcal{P}_{\mathcal{Z}})} \lesssim 2^{-\gamma_1 K}\|f_{\mathrm{str}} - f_{N_x}\|_{L^2(\mathcal{P}_{\mathcal{X}})} \lesssim N_x^{-\gamma_1/d_x} N_x^{-s/d_x}.$$

Here, we have bounded the $L^2(\mathcal{P}_{\mathcal{X}})$-norm by the $L^\infty(\mathcal{X})$-norm in the continuous regime $s \geq d_x/p$ and by the $L^2(\mathcal{X})$-norm in the discontinuous regime $s < d_x/p$ via Assumption 2.

On the other hand, by the proof of Theorem C.8, there exists a sigmoid neural network $\check{f}_{N_x}$ such that

$$\|\check{f}_{N_x} - f_{N_x}\|_{W_p^r(\mathcal{X})} + \|\check{f}_{N_x} - f_{N_x}\|_{L^\infty(\mathcal{X})} \lesssim \mathrm{poly}(N_x)\epsilon$$

with weights at most $\mathrm{poly}(\epsilon^{-1})$. Thus choosing $\epsilon$ so that this error is dominated by $N_x^{-(s+\gamma_1)/d_x}$, possibly by increasing the norm bound of $\mathcal{F}_x$ by a $\mathrm{poly}(N_x)$ factor, we can ensure

$$\|Tf_{\mathrm{str}} - T\check{f}_{N_x}\|_{L^2(\mathcal{P}_{\mathcal{Z}})} \leq \|T(f_{\mathrm{str}} - f_{N_x})\|_{L^2(\mathcal{P}_{\mathcal{Z}})} + \|\check{f}_{N_x} - f_{N_x}\|_{L^2(\mathcal{P}_{\mathcal{X}})} \lesssim N_x^{-(s+\gamma_1)/d_x}$$

for some $\check{f}_{N_x} \in \mathcal{F}_x$. Since $\|\check{f}_{N_x}\|_{B_{p,q}^s(\mathcal{X})}$ is bounded, this construction is valid even with domain restriction (10) for a suitable $C_W$.

Next, under Assumption 4* or Assumption 4 with domain restriction and $C_T = \Theta(C_W)$, it holds for all $\theta_x \in \Theta_x$ that $T\psi_{\theta_x} \in C_T \cdot \mathbb{U}(B_{p',q'}^{s'}(\mathcal{Z}))$ by replacing $\Theta_x$ by (10) if necessary. We also have that

$$\|T\psi_{\theta_x}\|_{L^\infty(\mathcal{Z})} \leq \|\psi_{\theta_x}\|_{L^\infty(\mathcal{X})} \leq \bar{C} < \underline{C}_z.$$

Thus by Theorem C.8 (slightly modified to account for the constant factor $C_T$), we are guaranteed the existence of $\check{f}_{N_z} \in \mathcal{F}_z$ satisfying $\|\check{f}_{N_z} - T\psi_{\theta_x}\|_{L^\infty(\mathcal{Z})} \lesssim N_z^{-s'/d_z}$, so that we also have

$$\sup_{\theta_x \in \Theta_x} \left[ \inf_{\theta_z \in \Theta_z} \|T\psi_{\theta_x} - \phi_{\theta_z}\|_{L^2(\mathcal{P}_{\mathcal{Z}})}^2 \right] \lesssim N_z^{-2s'/d_z}.$$

Furthermore, it follows from Lemma C.9 that $\mathcal{N}_x \lesssim N_x \log(\delta_x^{-1} N_x)$ and $\mathcal{N}_z \lesssim N_z \log(\delta_z^{-1} N_z)$. Hence by setting $\delta_x \asymp N_x^{-(s+\gamma_1)/d_x}$, $\delta_z \asymp N_z^{-2s'/d_z}$ and substituting in (44), it follows that

$$\mathbb{E}_{\mathcal{D}_1, \mathcal{D}_2}\left[\|Tf_{\mathrm{str}} - T\hat{f}_{\mathrm{str}}\|_{L^2(\mathcal{P}_{\mathcal{Z}})}^2\right] \lesssim N_x^{-\frac{2s+2\gamma_1}{d_x}} + N_z^{-\frac{2s'}{d_z}} + \frac{N_x \log N_x}{m} + \frac{N_z \log N_z}{m \wedge n},$$

and taking $N_x \asymp m^{\frac{d_x}{2s+2\gamma_1+d_x}}$ and $N_z \asymp (m \wedge n)^{\frac{d_z}{2s'+d_z}}$, we finally conclude:

$$\mathbb{E}_{\mathcal{D}_1, \mathcal{D}_2}\left[\|Tf_{\mathrm{str}} - T\hat{f}_{\mathrm{str}}\|_{L^2(\mathcal{P}_{\mathcal{Z}})}^2\right] \lesssim m^{-\frac{2s+2\gamma_1}{2s+2\gamma_1+d_x}} \log m + (m \wedge n)^{-\frac{2s'}{2s'+d_z}} \log(m \wedge n).$$

If ReLU DNNs are used instead for $\mathcal{F}_z$, the approximation error and covering number estimates are replaced by Proposition 1 and Lemma 3 of Suzuki (2019), respectively. In this case, the depth must scale as $\log(m \wedge n)$ and the sparsity also incurs an additional log factor. The resulting rates are the same except that the log factor must be replaced by $\log^3$ (rather than $\log^2$ suggested in the paper).

# E PROOF OF THEOREM 3.5

## E.1 ADDING SMOOTHNESS REGULARIZATION

The Stage 2 objective with a nonnegative regularizer $R : \Theta_x \to \mathbb{R}_{\geq 0}$ and regularization strength $\lambda > 0$ reads

$$\hat{\theta}_x = \operatorname*{arg\,min}_{\theta_x \in \Theta_x} \frac{1}{n} \sum_{i=1}^{n} \left( \tilde{y}_i - \hat{\mathbb{E}}_{X|Z}[\psi_{\theta_x}](\tilde{z}_i) \right)^2 + \lambda R(\theta_x),$$

where we take $R$ as in (11) later on. We begin by modifying the oracle inequality (Lemma D.1) to include regularization.

**Lemma E.1.** *For $\mathcal{N}_z := \mathcal{N}(\mathcal{F}_z, \|\cdot\|_{L^\infty(\mathcal{Z})}, \delta_z)$, there exists a constant $C_1$ such that for all $\delta_z > 0$ with $\log \mathcal{N}_z > 1$ it holds conditional on $\mathcal{D}_1$,*

$$\mathbb{E}_{\mathcal{D}_2} \left[ \|T f_{\mathrm{str}} - \hat{\mathbb{E}}_{X|Z}[\psi_{\hat{\theta}_x}]\|_{L^2(\mathcal{P}_{\mathcal{Z}})}^2 + \lambda R(\hat{\theta}_x) \right]$$

$$\leq 4 \inf_{\theta_x \in \Theta_x} \left( \|T f_{\mathrm{str}} - \hat{\mathbb{E}}_{X|Z}[\psi_{\theta_x}]\|_{L^2(\mathcal{P}_{\mathcal{Z}})}^2 + \lambda R(\theta_x) \right) + C_1 \left( \frac{\log \mathcal{N}_z}{n} + \delta_z \right).$$

*Proof.* For the quantities

$$\Pi(\theta_x) = \|T f_{\mathrm{str}} - \hat{\mathbb{E}}_{X|Z}[\psi_{\theta_x}]\|_{L^2(\mathcal{P}_{\mathcal{Z}})}^2 + \lambda R(\theta_x),$$

$$\hat{\Pi}(\theta_x) = \mathbb{E}_{\mathcal{D}_2} \left[ \frac{1}{n} \sum_{i=1}^{n} \left( T f_{\mathrm{str}}(\tilde{z}_i) - \hat{\mathbb{E}}_{X|Z}[\psi_{\theta_x}](\tilde{z}_i) \right)^2 \right] + \lambda R(\theta_x),$$

it still follows that

$$\left| \hat{\Pi}(\hat{\theta}_x) - \Pi(\hat{\theta}_x) \right| \leq \frac{1}{2} \|T f_{\mathrm{str}} - \hat{\mathbb{E}}_{X|Z}[\psi_{\hat{\theta}_x}]\|_{L^2(\mathcal{P}_{\mathcal{Z}})}^2 + C_1' \left( \frac{\log \mathcal{N}_z}{n} + \delta_z \right)$$

$$\leq \frac{1}{2} \Pi(\hat{\theta}_x) + C_1' \left( \frac{\log \mathcal{N}_z}{n} + \delta_z \right)$$

since $R$ is nonnegative. Moreover for all $\theta_x \in \Theta_x$, from the inequality

$$\frac{1}{n} \sum_{i=1}^{n} \left( \tilde{y}_i - \hat{\mathbb{E}}_{X|Z}[\psi_{\hat{\theta}_x}](\tilde{z}_i) \right)^2 + \lambda R(\hat{\theta}_x) \leq \frac{1}{n} \sum_{i=1}^{n} \left( \tilde{y}_i - \hat{\mathbb{E}}_{X|Z}[\psi_{\theta_x}](\tilde{z}_i) \right)^2 + \lambda R(\theta_x),$$

we have

$$\hat{\Pi}(\hat{\theta}_x) \leq \Pi(\theta_x) + \mathbb{E}_{\mathcal{D}_2} \left[ \frac{2}{n} \sum_{i=1}^{n} (\tilde{y}_i - T f_{\mathrm{str}}(\tilde{z}_i)) \left( \hat{\mathbb{E}}_{X|Z}[\psi_{\hat{\theta}_x}](\tilde{z}_i) - \hat{\mathbb{E}}_{X|Z}[\psi_{\theta_x}](\tilde{z}_i) \right) \right]$$

$$\leq \Pi(\theta_x) + \mathbb{E}_{\mathcal{D}_2} \left[ \frac{1}{2n} \sum_{i=1}^{n} \left( T f_{\mathrm{str}}(\tilde{z}_i) - \hat{\mathbb{E}}_{X|Z}[\psi_{\theta_x}](\tilde{z}_i) \right)^2 \right] + C_1'' \left( \frac{\log \mathcal{N}_z}{n} + \delta_z \right)$$

$$\leq \Pi(\theta_x) + \frac{1}{2} \hat{\Pi}(\hat{\theta}_x) + C_1'' \left( \frac{\log \mathcal{N}_z}{n} + \delta_z \right).$$

Combining the two inequalities concludes the statement. $\qquad \square$

We now complete the proof of Theorem 3.1 with regularization. By inserting additional $R$ terms, the projected error can be bounded similarly as in Section D.1 as

$$\mathbb{E}_{\mathcal{D}_2} \left[ \|T f_{\mathrm{str}} - T \hat{f}_{\mathrm{str}}\|_{L^2(\mathcal{P}_{\mathcal{Z}})}^2 + \lambda R(\hat{\theta}_x) \right]$$

$$\leq 2 \mathbb{E}_{\mathcal{D}_2} \left[ \|T f_{\mathrm{str}} - \hat{\mathbb{E}}_{X|Z}[\psi_{\hat{\theta}_x}]\|_{L^2(\mathcal{P}_{\mathcal{Z}})}^2 + \lambda R(\hat{\theta}_x) \right] + 2 \mathbb{E}_{\mathcal{D}_2} \left[ \|\hat{\mathbb{E}}_{X|Z}[\psi_{\hat{\theta}_x}] - T \hat{f}_{\mathrm{str}}\|_{L^2(\mathcal{P}_{\mathcal{Z}})}^2 \right]$$

$$\leq 8 \inf_{\theta_x \in \Theta_x} \left( \|T f_{\mathrm{str}} - \hat{\mathbb{E}}_{X|Z}[\psi_{\theta_x}]\|_{L^2(\mathcal{P}_{\mathcal{Z}})}^2 + \lambda R(\theta_x) \right) + 2 C_1 \left( \frac{\log \mathcal{N}_z}{n} + \delta_z \right)$$

$$+ 2\mathbb{E}_{\mathcal{D}_2}\left[\|\hat{\mathbb{E}}_{X|Z}[\psi_{\hat{\theta}_x}] - T\hat{f}_{\text{str}}\|^2_{L^2(\mathcal{P}_{\mathcal{Z}})}\right]$$

$$\leq 16\|Tf_{\text{str}} - T\psi_{\theta_x^*}\|^2_{L^2(\mathcal{P}_{\mathcal{Z}})} + 16\|T\psi_{\theta_x^*} - \hat{\mathbb{E}}_{X|Z}[\psi_{\theta_x^*}]\|^2_{L^2(\mathcal{P}_{\mathcal{Z}})} + 8\lambda R(\theta_x^*)$$

$$+ 2\|T\psi_{\hat{\theta}_x} - \hat{\mathbb{E}}_{X|Z}[\psi_{\hat{\theta}_x}]\|^2_{L^2(\mathcal{P}_{\mathcal{Z}})} + 2C_1\left(\frac{\log \mathcal{N}_z}{n} + \delta_z\right).$$

Note that we have not reduced the Stage 1 error for $\hat{\theta}_x$ and $\theta_x^*$ to the supremum over $\Theta_x$. Indeed, we may retrace the arguments in Section D.2 through D.6 without reducing any of the terms to the corresponding supremum over $\hat{\mathcal{C}}, \mathcal{C}^*$ or $\Theta_x$ but retaining their specific value (e.g. the specific joint empirical or population approximators) for $\hat{\theta}_x, \theta_x^*$ until the final step. Then we see that the Stage 1 error bound (43) also holds with the supremal approximation error replaced by the pointwise error for the estimate,

$$\mathbb{E}\left[\|T\psi_{\hat{\theta}_x} - \hat{\mathbb{E}}_{X|Z}[\psi_{\hat{\theta}_x}]\|^2_{L^2(\mathcal{P}_{\mathcal{Z}})}\right]$$

$$\leq 9\mathbb{E}\left[\inf_{\theta_z \in \Theta_z}\|T\psi_{\hat{\theta}_x} - \phi_{\theta_z}\|^2_{L^2(\mathcal{P}_{\mathcal{Z}})}\right] + C_2\left(\frac{\log \mathcal{N}_x + \log \mathcal{N}_z}{m} + \delta_x + \delta_z\right),$$

and similarly for $\theta_x^*$. Moreover taking $\mathcal{F}_x, \mathcal{F}_z$ as in Section D.6 and applying Theorem C.8 as before, the Stage 1 approximation error can be bounded as

$$\inf_{\theta_z \in \Theta_z}\|T\psi_{\theta_x} - \phi_{\theta_z}\|_{L^2(\mathcal{P}_{\mathcal{Z}})} \lesssim N_z^{-s'/d_z}\|T\psi_{\theta_x}\|_{B^{s'}_{p',q'}(\mathcal{Z})} \lesssim N_z^{-s'/d_z}\|\psi_{\theta_x}\|_{B^s_{p,q}(\mathcal{X})}$$

for both $\hat{\theta}_x, \theta_x^*$. Plugging this and (11) into the above, we obtain that

$$\mathbb{E}\left[\|Tf_{\text{str}} - T\hat{f}_{\text{str}}\|^2_{L^2(\mathcal{P}_{\mathcal{Z}})} + \lambda R(\hat{\theta}_x)\right]$$

$$\lesssim \|Tf_{\text{str}} - T\psi_{\theta_x^*}\|^2_{L^2(\mathcal{P}_{\mathcal{Z}})} + \lambda R(\theta_x^*) + N_z^{-2s'/d_z}\left(\|\psi_{\theta_x^*}\|^2_{B^s_{p,q}(\mathcal{X})} + \mathbb{E}\left[\|\psi_{\hat{\theta}_x}\|^2_{B^s_{p,q}(\mathcal{X})}\right]\right)$$

$$+ \frac{\log \mathcal{N}_x}{m} + \frac{\log \mathcal{N}_x}{m \wedge n} + \delta_x + \delta_z.$$

Again by Theorem C.8, there exists $\theta_x^* \in \Theta_x$ such that $\|Tf_{\text{str}} - T\psi_{\theta_x^*}\|_{L^2(\mathcal{P}_{\mathcal{Z}})} \lesssim N_x^{-(s+\gamma_1)/d_x}$ and $\|\psi_{\theta_x^*}\|_{B^s_{p,q}(\mathcal{X})}$ is bounded, so that $R(\theta_x^*)$ is also bounded. Hence taking $N_x \asymp m^{\frac{d_x}{2s+2\gamma_1+d_x}}$ and $N_z \asymp (m \wedge n)^{\frac{d_z}{2s'+d_z}}$ as before, it holds that

$$\mathbb{E}\left[\|Tf_{\text{str}} - T\hat{f}_{\text{str}}\|^2_{L^2(\mathcal{P}_{\mathcal{Z}})} + \lambda R(\hat{\theta}_x)\right]$$

$$\lesssim N_x^{-\frac{2s+2\gamma_1}{d_x}} + \lambda + N_z^{-\frac{2s'}{d_z}}\mathbb{E}\left[\|\psi_{\hat{\theta}_x}\|^2_{B^s_{p,q}(\mathcal{X})}\right] + N_z^{-\frac{2s'}{d_z}} + \frac{N_x \log N_x}{m} + \frac{N_z \log N_z}{m \wedge n}$$

$$\lesssim m^{-\frac{2s+2\gamma_1}{2s+2\gamma_1+d_x}}\log m + \lambda + (m \wedge n)^{-\frac{2s'}{2s'+d_z}}\left(\mathbb{E}\left[\|\psi_{\hat{\theta}_x}\|^2_{B^s_{p,q}(\mathcal{X})}\right] + \log(m \wedge n)\right).$$

By further setting

$$\lambda \asymp m^{-\frac{2s+2\gamma_1}{2s+2\gamma_1+d_x}}\log m + (m \wedge n)^{-\frac{2s'}{2s'+d_z}}\log(m \wedge n), \tag{45}$$

we can guarantee that

$$\mathbb{E}\left[\|Tf_{\text{str}} - T\hat{f}_{\text{str}}\|^2_{L^2(\mathcal{P}_{\mathcal{Z}})} + \lambda R(\hat{\theta}_x)\right] \lesssim \lambda\left(\mathbb{E}\left[|\psi_{\hat{\theta}_x}|^2_{B^s_{p,q}(\mathcal{X})}\right] + 1\right).$$

In particular, since $\bar{q} > 2$, isolating the regularizer yields by Jensen's inequality

$$\mathbb{E}[R(\hat{\theta}_x)] \lesssim \mathbb{E}\left[|\psi_{\hat{\theta}_x}|^2_{B^s_{p,q}(\mathcal{X})}\right] + 1 \leq \mathbb{E}[R(\hat{\theta}_x)]^{2/\bar{q}} + 1$$

for both choices in (11), and hence $\mathbb{E}[R(\hat{\theta}_x)]$ must be bounded above. From this and the preceding inequality, we conclude that $\mathbb{E}\left[\|Tf_{\text{str}} - T\hat{f}_{\text{str}}\|^2_{L^2(\mathcal{P}_{\mathcal{Z}})}\right] \lesssim \lambda$. $\qquad\square$

### E.2 OBTAINING NON-PROJECTED RATES

For a sufficiently large threshold $N$, let $f_N$ be the $N$-term B-spline approximation of $f_{\text{str}}$ in Lemma C.7 such that $\|f_{\text{str}} - f_N\|_{L^2(\mathcal{P}_{\mathcal{X}})} \lesssim N^{-s/d_x}$. Again, we have bounded the $L^2(\mathcal{P}_{\mathcal{X}})$-norm by the $L^\infty(\mathcal{X})$-norm if $s \geq d_x/p$ and by the $L^2(\mathcal{X})$-norm if $s < d_x/p$. Since $p \geq 2$, it suffices to take $K^* = K$ so that $f_N \in P_r^{/K}$ and $2^{Kd_x} \asymp N$. Similarly, let $\hat{f}_N$ be the $N$-term approximation of $\hat{f}_{\text{str}}$, for which it holds that $\|\hat{f}_{\text{str}} - \hat{f}_N\|_{L^2(\mathcal{P}_{\mathcal{X}})} \lesssim N^{-s/d_x}\|\hat{f}_{\text{str}}\|_{B_{p,q}^s(\mathcal{X})}$. Then we have that

$$\|\hat{f}_{\text{str}} - f_{\text{str}}\|_{L^2(\mathcal{P}_{\mathcal{X}})}$$
$$\lesssim \|\hat{f}_{\text{str}} - \hat{f}_N\|_{L^2(\mathcal{P}_{\mathcal{X}})} + 2^{\gamma_0 K}\|T\hat{f}_N - Tf_N\|_{L^2(\mathcal{P}_{\mathcal{Z}})} + \|f_{\text{str}} - f_N\|_{L^2(\mathcal{P}_{\mathcal{X}})}$$
$$\leq 2^{\gamma_0 K}\left(\|T\hat{f}_{\text{str}} - T\hat{f}_N\|_{L^2(\mathcal{P}_{\mathcal{Z}})} + \|T\hat{f}_{\text{str}} - Tf_{\text{str}}\|_{L^2(\mathcal{P}_{\mathcal{Z}})} + \|Tf_{\text{str}} - Tf_N\|_{L^2(\mathcal{P}_{\mathcal{Z}})}\right)$$
$$\quad + \|\hat{f}_{\text{str}} - \hat{f}_N\|_{L^2(\mathcal{P}_{\mathcal{X}})} + \|f_{\text{str}} - f_N\|_{L^2(\mathcal{P}_{\mathcal{X}})}$$
$$\lesssim \left(2^{(\gamma_0-\gamma_1)K} + 1\right)\left(\|\hat{f}_{\text{str}} - \hat{f}_N\|_{L^2(\mathcal{P}_{\mathcal{X}})} + \|f_{\text{str}} - f_N\|_{L^2(\mathcal{P}_{\mathcal{X}})}\right) + 2^{\gamma_0 K}\|T\hat{f}_{\text{str}} - Tf_{\text{str}}\|_{L^2(\mathcal{P}_{\mathcal{Z}})}$$
$$\lesssim N^{-(s-\gamma_0+\gamma_1)/d_x}\left(\|\hat{f}_{\text{str}}\|_{B_{p,q}^s(\mathcal{X})} + 1\right) + N^{\gamma_0/d_x}\|T\hat{f}_{\text{str}} - Tf_{\text{str}}\|_{L^2(\mathcal{P}_{\mathcal{Z}})}$$

by applying Assumptions 3, 6. Furthermore, $\|\hat{f}_{\text{str}}\|_{B_{p,q}^s(\mathcal{X})}^2$ is bounded above with domain restriction, or bounded in expectation with regularization via the argument in the previous section by taking $\lambda$ as in (45). Squaring both sides and taking expectations, it follows that

$$\mathbb{E}_{\mathcal{D}_1,\mathcal{D}_2}\left[\|\hat{f}_{\text{str}} - f_{\text{str}}\|_{L^2(\mathcal{P}_{\mathcal{X}})}^2\right] \lesssim N^{-\frac{2(s-\gamma_0+\gamma_1)}{d_x}} + N^{\frac{2\gamma_0}{d_x}}\lambda.$$

Finally, setting $N \asymp \lambda^{-\frac{d_x}{2s+2\gamma_1}}$ yields the desired rate. □

**Proof of Corollary 4.2.** When $0 < p < 2$, the approximations $f_N, \hat{f}_N$ are adaptively constructed from B-splines up to resolution $K^* = \lceil C^*K \rceil$; nonetheless, the total number of elements used is bounded by $N \asymp 2^{Kd_x}$, and we can also ensure compatibility with the forward link condition $\Pi_r^{/K}(f_{\text{str}} - f_N) = 0$ as before. Hence the proof above can be repeated to obtain the same rate by applying the extended reverse link condition to the difference $\hat{f}_N - f_N$ which is comprised of at most $2N$ B-splines.

Furthermore, the resulting upper bound (15) when $p < 2$, $\Delta > 0$ is strictly faster than the linear lower bound proved in Theorem 4.1 if (in the case of $\gamma_0 = \gamma_1$)

$$\frac{2s'}{2s'+d_z}\frac{s}{s+\gamma_0} > \frac{2(s-\Delta)}{2(s-\Delta)+2\gamma_1+d_x}$$

which is equivalent to

$$\frac{s'}{d_z} > \frac{(s+\gamma_0)(s-\Delta)}{sd_x+2\Delta\gamma_0} = \frac{(s-\Delta)d_x}{(s-\Delta)d_x+\Delta(2\gamma_0+d_x)}\frac{s+\gamma_0}{d_x}.$$

In the same manner, a separation can be obtained even if $\gamma_0 \neq \gamma_1$; we omit the details for clarity of presentation. □

## F PROOFS OF MINIMAX LOWER BOUNDS

### F.1 PROOF OF PROPOSITIONS 3.3, 3.7

Recall that the NPIR model corresponding to the NPIV model is given as

$$Y = Tf_{\text{str}}(Z) + \eta, \quad \eta = f_{\text{str}}(X) - Tf_{\text{str}}(Z) + \xi, \quad \mathbb{E}[\eta|Z] = 0.$$

It can be shown that NPIV is at least as difficult as NPIR (where the operator $T$ is assumed to be known) in the minimax sense: an estimator for NPIV can be utilized to solve NPIR with the same expected risk by generating the corresponding treatments from the conditional distribution of $X$

given data $Z = z_i$ (Chen & Reiss, 2011). This is true for any $m$, and thus yields a lower bound in $n$ valid even when $m \to \infty$. The goal now is to lower bound the minimax risk over the Besov space $\mathbb{U}(B_{p,q}^s(\mathcal{X}))$ for the non-projected case or $T[\mathbb{U}(B_{p,q}^s(\mathcal{X}))]$ for the projected case, when only samples $\{(\tilde{z}_i, \tilde{y}_i)\}_{i=1}^n$ from the indirect model are available. We obtain this via a modification of the Yang-Barron method (Yang & Barron, 1999).

First note that $\operatorname{supp} \omega_{k,\ell}$ covers $r + 1$ dyadic cubes with side length $2^{-k}$ in each dimension. By e.g. only considering $\ell \in I_k$ such that all components are multiples of $r + 1$, we can construct a subset $J_k \subset I_k$ such that $\operatorname{supp} \omega_{k,\ell}$ are contained in $\mathcal{X}$ and pairwise disjoint for all $\ell \in J_k$ and

$$|J_k| \asymp \frac{|I_k|}{(r+1)^{d_x}} \asymp 2^{kd_x}.$$

Consider the best approximation $\pi_{1,0}$ of $\omega_{1,0}$ w.r.t. the $L^2$-norm on $\operatorname{supp} \omega_{1,0}$ in the span of the B-spline $\omega_{0,0}$ and its integer translates. Note that $P_r^{/k-1}$ is equal to the span of all B-splines with resolution exactly $k - 1$. For each $\omega_{k,\ell}$ with $\ell \in J_k$, its best approximation in $P_r^{/k-1}$ will be the correspondingly scaled and translated version $\pi_{k,\ell}$ of $\pi_{1,0}$, so that

$$\|\omega_{k,\ell} - \Pi_r^{/k-1}\omega_{k,\ell}\|_{L^2(\mathcal{X})} = \|\omega_{k,\ell} - \pi_{k,\ell}\|_{L^2(\mathcal{X})} = 2^{-kd_x/2}\|\omega_{1,0} - \pi_{1,0}\|_2. \tag{46}$$

Now set $f_0 = \tilde{f}_0 \equiv 0$ and define the functions

$$f_v = 2^{-ks}\epsilon \sum_{\ell \in J_k} \beta_{v,\ell}\omega_{k,\ell}, \quad v = 1, \cdots, 2^{|J_k|}$$

and

$$\tilde{f}_v = f_v - \Pi_r^{/k-1} f_v = 2^{-ks}\epsilon \sum_{\ell \in J_k} \beta_{v,\ell}(\omega_{k,\ell} - \pi_{k,\ell}),$$

where $\epsilon$ is a suitably small positive number and $\beta_v = (\beta_{v,\ell})_{\ell \in J_k}$ is an enumeration of the vertices of the hypercube $\{1, -1\}^{|J_k|}$. It follows from the sequence norm equivalence (DeVore & Popov, 1988, Theorem 5.1) that

$$\|f_v\|_{B_{p,q}^s(\mathcal{X})} \asymp 2^{k(s-d_x/p)}\left(\sum_{\ell \in J_k} |2^{-ks}\epsilon\beta_{v,\ell}|^p\right)^{1/p} = 2^{-kd_x/p}|J_k|^{1/p}\epsilon \asymp \epsilon.$$

Also, the coefficients of $\pi_{k,\ell}$ (a linear combination of B-splines $\omega_{k-1,\ell'}$ at resolution $k - 1$) for each $\ell \in J_k$ are different translates of the fixed coefficient sequence for $\pi_{1,0}$. Denoting its $\ell^1$-norm by $A$, it follows that the coefficient of each B-spline $\omega_{k-1,\ell'}$ in the sum

$$\Pi_r^{/k-1} f_v = 2^{-ks}\epsilon \sum_{\ell \in J_k} \beta_{v,\ell}\pi_{k,\ell}$$

is also uniformly bounded by $2^{-ks}\epsilon A$, and so $\Pi_r^{/k-1}\|f_v\|_{B_{p,q}^s(\mathcal{X})} \lesssim \epsilon$ by a similar computation at resolution $k - 1$. Hence

$$\|\tilde{f}_v\|_{B_{p,q}^s(\mathcal{X})} \lesssim \|f_v\|_{B_{p,q}^s(\mathcal{X})} + \|\Pi_r^{k-1} f_v\|_{B_{p,q}^s(\mathcal{X})} \lesssim \epsilon,$$

so we can ensure that each $\tilde{f}_v$ is contained in the target class $\mathbb{U}(B_{p,q}^s(\mathcal{X}))$ by choosing $\epsilon$ to be suitably small.

Furthermore, by the Gilbert-Varshamov bound, there exists a well-separated subset $V_k$ of the index set $\{1, \cdots, 2^{|J_k|}\}$ with $\log|V_k| \asymp |J_k| \asymp 2^{kd_x}$ such that

$$\frac{1}{2^p}\sum_{\ell \in J_k} |\beta_{v,\ell} - \beta_{v',\ell}|^p \gtrsim |J_k|, \quad v \neq v' \in V_k,$$

which guarantees the separation

$$\|\tilde{f}_v - \tilde{f}_{v'}\|_{L^2(\mathcal{X})}^2 \geq 2^{-2ks}\epsilon^2 \sum_{\ell \in J_k : \beta_{v,\ell} \neq \beta_{v',\ell}} 4\|\omega_{k,\ell} - \pi_{k,\ell}\|_{L^2(\mathcal{X})}^2$$

$$\gtrsim 2^{-2ks}\epsilon^2 \cdot |J_k| 2^{-kd_x} \asymp 2^{-2ks}\epsilon^2$$

by (46). The KL divergence between the sample distributions $(\tilde{y}_i)_{i=1}^n$ from the NPIR model with $f_{\mathrm{str}} = \tilde{f}_0, \tilde{f}_v$ is then bounded as

$$
\begin{aligned}
\mathrm{KL}(P_{\tilde{f}_0} \| P_{\tilde{f}_v}) &= \mathbb{E}_{(\tilde{z}_i)_{i=1}^n} \left[ \mathrm{KL}(P_{\tilde{f}_0} | (\tilde{z}_i)_{i=1}^n \| P_{\tilde{f}_v} | (\tilde{z}_i)_{i=1}^n) \right] \\
&\leq \sum_{i=1}^n \frac{\|T\tilde{f}_v\|_{L^2(\mathcal{P}_{\mathcal{Z}})}^2}{2\sigma_0^2} \\
&\lesssim 2^{-2ks}\epsilon^2 \cdot 2^{-2\gamma_1 k} n \frac{\|\tilde{f}_v\|_{L^2(\mathcal{X})}^2}{2\sigma_0^2} \\
&\lesssim 2^{-2k(s+\gamma_1)} n
\end{aligned}
$$

due to Assumption 1 and the link condition. Here we have utilized the near-sparsity of $\omega_{k,\ell}$: extending the definition of B-splines to all locations $\ell \in \mathbb{Z}^{d_x}$, it is easy to see that they form a partition of unity, $\sum_{\ell \in \mathbb{Z}^{d_x}} \omega_{k,\ell} \equiv 1$ and so

$$\|\tilde{f}_v\|_{L^2(\mathcal{X})} \leq \|f_v\|_{L^2(\mathcal{X})} \leq \left\| \sum_{\ell \in I_k} \beta_{v,\ell} \omega_{k,\ell} \right\|_{L^2(\mathcal{X})} \leq \left\| \sum_{\ell \in I_k} \beta_{v,\ell} \omega_{k,\ell} \right\|_{L^\infty(\mathcal{X})} \leq 1.$$

Hence the inequality

$$\frac{1}{|V_k|} \sum_{v \in V_k} \mathrm{KL}(P_{f_0} \| P_{f_v}) \lesssim \log |V_k| \asymp 2^{kd_x}$$

is satisfied by scaling the resolution as $2^{(2s+2\gamma_1+d_x)k} \asymp n$. Finally, applying the Yang-Barron method proves that the $L^2(\mathcal{X})$ minimax rate is lower bounded as

$$\inf_{\hat{f}:\mathrm{NPIR}} \sup_{f_{\mathrm{str}} \in \mathbb{U}(B_{p,q}^s(\mathcal{X}))} \mathbb{E}[\|\hat{f} - f_{\mathrm{str}}\|_{L^2(\mathcal{X})}^2] \gtrsim 2^{-2ks}\epsilon^2 \asymp n^{-\frac{2s}{2s+2\gamma_1+d_x}},$$

proving Proposition 3.7. We can check that the ordinary rate $n^{-\frac{2s}{2s+d_x}}$ is retrieved when $T = \mathrm{id}_{\mathcal{X}}$ and $\gamma_1 = 0$, while the rate becomes much worse if $T$ decays exponentially. The same rate holds in the $L^2(\mathcal{P}_{\mathcal{X}})$-norm under Assumption 5.

For Proposition 3.3, we can also derive the projected lower bound in the same manner by noting that for any estimator $\hat{f}$, $T\hat{f}$ is also a valid estimator for the projected target $Tf_{\mathrm{str}}$ since $T$ is assumed to be known in the NPIR setting. Repeating the same construction given above, since $\tilde{f}_v$ is contained in $P_r^{/k}$, the separation in the projected MSE now becomes

$$\|T\tilde{f}_v - T\tilde{f}_{v'}\|_{L^2(\mathcal{P}_{\mathcal{Z}})}^2 \gtrsim 2^{-2\gamma_0 k} \|\tilde{f}_v - \tilde{f}_{v'}\|_{L^2(\mathcal{P}_{\mathcal{X}})}^2 \gtrsim 2^{-(2s+2\gamma_0)k}\epsilon^2$$

due to the reverse link condition and Assumption 5. Hence we conclude that

$$\inf_{\hat{f}:\mathrm{NPIR}} \sup_{f_{\mathrm{str}} \in \mathbb{U}(B_{p,q}^s(\mathcal{X}))} \mathbb{E}[\|T\hat{f} - Tf_{\mathrm{str}}\|_{L^2(\mathcal{P}_{\mathcal{Z}})}^2] \gtrsim n^{-\frac{2s+2\gamma_0}{2s+2\gamma_1+d_x}}. \qquad \square$$

## F.2 PROOF OF LEMMA 3.2

The metric entropy of Besov spaces is classical:

**Theorem F.1** (Giné & Nickl (2015), Theorem 4.3.36). *If $s > d(1/p - 1/2)_+$, then the Besov norm unit ball $\mathcal{B} = \mathbb{U}(B_{p,q}^s([0,1]^d))$ is relatively compact in $L^2([0,1]^d)$ and*

$$\log \mathcal{N}(\mathcal{B}, L^2([0,1]^d), \delta) \asymp \left(\frac{1}{\delta}\right)^{\frac{d}{s}}, \quad \forall \delta > 0.$$

On the other hand, the construction in the previous section gives a subset of $T[\mathbb{U}(B_{p,q}^s(\mathcal{X}))]$ with log cardinality $\log |V_k| \asymp 2^{kd_x}$ and separation $\|Tf_v - Tf_{v'}\|_{L^2(\mathcal{P}_{\mathcal{Z}})}^2 \gtrsim 2^{-(2s+2\gamma_0)k}\epsilon^2$. Assuming $\mathcal{P}_{\mathcal{Z}}$ has Lebesgue density bounded above, we may take $\delta \asymp 2^{-(s+\gamma_0)k}$ to satisfy

$$\delta < \frac{1}{2} \min_{v \neq v' \in V_k} \|Tf_v - Tf_{v'}\|_{L^2(\mathcal{Z})}$$

so that each $\delta$-ball in the projected image can cover at most one element $Tf_v$. It follows that

$$
\begin{aligned}
\log |V_k| &\lesssim \log \mathcal{N}(T[\mathbb{U}(B^s_{p,q}(\mathcal{X}))], L^2([0,1]^{d_z}), \delta) \\
&\lesssim \log \mathcal{N}(\mathbb{U}(B^{s'}_{p',q'}(\mathcal{Z})), L^2([0,1]^{d_z}), \delta) \\
&\asymp \left(2^{(s+\gamma_0)k}\right)^{\frac{d_z}{s'}},
\end{aligned}
$$

and taking the resolution of the subset $k \to \infty$, the constant factor can be eliminated, concluding that $s'/d_z \le (s + \gamma_0)/d_x$.

Finally, we verify that equality can be achieved if $d_x = d_z$ and $T$ acts on B-splines as $T\omega_{k,\ell} = 2^{-\gamma_0 k}\omega_{k,\ell}$. Let $f$ be an arbitrary element of $\mathbb{U}(B^s_{p,q}(\mathcal{X}))$ with B-spline decomposition

$$
f = \sum_{k=0}^{\infty}\sum_{\ell \in I_k}\beta_{k,\ell}\omega_{k,\ell}, \quad Tf = \sum_{k=0}^{\infty}\sum_{\ell \in I_k}2^{-\gamma_0 k}\beta_{k,\ell}\omega_{k,\ell}.
$$

Then it follows from the sequence norm equivalence that

$$
\begin{aligned}
\|Tf\|_{B^{s+\gamma_0}_{p,q}(\mathcal{Z})} &\asymp \left(\sum_{k=0}^{\infty}\left[2^{k(s+\gamma_0-d_z/p)}\left(\sum_{\ell \in I_k}|2^{-\gamma_0 k}\beta_{k,\ell}|^p\right)^{1/p}\right]^q\right)^{1/q} \\
&\asymp \left(\sum_{k=0}^{\infty}\left[2^{k(s-d_z/p)}\left(\sum_{\ell \in I_k}|\beta_{k,\ell}|^p\right)^{1/p}\right]^q\right)^{1/q} \asymp \|f\|_{B^s_{p,q}(\mathcal{X})},
\end{aligned}
$$

implying that $T[\mathbb{U}(B^s_{p,q}(\mathcal{X}))] \subseteq C_T \cdot \mathbb{U}(B^{s+\gamma_0}_{p,q}(\mathcal{Z}))$ for some constant $C_T$. Hence the $L^2$ rate of contraction specifies the degree of which smoothness is increased under the maximal smoothness assumption. $\qquad\square$

### F.3 PROOF OF THEOREM 4.1

Since the bound is equal to the overall minimax optimal rate when $p \ge 2$, we only consider the case $p < 2$.

Write $\tilde{z} = (\tilde{z}_1, \cdots, \tilde{z}_n)$ for brevity and denote its joint law by $\mathcal{P}_{\tilde{z}}$. We fix $k \ge 0$ and consider the B-spline $\omega_{k,\ell}$ on $\mathcal{X}$ for $\ell \in I_k$. Note that $\|T\omega_{k,\ell}\|_{L^\infty(\mathcal{Z})} \le 1$ and

$$
\|T\omega_{k,\ell}\|_{L^2(\mathcal{P}_\mathcal{Z})} \lesssim 2^{-\gamma_1 k}\|\omega_{k,\ell}\|_{L^2(\mathcal{P}_\mathcal{X})} \lesssim 2^{-\gamma_1 k}\|\omega_{k,\ell}\|_{L^2(\mathcal{X})} \asymp 2^{-k(\gamma_1+d_x/2)}
$$

by the link condition. This implies

$$
U_{\ell,i} := \frac{T\omega_{k,\ell}(\tilde{z}_i)^2}{\|T\omega_{k,\ell}\|^2_{L^2(\mathcal{P}_\mathcal{Z})}} \le C_u 2^{k(2\gamma_1+d_x)}, \quad \mathbb{E}[U^2_{\ell,i}] \le C_u 2^{k(2\gamma_1+d_x)}
$$

for some constant $C_u$. By Bernstein's inequality, it follows that

$$
P\left(\frac{1}{n}\sum_{i=1}^{n}U_{\ell,i} > 2\right) \le \exp\left(-\frac{n^2/2}{\sum_{i=1}^{n}\mathbb{E}[U^2_{\ell,i}] + C_u 2^{k(2\gamma_1+d_x)}n/3}\right)
$$

and by union bounding,

$$
P\left(\sup_{\ell \in I_k}\frac{1}{n}\sum_{i=1}^{n}U_{\ell,i} > 2\right) \lesssim 2^{kd_x}\exp\left(-\frac{3}{8C_u}2^{-k(2\gamma_1+d_x)}n\right).
$$

Now assuming $n \gtrsim 2^{k(2\gamma_1+d_x+\epsilon)}$ $(*)$ for some $\epsilon > 0$, the right-hand side converges to zero as $k \to \infty$, so that the event

$$
\mathcal{E} := \left\{\tilde{z} \in \mathcal{Z}^n : \sup_{\ell \in I_k}\frac{1}{n}\sum_{i=1}^{n}U_{\ell,i} \le 2\right\}
$$

satisfies $P(\mathcal{E}) = 1 - o_k(1)$.

Now as in Section F.1, we may reduce to the NPIR model (4) for known $T$ and only consider estimators of the form

$$\hat{f}_L(x) = \sum_{i=1}^n u_i(x, \tilde{z}) \tilde{y}_i, \quad u_1, \cdots, u_n : \mathcal{X} \times \mathcal{Z}^n \to \mathbb{R}. \tag{47}$$

Denote the corresponding linear minimax rate as

$$\mathcal{R}_L = \inf_{\hat{f}_L : \text{linear}} \sup_{f_{\text{str}} \in \mathbb{U}(B_{p,q}^s(\mathcal{X}))} \mathbb{E}\left[\|\hat{f}_L - f_{\text{str}}\|_{L^2(\mathcal{X})}^2\right]$$

and define the auxiliary function $q(x) = \sum_{i=1}^n \int_{\mathcal{E}} u_i(x, \tilde{z})^2 \, d\mathcal{P}_{\tilde{z}}$. For all $f \in \mathbb{U}(B_{p,q}^s(\mathcal{X}))$, we have from $\tilde{y}_i = Tf(\tilde{z}_i) + \eta_i$ that

$$\mathcal{R}_L \geq \mathbb{E}\left[\|\hat{f}_L - f\|_{L^2(\mathcal{X})}^2\right]$$

$$\geq \mathbb{E}\left[\left\|\sum_{i=1}^n Tf(\tilde{z}_i) u_i(\cdot, \tilde{z}) - f(\cdot)\right\|_{L^2(\mathcal{X})}^2\right] + \sigma_0^2 \cdot \mathbb{E}\left[\sum_{i=1}^n \|u_i(\cdot, \tilde{z})\|_{L^2(\mathcal{X})}^2\right]$$

$$\geq \int_{\mathcal{E}} \int_{\mathcal{X}} \left(\sum_{i=1}^n Tf(\tilde{z}_i) u_i(x, \tilde{z}) - f(x)\right)^2 dx \, d\mathcal{P}_{\tilde{z}} + \sigma_0^2 \int_{\mathcal{X}} q(x) \, dx. \tag{48}$$

Set $I_k^+ = I_k \cap \mathbb{N}^{d_x}$ and partition the domain into rectangles as

$$\mathcal{X} = \bigcup_{\ell \in I_k^+} A_{k,\ell} = \bigcup_{\ell \in I_k^+} \prod_{i=1}^{d_x} \left[\frac{\ell_i - 1}{2^k}, \frac{\ell_i}{2^k}\right).$$

It follows from (48) that there exists $\ell^* \in I_k^+$ satisfying

$$\int_{A_{k,\ell^*}} q(x) \, dx \leq 2^{-kd_x} \sigma_0^{-2} \mathcal{R}_L.$$

To each $A_{k,\ell}$ we will associate the B-spline $\omega_{k,\ell-\ell_0}$ where $\ell_0 = (\lfloor r/2 \rfloor, \cdots, \lfloor r/2 \rfloor)$. It can be seen that there exists a constant $C_r$ depending only on $r$ such that

$$\text{vol}\{x \in A_{0,0} \mid \omega_{0,-\ell_0} \geq C_r\} \geq C_r.$$

Hence for the sets

$$G := \{x \in A_{k,\ell^*} \mid \omega_{k,\ell^*-\ell_0}(x) \geq C_r\},$$
$$H := \{x \in A_{k,\ell^*} \mid q(x) \leq 2\sigma_0^{-2} C_r \mathcal{R}_L\},$$

we have $\text{Vol}\, G \geq 2^{-kd_x} C_r$ and $\text{Vol}(A_{k,\ell^*} \setminus H) \leq 2^{-kd_x - 1} C_r$, so that $\text{Vol}(G \cap H) \geq 2^{-kd_x - 1} C_r$. Moreover from the definition of $\mathcal{E}$, for all $x \in H$ we have

$$\int_{\mathcal{E}} \left(\sum_{i=1}^n T\omega_{k,\ell^*-\ell_0}(\tilde{z}_i) u_i(x, \tilde{z})\right)^2 d\mathcal{P}_{\tilde{z}}$$

$$\leq \int_{\mathcal{E}} \sup_{\ell \in I_k} \sum_{i=1}^n T\omega_{k,\ell}(\tilde{z}_i)^2 \cdot \sum_{i=1}^n u_i(x, \tilde{z})^2 \, d\mathcal{P}_{\tilde{z}}$$

$$\leq 2n \|T\omega_{k,\ell}\|_{L^2(\mathcal{P}_{\mathcal{Z}})}^2 \cdot q(x) \leq \frac{4C_r n}{C_u \sigma_0^2} \cdot 2^{-k(2\gamma_1 + d_x)} \mathcal{R}_L.$$

For a moment, suppose that

$$\frac{4C_r n}{C_u \sigma_0^2} \cdot 2^{-k(2\gamma_1 + d_x)} \mathcal{R}_L \leq \frac{C_r^2}{4}. \tag{49}$$

We apply (48) to the scaled B-spline $f = 2^{-k(s-d_x/p)}\epsilon\omega_{k,\ell^*-\ell_0}$, where $\|f\|_{B_{p,q}^s(\mathcal{X})} \asymp \epsilon$ and $\epsilon = \Theta(1)$ is chosen so that $f \in \mathbb{U}(B_{p,q}^s(\mathcal{X}))$. It follows that

$$
\mathcal{R}_L \geq 2^{-2k(s-d_x/p)}\epsilon^2 \int_{G \cap H} \int_{\mathcal{E}} \left( \sum_{i=1}^{n} T\omega_{k,\ell^*-\ell_0}(\tilde{z}_i)u_i(x,\tilde{z}) - \omega_{k,\ell^*-\ell_0}(x) \right)^2 \mathrm{d}\mathcal{P}_{\tilde{z}}\,\mathrm{d}x
$$

$$
\geq 2^{-2k(s-d_x/p)}\epsilon^2 \int_{G \cap H} \left[ \left( \int_{\mathcal{E}} \omega_{k,\ell^*-\ell_0}(x)^2 \,\mathrm{d}\mathcal{P}_{\tilde{z}} \right)^{1/2} \right.
$$

$$
\left. - \left( \int_{\mathcal{E}} \left( \sum_{i=1}^{n} T\omega_{k,\ell^*-\ell_0}(\tilde{z}_i)u_i(x,\tilde{z}) \right)^2 \mathrm{d}\mathcal{P}_{\tilde{z}} \right)^{1/2} \right]^2 \mathrm{d}x
$$

$$
\geq 2^{-2k(s-d_x/p)}\epsilon^2 \cdot \mathrm{Vol}(G \cap H) \left[ P(\mathcal{E})^{1/2}C_r - \left( \frac{4C_r n}{C_u \sigma_0^2} \cdot 2^{-k(2\gamma_1+d_x)}\mathcal{R}_L \right)^{1/2} \right]^2
$$

$$
\geq \frac{(1-o_k(1))C_r^3\epsilon^2}{8} \cdot 2^{-2k(s-d_x(1/p-1/2))}.
$$

Comparing with (49), we have shown that either of the following must hold:

$$
\mathcal{R}_L \gtrsim \frac{2^{k(2\gamma_1+d_x)}}{n} \quad \text{or} \quad \mathcal{R}_L \gtrsim 2^{-2k(s-\Delta)}.
$$

Finally taking $k$ such that $n \asymp 2^{k(2(s-\Delta)+2\gamma_1+d_x)}$, we can verify that condition $(*)$ is satisfied since $s > \Delta$, and therefore $\mathcal{R}_L \gtrsim n^{-\frac{2(s-\Delta)}{2(s-\Delta)+2\gamma_1+d_x}}$. $\qquad\square$

### F.4  SEPARATION IN PROJECTED RATES

To obtain a lower bound for the projected MSE of linear IV estimators, we require that $\mathcal{Z}$ to be sufficiently spatially covered by the projected class so as to be difficult to learn for non-adaptive estimators; one such sufficient condition is outlined below.

**Theorem F.2.** *Suppose that $d_z \leq d_x$. For any cube $S$ in the dyadic partition of $\mathcal{Z}$ into $2^{kd_z}$ cubes of all side lengths $2^{-k}$, we assume there exists $\ell \in I_k$ such that*

$$
\mathrm{Vol}\left\{ z \in S \mid |T\omega_{k,\ell}(z)| \geq \mu_k \right\} \geq c_1 \mathrm{Vol}\, S, \quad \mu_k = c_2 \cdot 2^{-k(2\gamma_0+d_x-d_z)/2}.
$$

*for constants $c_1, c_2 > 0$. Then under the conditions of Theorem 4.1, the projected linear minimax rate is $\widetilde{\Omega}\left( n^{-\frac{2(s-\Delta)+2\gamma_0}{(2(s-\Delta)+2\gamma_1+d_z)\vee(2\gamma_1+d_x)}} \right)$.*

The threshold $\mu_k$ corresponds to the natural magnitude if the mass $\|T\omega_{k,\ell}\|_{L^2(\mathcal{Z})}^2 \gtrsim 2^{-k(2\gamma_0+d_x)}$ is distributed uniformly on $S$. Since each B-spline can cover asymptotically finitely many cubes at the level $\mu_k$, the implication $d_z \leq d_x$ follows from a counting argument. Hence comparing with Theorem 3.1 when $p < 2$, we conclude that DFIV achieves faster projected rates compared to any linear IV estimator if $\gamma_0 = \gamma_1$, $T$ has maximal smoothness and $d_z > (d_x - 2(s - \Delta)) \vee \frac{s-\Delta+\gamma_0}{s+\gamma_0}d_x$.[5]

*Proof.* The proof is similar to Section F.3. We will show the lower bound for all linear estimators of $Tf_{\mathrm{str}}$ of the form $\hat{g}_L(z) = \sum_{i=1}^n u_i(z,\tilde{z})\tilde{y}_i$, which includes the projection $T\hat{f}_L = \sum_{i=1}^n Tu_i(\cdot,\tilde{z})\tilde{y}_i$ of any NPIR estimator (47). Denote the desired rate by $\mathcal{R}'_L$ and replace the auxiliary function $q(x)$ by $q(z) = \sum_{i=1}^n \int_{\mathcal{E}} u_i(z,\tilde{z})^2 \,\mathrm{d}\mathcal{P}_{\tilde{z}}$. It follows for all $f \in \mathbb{U}(B_{p,q}^s(\mathcal{X}))$ that

$$
\mathcal{R}'_L \geq \int_{\mathcal{E}} \int_{\mathcal{Z}} \left( \sum_{i=1}^{n} Tf(\tilde{z}_i)u_i(z,\tilde{z}) - Tf(x) \right)^2 \mathrm{d}z\,\mathrm{d}\mathcal{P}_{\tilde{z}} + \sigma_0^2 \int_{\mathcal{Z}} q(z)\,\mathrm{d}z.
$$

---

[5]The derived rate shows that it is possible for separation to be achieved even if $d_z \leq d_x - 2(s - \Delta)$ and $s'/d_z < (s + \gamma_0)/d_x$ in certain smoothness regimes. We omit a precise characterization.

Then there exists a cube $S$ in the dyadic partition of $\mathcal{Z}$ satisfying $\int_S q(z)\,\mathrm{d}z \leq 2^{-kd_z}\sigma_0^{-2}\mathcal{R}'_L$. By assumption, there exists $\ell \in I_k$ such that for

$$G = \{z \in S \mid |T\omega_{k,\ell}(z)| \geq \mu_k\},$$
$$H = \{z \in S \mid q(z) \leq 2c_1^{-1}\sigma_0^{-2}\mathcal{R}'_L\},$$

it holds that $\mathrm{Vol}\,G \geq c_1 \cdot 2^{-kd_z}$ and $\mathrm{Vol}(G \cap H) \geq c_1 \cdot 2^{-kd_z-1}$, moreover

$$\int_{\mathcal{E}} \left(\sum_{i=1}^{n} T\omega_{k,\ell^*-\ell_0}(\tilde{z}_i)u_i(x,\tilde{z})\right)^2 \mathrm{d}\mathcal{P}_{\tilde{z}} \leq \frac{4n}{C_u c_1 \sigma_0^2} \cdot 2^{-k(2\gamma_1+d_x)}\mathcal{R}'_L.$$

Then repeating the above line of reasoning gives that either of the following must hold:

$$\mathcal{R}'_L \gtrsim \frac{2^{k(2\gamma_1-2\gamma_0+d_z)}}{n} \quad \text{or} \quad \mathcal{R}'_L \gtrsim 2^{-2k(s-\Delta+\gamma_0)}.$$

If $d_z > d_x - 2(s-\Delta)$, we may take $n \asymp 2^{k(2(s-\Delta)+2\gamma_1+d_z)}$ so that condition $(*)$ is satisfied. If $d_z \leq d_x - 2(s-\Delta)$, we instead take $n \asymp 2^{k(2\gamma_1+d_x+\epsilon)}$ for arbitrary $\epsilon > 0$. The stated lower bound can be verified in both cases. $\qquad\square$

**Remark F.3.** The dependency of the optimal rates on dimension $d_x$ is an intrinsic property of Besov spaces; however, this can be removed by instead considering *mixed* (Schmeisser, 1987) or *anisotropic* Besov spaces (Berkolaiko & Novikov, 1994). By applying the results in Appendices C, D to the appropriate wavelet systems, existing learning-theoretic analyses for these spaces (Suzuki, 2019; Suzuki & Nitanda, 2021) can also be adapted to incorporate smooth DNN classes and obtain the corresponding dimension-free bounds.

