# OpenReview forum: "Optimality and Adaptivity of Deep Neural Features for Instrumental Variable Regression"
_ICLR.cc/2025/Conference — ICLR 2025 Poster_

### Official Review · Reviewer_voVt · 2024-10-29

**Soundness:** 3
**Presentation:** 3
**Contribution:** 3
**Rating:** 6
**Confidence:** 3

**Summary:**

This paper provides theoretical guarantees for the DFIV method, which uses adpative deep features to solve the non-parametric IV regression problem. The authors prove an upper bound for the risk of the DFIV estimator when the true structual function lies in a Besov space. Secondly, the paper presents an information-theoretic lower bounds for NPIV in Besov spaces and shows that DFIV can achieve the minimax optimal rate with a balanced number of Stage 1 and 2 samples. Lastly, the paper demonstrates that DFIV has superior empirical performances when compared with linear fixed-feature methods.

**Strengths:**

- Showed that DFIV can achieve the minimax optimal rates, which is positive result.
- Included an analysis on the sample sizes required for Stage 1 and Stage 2, which can provide more insights into the algorithm.

**Weaknesses:**

- This paper would benefit from including a brief discussion on the technical novelty in the analysis/proofs, which is important since this paper mainly presents theoretical results.
- Section 4 (separation with fixed-feature IV) would also be better supported with some experimental results showcasing the improvement of DFIV over linear estimators.
- DFIV is an iterative method, but the paper did not include any results/rates of convergence, which can be an important practical issue.

**Questions:**

I am curious to see a discussion on the smoothness assumptions that is used in this work and the $\beta$-source condition that is often assumed for inverse problems (and also used to study NPIV).

---

> ### Author Response · Authors · 2024-11-21
>
> Thank you for reviewing our paper and providing valuable suggestions! Below are our responses. We would be happy to further discuss any additional questions that might arise. If we have satisfactorily addressed your questions, we would be grateful if you would consider increasing your score.
>
> **Response to Weaknesses.**
>
> **W1.** Compared to existing analyses for ordinary regression, the two-stage regression problem presents the following new technical challenges. We will make this clearer in the main text.
>
> * We need to prove a novel general two-stage oracle bound (44), which is key to all DFIV upper bounds. The proof is highly nontrivial and involved, and is detailed throughout Appendix D.1-D.6. A proof sketch is also given in Appendix B. The resulting bound is natural and can be applied to any two-stage risk minimization estimator.
> * Since Stage 1 is effectively learning the $T$-projected Stage 2 network, the regularity of the DNN estimator must be controlled to analyze the Stage 1 error, which is a new consideration in this line of analysis. This leads to the discussion of smoothness in Section 3.1.
> * A new smooth approximation analysis for sigmoid DNNs involving the Besov norm is needed, which we develop in Appendix C.
> * Our linear IV minimax lower bounds (Theorems 4.1 and F.2) are also new and establish a novel separation. In comparison, our minimax lower bounds (Proposition 3.3 and 3.7) generalize well-known results in the IV literature (for Sobolev balls, RKHS, etc) to Besov spaces.
>
> **W2.** In the IV literature, DFIV has already been empirically shown to perform well at a variety of tasks. In [1], Section 4.3 and [2], Section 5.2, DFIV is shown to outperform competitors including Kernel IV on demand design and policy evaluation tasks in reinforcement learning such as Behaviour Suite tasks. For example, Mountain Car requires the agent to accelerate right on the left side of the valley and left on the right side to gain momentum, which results in a discontinuous target function. This closely aligns with our results on Section 4, which predicts that DFIV outperforms kernel estimators for target functions with both high and low smoothness regions.
>
> **W3.** Our work focuses on sample complexity rather than optimization analysis. There is no satisfactory optimization theory for DNNs even in the basic supervised setting, so understanding the iterative optimization of two separate DNNs is extremely challenging. Nevertheless, understanding the statistical properties of the ideal solution is still an important problem as optimization can only guarantee that the gap from the ideal solution converges to zero as training progresses.
>
> Practical issues concerning convergence of DFIV have been discussed extensively in [1]. For example, in Appendix F they also considered joint optimization of the Stage 2 loss but found this to be too unstable. Alternatively, a recent paper [3] develops a functional bilevel optimization algorithm with convergence guarantees which can be directly applied to IV regression (Section 4.1). We are also developing a follow-up work analyzing the dynamics of joint optimization for two-stage regression in e.g. two-layer NN settings.
>
> **Response to Questions.**
>
> We thank the reviewer for the interesting technical question. Indeed, our smoothness assumption is closely related to the $\beta$-source condition popular in the nonparametric statistics literature and also the NPIV literature [4][5]. To explicitly explain the relationship, we provide an example in the kernel regression scenario.
>
> Suppose our target function lives in a Besov space with smoothness parameters $s,2,2$ i.e. $f_{str} \in B_{2,2}^s(X)$ and we employ a Matern kernel $k_r$ with parameter $r \geq d_x/2$, inducing a reproducing kernel Hilbert space (RKHS) $H_r(X)$.
> Furthermore, suppose that the marginal distribution is equivalent to the Lebesgue measure, i.e., the density $p$ is both upper and lower bounded by some constant. Define the integral operator $L$ acting on functions as $Lf(x) = \int k_r(y,x)f(y) dp(y)$. The $\beta$-source condition used in nonparametric kernel learning reads $\|L^{-\frac{\beta}{2}}f\|_{L_2(p)} \leq B$.
>
> One can then show that $f_{str} \in B_{2,2}^s(X)$ is equivalent to $\|L^{-\frac{\beta}{2}}f_{str}\|_{L_2(p)} \leq B$ with $\beta = \frac{s}{r}$ [6].
>
>
> [1] Xu et al. Learning Deep Features in Instrumental Variable Regression. ICLR 2021.
>
> [2] Chen et al. On Instrumental Variable Regression for Deep Offline Policy Evaluation. JMLR 2022.
>
> [3] Petrulionyte et al. Functional Bilevel Optimization for Machine Learning. NeurIPS 2024.
>
> [4] Chen et al., On Rate Optimality For Ill-posed Inverse Problems in Econometrics. Econometric Theory 2011.
>
> [5] Chen and Christensen. Optimal Sup-norm Rates and Uniform Inference on Nonlinear Functionals of Nonparametric IV Regression. Quantitative Economics 2018.
>
> [6] Fisher and Steinwart. Sobolev Norm Learning Rates for Regularized Least-Squares Algorithms. JMLR 2020.

---

> > ### Comment · Reviewer_voVt · 2024-11-23
> >
> > Thank you for the detailed response and discussion. I think this is a good paper but since there isn't a score 7, I will keep my current score.

---

### Official Review · Reviewer_HafH · 2024-11-04

**Soundness:** 3
**Presentation:** 2
**Contribution:** 3
**Rating:** 6
**Confidence:** 4

**Summary:**

The article provides a convergence analysis of the DFIV algorithm under certain assumptions and proves its minimax optimality. It compares DFIV with fixed feature methods, demonstrating its superiority in low spatial homogeneity and data efficiency.

**Strengths:**

1.	The convergence and optimality analysis in this paper is necessary to understand the DFIV algorithm.

2.	The motivation is well-presented.

3.	The theoretical analysis is solid and It provides a rigorous foundation for DFIV.

**Weaknesses:**

1.	This article is symbol-dense and hard to follow. It lacks some illustrative diagrams or symbol tables.

2.	The rationality of some assumptions (e.g., Equation (2)), the reasons for transforming problems (e.g., from Equation (3) to (4), and the role of mapping \(T\) are not thoroughly explained in its introduction part. In addition, where and why is Definition 2.2 used… The text lacks explanations for these points, and the necessity of many techniques used is not clearly articulated.

3.	See Questions.

**Questions:**

1.	The article mentions many assumptions; so what does the term "Additional assumption" in the abstract specifically refer to? Is it Equation (2) or others? If this assumption is crucial for the derivation, the abstract should succinctly highlight its necessity rather than merely emphasizing its existence.

2.	Regarding Equation (2), a graphical illustration could be added to provide a more intuitive understanding of this part.

3.	The title mentions "adaptivity." Is there a distinction between "data adaptivity" and "data efficiency"(in abstract)?

4.	Line 41: The functional equation is not direct, and a reference to Equation (4) in Section 2 should be added here.

5.	Comparing existing statistical analyses in studying ordinary regression and here two-stage setting, what’s the difference between these two types of theoretical tools? Under the current setup, what technical challenges are brought?

6.	Line 128-143: Why is Problem/Equation (3) transformed into Problem/Equation (4)? Given that the resulting problem (4) is generally ill-posed and the solution may be ill-behaved or not unique, why is it necessary to introduce the operator \(T\)? Can't the original problem be directly solved? The role of \(T\) needs to be explained in detail here.

---

> ### Author Response · Authors · 2024-11-21
>
> Thank you for your detailed review and insightful suggestions, which have helped us greatly to improve the manuscript! Below are our responses. We would be happy to further discuss any additional questions that might arise. If we have satisfactorily addressed your questions, we would be grateful if you would consider increasing your score.
>
> **Response to Weaknesses.**
>
> **W1.** We have added a Table of Symbols at the beginning of the appendix. Also, a diagram illustrating the causal graph of Eq.(2) has been added to the beginning of the introduction.
>
> **W2.**
> * Eq.(2) expresses the basic setting of IV regression, which is always assumed in modern formulations of IV. Please see Figure 1 for an illustration.
>
> * For Equation (3) to (4), and the role of the mapping $T$, please see our answer to Question 6.
>
> * For Definition 2.2: The B-spline basis is a natural hierarchical basis for Besov spaces. It has been extensively studied in the context of approximation theory and multiresolution analysis due to its localized nature (as opposed to e.g. Fourier basis which is used for global frequency analysis), see e.g. [1]. More recently, it has also been used to analyze the sample complexity of ordinary regression with DNNs [2]. We require a concrete basis to conduct smooth approximation analysis (Appendix C) and compute minimax rates. The localized nature of B-splines also helps us derive the separation with kernel methods (Section 4 and Appendix F.4).
>
> **Response to Questions.**
>
> **Q1.** The additional assumption refers to Assumption 4 on the smoothness of $T$. We have mentioned its necessity in the abstract (it essentially allows us to control the difficulty of the Stage 1 regression).
>
> **Q2.** A diagram illustrating the causal graph of Eq.(2) has been added to the beginning of the introduction, see Figure 1.
>
> **Q3.** Indeed, adaptivity and efficiency refer to two different key aspects of our contributions. **Adaptivity** (of features w.r.t. data) fundamentally enables DFIV to achieve the optimal rate when the target function possesses low spatial homogeneity, while any fixed-feature method is provably worse; these results are presented in Section 4. **Efficiency** in the Stage 1 samples means that $m\approx n$ suffices to achieve the optimal rate in $n$, while previous studies on kernel IV all require $m\gg n$; this is proved in Corollary 3.4, 3.8 and explained in depth in the **Optimal splitting** paragraph (line 450-455). Both results are novel improvements over existing NPIV studies to the best of our knowledge.
>
> **Q4.** We have added that the functional equation follows from taking the conditional expectation w.r.t. $Z$. (Referencing Eq.(4) is also possible, but we feel it is better to explain using words since Eq.(4) comes later in the text.)
>
> (continued)

---

> ### Author Response · Authors · 2024-11-21
>
> (continued)
>
> **Q5.** Compared to existing analyses for ordinary regression, the two-stage regression problem presents the following new technical challenges.
>
> * We need to prove a novel general two-stage oracle bound (44), which is key to all DFIV upper bounds. The proof is highly nontrivial and involved, and is detailed throughout Appendix D.1-D.6. A proof sketch is also given in Appendix B. The resulting bound is natural and can be applied to any two-stage risk minimization estimator.
> * Since Stage 1 is effectively learning the $T$-projected Stage 2 network, the regularity of the DNN estimator must be controlled to analyze the Stage 1 error, which is a new consideration in this line of analysis. This leads to the discussion of smoothness in Section 3.1.
> * A new smooth approximation analysis for sigmoid DNNs involving the Besov norm is needed, which we develop in Appendix C.
> * Our linear IV minimax lower bounds (Theorems 4.1 and F.2) are also new and establish a novel separation. In comparison, our minimax lower bounds (Proposition 3.3 and 3.7) generalize well-known results in the IV literature (for Sobolev balls, RKHS, etc) to Besov spaces.
>
> **Q6.** The original problem Eq.(3) cannot be solved directly by e.g. minimizing $\lVert Y-\hat{f}(X)\rVert^2$ since $X$ is correlated with the noise; it will result in a biased estimator, i.e. $\hat{f}$ will not converge to $f$ even with many samples. Reformulating this as Eq.(4) by isolating the $Z$-independent noise component shows that the data can be essentially seen as generated from the projected function $Tf(Z)$. (Note that this is an equivalent reformulation, so we are not making the problem harder to solve in any way.) This motivates the family of two-stage estimators: learn $T\hat{f}$ in Stage 1 and use this to minimize $\lVert Y-T\hat{f}(Z)\rVert^2$ in Stage 2. This does result in an asymptotically unbiased estimator and in fact achieves the optimal convergence rate as we show in our paper.
>
> Moreover, this shows that $T$ also affects the difficulty of learning (as we essentially receive data from the projection $Tf$ rather than $f$), which is why the optimal rates depend on the spectral decay rate of $T$. As such, the transformation Eq.(4) and the operator $T$ are fundamental to the IV analysis literature; please see [3][4][5] and the references within for further details.
>
> [1] Ronald A. DeVore and Vasil A. Popov. Interpolation of Besov spaces. Transactions of the American Mathematical Society, 305(1):397–414, 1988.
>
> [2] Taiji Suzuki. Adaptivity of deep ReLU network for learning in Besov and mixed smooth Besov spaces: optimal rate and curse of dimensionality. ICLR 2019.
>
> [3] Xiaohong Chen and Markus Reiss. On rate optimality for ill-posed inverse problems in econometrics. Econometric Theory, 27(3):497–521, 2011.
>
> [4] Whitney K. Newey and James L. Powell. Instrumental variable estimation of nonparametric models. Econometrica, 71(5):1565–1578, 2003.
>
> [5] Serge Darolles, Yanqin Fan, Jean-Pierre Florens, and Eric Renault. Nonparametric instrumental regression. Econometrica, 79(5):1541–1565, 2011.

---

> ### Comment · Reviewer_HafH · 2024-11-26
>
> Most of considerations have been resolved, and I will increase my score.

---

### Official Review · Reviewer_1NJE · 2024-11-05

**Soundness:** 3
**Presentation:** 3
**Contribution:** 3
**Rating:** 8
**Confidence:** 2

**Summary:**

**Summary:**
Nonparametric two-stage approaches to instrumental variable regression can be classed as either using fixed features  features (e.g. RKHS, spline, dictionary) or data-adaptive features (e.g. deep neural network). Of the latter approaches. deep feature instrumental variable regression has been shown to empirically perform very well, but lacks theoretical analysis. The authors prove that DFIV achieves minimax optimal rates when the target structural equation belongs to a certain class (Besov space). In order to do so, the authors use standard nonparametric IV assumptions plus one more assuption on the conditioanl distribution of the covariate given the instrument. DFIV is also superior to fixed-feature methods for two other reasons: when the target function has low spatial homogeneity (it can be smooth or bumpy in places), DFIV still attains the optimal rate but fixed feature methods fail. Secondly, the two-stage regression models are more data efficient in the first stage than kernel methods.

**Strengths:**

**Strengths:**
- It is mentioned on line 450 that DFIV is shown to require strictly less data in stage 1 than in stage 2, whereas for kernel IV, an assymetric amount of data is needed in stage 1 and stage 2.
- I think this paper provides some significant results: namely, it provides a rigorous analysis of DFIV and shows advantages in terms of adaptivity, sample efficiency and balacning in stage 1 and 2, and justifies the use of neural features.

**Weaknesses:**

**Questions/Weaknesses:**
- "The above loss generally cannot be directly optimized for θx as ˆθz cannot be backpropagated through". Is this for practical or conceptual reasons? One can imagine either running some iterations of an iterative solver (e.g. Newton's method, SGD, ...) and then backpropagating through the iterates of the solver (often called "unrolling"). Alternatively, one can compute the derivatives using the implicit function theorem, by computing the solution and then plugging directly into a known form. But my guess is that either of these approaches fails to yield a useful gradient in practice?
- Assumption 5 - $\mathcal{P}_{\mathcal{X}}$ has Lebesgue density bounded below. Does this exclude e.g. Gaussian densities? Is this a restrictive assumption? One way I can see this assumption satisfied is if the support is compact and the density is bounded from below (otherwise, without a compact support, I think the density may not even be integrable?)
- As far as I understand, neural network models are used (belonging to class on line 245) as the class in stage 1 and stage 2 problems (7) and (8). It seems to be assumed that a minimum exists (and perhaps is unique?), and even more strongly, that this minimum can be obtained by the algorithm. The derived minimax rates assume that the actual minimum is found, which is not realistic. Or maybe with sufficient overparameterisation, it is realistic?
- The smoothness of stage 2 needs to be controlled, and the authors propose two ways to do this. One is to restrict the function class, which in practice requires somehow discarding ill-behaved bumpy solutions after random intialisation. However, practically instantiating this procedure is not really discussed. Another option is to penalise the Besov norm, and I am not sure how hard it is to actually compute a Besov norm for a given neural network. Are these smoothness penalties/constraints actually practical?
- This paper is very theoretical, containing no experiments. This is a little bit (but probably not unheard of) for ICLR. A venue like COLT might attract readers more strongly aligned with its contents. Alternatively, experiments could be done to actually demonstrate that the assumptions made are realistic. For example, it could be investigated how hard it is in practice for the smoothness of the stage 2 by discarding ill-behaved solutions, or whether a zero stage one (and/or small stage 2 loss) can be obtained with a sufficiently rich neural network. None of these experiments are hinted at.

**Minor:**
- Grammar. "...while the rate in Stage 1 samples m is depends on the smoothness of both..."
- The sentence on line 368 is a bit confusing. Are you saying $\log(\cdot)$ is replaced by $(\log(\cdot))^3$?
- Typo on line 446.

**Questions:**

Please address the weakness/questions above. Referring to the details above:
- Why can't stage 2 derivatives go through to stage 1?
- Is Assumption 5 restrictive (in particular, does it include Gaussian?)
- Is it assumed that stage 1 and stage 2 is solved over this neural network class? How realistic is this assumption?
- Are the smoothness penalties/constraints actually practical, or do they only allow the theory to work? If they are not practical, then perhaps it is overselling to claim that the method of Xu et al., 2021 really achieves the minimax rate, because it is after all a practical algorithm.
- Is it possible to show how mild these assumptions or regularisers are by doing a toy experiment?

---

> ### Author Response · Authors · 2024-11-21
>
> Thank you for your positive assessment of our contribution and valuable comments! Below are our responses.
>
> **Q1.** While unrolling is conceptually possible, the original DFIV paper [1] states "...we do not attempt to backpropagate through the estimate $\hat{\theta}_z$ to do this, however, as this would be too computationally expensive." They also considered joint optimization of the Stage 2 loss w.r.t. $\hat{\theta}_x, \hat{\theta}_z$ but found this to be too unstable. Alternatively, a recent paper [2] develops a functional bilevel optimization algorithm with convergence guarantees which can be directly applied to IV regression (Section 4.1).
>
> **Q2.** Integrability is not an issue since we assumed the support of $X,Z$ is compact at the beginning, although this can be relaxed to the noncompact case by assuming suitable tail decay; hence any lower bounded density suffices. Moreover, Assumption 5 is only required for the projected minimax lower bound (Prop 3.3). It is not needed for the full minimax lower bound (Prop 3.7) nor the DFIV upper bounds (Thms 3.1, 3.5) which are our key contributions.
>
> Intuitively, minimax lower bounds are proved by constructing large well-separated subsets of the target space. Assumption 5 is needed in Prop 3.3 to translate the separation in the projected MSE, only guaranteed in $L^2(P_X)$-norm due to how the reverse link condition is defined, to $L^2(X)$-norm in line 2012. It is not needed for the full lower bound since we are not interested in the projected MSE. (To be completely precise, since the full upper bound is given in the $L^2(P_X)$-norm and the full lower bound is given in the $L^2(X)$-norm, Assumption 5 is also needed to conclude that the rates match, but this is just a technicality of minimax analysis.)
>
> **Q3.** The minimum of Stage 1 always exists since the DNN parameter space is compact (all parameters in $\theta_z$ are bounded above by $M$) and the loss depends continuously on the DNN output. The minimum of Stage 2 is not guaranteed to exist as the output $\hat{\theta}_z$ of Stage 1 may not depend continuously on $\theta_x$, but in this case we may simply take an approximate minimizer $\epsilon$-close to the infimum of the loss and obtain the same result by taking $\epsilon$ small enough. This remark has been added to the main text.
>
> The underlying assumption that the empirical risk minimizer can be found is unavoidable in our sample complexity analysis. There is no satisfactory optimization theory for DNNs even in the basic supervised setting, so understanding the iterative optimization of two separate DNNs is extremely challenging. Nevertheless, understanding the statistical properties of the ideal solution is still an important problem as optimization can only guarantee that the gap from the ideal solution converges to zero as training progresses. We are also developing a follow-up work analyzing the dynamics of joint optimization for two-stage regression in e.g. two-layer NN settings.
>
> **Q4.** While the smoothness restriction/regularization is admittedly unusual, there are similar precedents in neural network training; please see the references on line 348-350. Moreover, the norm is easy to estimate if $s$ is an integer since it is equivalent to the Sobolev norm. This can be obtained by automatically computing the order $s$ derivatives of the network w.r.t. the input and numerically integrating. If $s$ is not an integer, the norm can still be computed with the discretization (line 339) (which can be backpropagated through) or bounded above by the order $\lceil s\rceil$ Sobolev norm (which suffices for the restriction-based approach).
>
> **Q5.** In the IV literature, DFIV has already been empirically shown to excel at a variety of tasks, in particular RL/control tasks with discontinuous/low smoothness target functions ([1], Section 4.3 and [3], Section 5.2), aligning with our results. While additional experiments on the smoothness aspect could be interesting, we have elected to focus on the perspective of statistical learning theory. We also mention that sample complexity papers, such as the original paper on DNN regression [4], usually do not conduct new experiments.
>
> Minor: Thank you for pointing out the mistakes, they have been fixed. Line 368 indeed means that $\log m,\log (m\wedge n)$ must be replaced by $(\log m)^3, (\log (m\wedge n))^3$, respectively.
>
> [1] Xu et al. Learning Deep Features in Instrumental Variable Regression. ICLR 2021.
>
> [2] Petrulionyte et al. Functional Bilevel Optimization for Machine Learning. NeurIPS 2024.
>
> [3] Chen et al. On Instrumental Variable Regression for Deep Offline Policy Evaluation. JMLR 2022.
>
> [4] Taiji Suzuki. Adaptivity of deep ReLU network for learning in Besov and mixed smooth Besov spaces: optimal rate and curse of dimensionality. ICLR 2019.

---

> > ### Comment · Reviewer_1NJE · 2024-11-26
> >
> > Thanks for your response. I still think it would be helpful to try and bridge the gap between theory and practice (e.g. DNN optimisation, actually computing norms and penalties, etc.) with some experiments. But the contributions are nice and worthy of publication as they are. I'll keep my score of 8, and recommendation that this paper be accepted.

---

### Meta-Review · Area_Chair_rw16 · 2024-12-17

**Metareview:**

This paper provides theoretical results for deep feature instrumental variable, such as convergence and adaptivity. All reviewers agree on a positive evaluation and think this is a valuable contribution. It would be valuable to provide some experiments to support the theory so that the work can reach a broader audience at ICLR.

**Additional Comments On Reviewer Discussion:**

The authors addressed each of the issues raised by the reviewers, except the experiments (which were not viewed as critical by the reviewers in any case).

---

### Decision · Program_Chairs · 2025-01-22

Accept (Poster)